



# Why observed and modelled ozone production rates and sensitives differ, a case study at rural site in CHINA

Bowen Zhong[1,#], Bin Jiang[1,#], Jun Zhou[1*], Tao Zhang[2], Duohong Chen[2], Yuhong Zhai[2], Junqing Luo[1], Minhui Deng[1], Mao Xiao[3,4], Jianhui Jiang[5,6], Jing Li[1], Min Shao[1*]

[1]College of Environment and Climate, Institute for Environment and Climate Research, Guangdong-Hongkong-Macau Joint Laboratory of Collaborative Innovation for Environmental Quality, Jinan University, Guangzhou 511443, China
[2]Environmental Key Laboratory of Regional Air Quality Monitoring,Ministry of Ecology and Environment,Guangdong Ecological and Environmental Monitoring Center, Guangzhou 511443, China
[3]Sichuan Academy of Environmental Sciences, Chengdu 610041, China
[4]Biogas Institute of Ministry of Agriculture and Rural Affairs, Chengdu 610041, China
[5]Global Institute for Urban and Regional Sustainability, School of Ecological and Environmental Sciences, East China Normal University, Shanghai 200241, China
[6]Institute of Eco-Chongming, East China Normal University, Shanghai 200241, China

[#] These authors contribute equally to this article

*Correspondence to: Jun Zhou (junzhou@jnu.edu.cn) & Min Shao (mshao@jnu.edu.cn)

**Abstract.** Ground-level ozone ($O_3$) pollution has recently become of increasing concern in China. Studies have shown that conventional models often fail to predict accurately the net $O_3$ production rate ($P(O_3)_{net}$) due to the absence of certain mechanisms, particularly the kinetics from missing reactive volatile organic compounds (VOCs) species, and hence affects the reliability of evaluation for $O_3$ formation sensitivity (OFS). Therefore, we conducted a field observation of $P(O_3)_{net}$ and OFS using a $P(O_3)_{net}$ (NPOPR) detection system based on a dual-channel reaction chamber technique at the Guangdong Atmospheric Supersite of China in Heshan, Pearl River Delta (PRD) in autumn of 2023. The in-situ monitoring data were then compared with results from a zero-dimensional model incorporating the Master Chemical Mechanism (MCM v3.3.1). We tested the model performance by incorporating parameterization for 4 processes including $HO_2$ uptake by ambient aerosols, dry deposition, $N_2O_5$ uptake, and $ClNO_2$ photolysis, and found that the discrepancies between the modelled $P(O_3)_{net}$ ($P(O_3)_{net}$_Mod) and measured data ($P(O_3)_{net}$_Mea) did not change evidently, the maximum daily $P(O_3)_{net}$ differed by ~44.8 %. Meanwhile, we found that the agreement of OFS assessment results between the direct measurements and the modelling study was lower in the $P(O_3)_{net}$ rising phase (08:00-09:00, 63.6%) than in the $P(O_3)_{net}$ stable phase (10:00-12:00, 72.7%) and $P(O_3)_{net}$ declining phase (13:00-17:00, 72.7 %). The only approach to fill the gap between observation and computation was to add possible unmeasured reactive VOCs, especially oxygenated VOCs (OVOCs) in box model, this was true for both $P(O_3)_{net}$ and consequent OFS, hinting clearly at the importance of quantitative understanding the total reactivity of VOCs in $O_3$ chemistry.

## 1 Introduction

Ground-level ozone ($O_3$) pollution has garnered widespread attention due to its adverse effects on human health (Chen et al., 2023), vegetation growth (Wang et al., 2017), and climate change (Li et al., 2016). Since the implementation of the *Air Pollution Prevention and Control Action Plan* by the State Council in 2013, particulate matter pollution in China has significantly decreased. However, ground-level $O_3$ pollution remains severe, and $O_3$ has become the primary pollutant affecting



air quality in China (*China Environmental Status Bulletin*, 2013–2024). The variation in ground-level $O_3$ concentration is influenced by local photochemical production, surface deposition, and transport processes, which the following equation can express:

$$\frac{\partial [O_3]}{\partial t} = P(O_3)_{net} - \frac{u_d}{H}[O_3] - \upsilon \cdot \nabla [O_3] \tag{1}$$

In Equation (1), $\frac{\partial [O_3]}{\partial t}$ represents the change in $O_3$ concentration, $P(O_3)_{net}$ denotes the net $O_3$ photochemical production rate, $u_d$ is the $O_3$ deposition rate, H stands for the mixed layer height, and $\upsilon$ represents the wind speed. The in-situ photochemical production of ground-level $O_3$ primarily results from the photochemical reactions of precursor volatile organic compounds (VOCs) and nitrogen oxides ($NO_X$: $NO+NO_2$) under sunlight. The $P(O_3)_{net}$ is a critical indicator for evaluating local photochemical formation. The budget analysis of ground-level $O_3$ production ($P(O_3)$) and consumption ($D(O_3)$) can be calculated using the following equation:

$$P(O_3) = k_{HO_2+NO}[HO_2][NO] + \sum_i k_{RO_{2,i}+NO}[RO_{2i}][NO]\varphi_i \tag{2}$$

$$D(O_3) = k_{O(^1D)+H_2O}[O(^1D)][H_2O] + k_{OH+O_3}[OH][O_3] + k_{HO_2+O_3}[HO_2][O_3]$$
$$+ \sum_i (k_{O_3+Alkene_i}[O_3][Alkene_i]) + k_{OH+NO_2}[OH][NO_2] + k_{RO_{2i}+NO_2}[RO_{2i}][NO_2] \tag{3}$$

$$P(O_3)_{net} = P(O_3) - D(O_3) \tag{4}$$

Equations (2)–(4) illustrate the nonlinear dependence of the $P(O_3)_{net}$ on the oxidation of precursors generating $HO_X$ ($=OH+HO_2$) (Tong et al., 2025). Here, the $P(O_3)_{net}$ is the difference between $P(O_3)$ and $D(O_3)$, $k_{M+N}$ is the reaction rate constant between two molecules, $\varphi_i$ represents the amount of $NO_2$ generated from the reaction of $RO_{2i}$ with $NO_2$, and $i$ denotes different $RO_2$ species. Currently, mainstream model simulation methods for calculating the $P(O_3)_{net}$ primarily involve indirectly solving radical concentrations. However, existing models cannot fully characterize the complex radical cycling processes in the real atmosphere (Wei et al., 2023). Specifically, the incomplete mechanisms of $RO_X$ ($=OH+HO_2+RO_2$) sources are particularly prominent, and these missing mechanisms affect the accuracy of $RO_2$ and $HO_2$ radical estimations to varying degrees. These include the neglect of contributions from carbonyl compounds, HONO, and OVOCs (Xu et al., 2022), as well as incomplete mechanisms for heterogeneous reactions on aerosol surfaces (Yang et al., 2022), dry deposition (Zhang et al., 2003), nitrosyl chloride photolysis (Whalley et al., 2021), and isomerization of isoprene peroxy radicals (Kanaya et al., 2012) remain inadequately understood. These gaps lead to systematic biases in the simulated $P(O_3)_{net}$ (Woodward-Massey et al., 2023; Tan et al., 2017; Tan et al., 2019), thereby affecting the accurate determination of $O_3$ formation sensitivity (OFS).

It is noteworthy that there is a strong causal relationship between the aforementioned mechanistic biases and the misjudgment of OFS. Studies by Baier et al. (2017) and Tan et al. (2019) found that the observation-based model (OBM) significantly underestimates $P(O_3)_{net}$ under high $NO_X$ conditions, leading to misjudgment of OFS. They pointed out that the unresolved VOC species and unspecified chemical mechanisms in the model are the primary causes of these biases. Similarly,



Whalley et al. (2021) demonstrated that the zero-dimensional (box) model exhibits deviations in simulating $P(O_3)_{net}$ under high VOCs concentrations. Further research by Wang et al. (2024b) highlighted that the contribution of unidentified VOCs reactivity in anthropogenic emissions to $O_3$ formation is severely underestimated, and the missing VOC species and chemical mechanisms in existing models lead to biases in the determination of OFS. Such diagnostic biases in OFS may result in misjudgment of precursor emission reduction measures, thereby affecting the effectiveness of $O_3$ pollution control.

Direct measurement of $P(O_3)_{net}$ based on the dual-reaction chamber technique can address the aforementioned challenges. This concept was first proposed by Jeffries (1971), who suggested determining the real value of the $P(O_3)_{net}$ in ambient air by comparing the difference in $O_X(=O_3+NO_2)$ between a photochemical reaction chamber and a reference chamber. To date, several $P(O_3)_{net}$ detection systems based on the dual-reaction chamber technique have been developed, referred to as measurement of $O_3$ production sensor (MOPS), $O_3$ production rate measurement system (O3PR), $O_3$ production rates instrument (OPRs), net photochemical $O_3$ production rate detection system (NPOPR), Mea-OPR, or $O_3$ production rate-cavity ring-down spectroscopy system (OPR-CRDS) (Baier et al., 2015; Sadanaga et al., 2017; Sklaveniti et al., 2018; Hao et al., 2023; Wang et al., 2024c; Tong et al., 2025). Through practical applications in field observations, scholars generally agree that these detection systems offer rapid stability and high precision, with measurement uncertainties around 10 %. Comparative studies have revealed that the underestimation of the simulated $P(O_3)_{net}$ can reach up to 50 % (Cazorla et al., 2012), highlighting the limitations of existing models in characterizing radical chemistry.

More importantly, the $P(O_3)_{net}$ detection system can diagnose OFS by quantifying changes in the measured $P(O_3)_{net}$ induced by different precursors through precursor addition experiments. Sklaveniti et al. (2018) first detected OFS in Indiana by adding NO to the sampling line of $P(O_3)_{net}$ detection systems, demonstrating the feasibility of directly measuring OFS with this device. Morino et al. (2023) combined a smog chamber with the $P(O_3)_{net}$ detection systems to directly measure OFS under baseline environmental conditions in Tokyo during summer. Chen et al. (2024) proposed the OPR_Adj parameter based on the $P(O_3)_{net}$ detection systems, which, through normalization of photolysis rates, diagnosed that $O_3$ photochemistry in Beijing is under VOCs control. These advancements indicate that the direct measurement method of OFS based on the $P(O_3)_{net}$ not only measures the actual OFS in ambient air but also quantifies the discrepancies between models and measurements.

In this study, we employed the developed NPOPR detection system based on the dual-reaction chamber technique to measure the $P(O_3)_{net}$ and OFS at the Guangdong Atmospheric Supersite of China in Heshan City, Pearl River Delta (PRD), in October 2023. Based on the observational data, we used the box model equipped with the Master Chemical Mechanism (MCM v3.3.1) to simulate the radical chemistry during the observation period. We compared and investigated the differences and influencing factors between the model-simulated values (abbreviated as $P(O_3)_{net}\_Mod$) and the directly measured values (abbreviated as $P(O_3)_{net}\_Mea$) in calculating the $P(O_3)_{net}$ and assessing OFS.

## 2 Methods and materials

### 2.1 Field measurements

Field observations were continuously conducted from 4–26 October 2023 at the Guangdong Atmospheric Supersite of



China in Heshan City, located in northern Jiangmen, Guangdong Province (112.93°E, 22.73°N). The supersite is situated in the downwind area of Guangzhou and Foshan and is characterized by active secondary reactions. It lies at the intersection of forest-agricultural and urban systems, representing a typical rural station. The surrounding area primarily consists of farmland conservation zones and forested areas, with no significant industrial emissions. It is suitable for comprehensive monitoring and research on regional atmospheric complex pollution in the PRD (Mazaheri et al., 2019). The geographical location is shown in Fig. S1.

The $P(O_3)_{net}$ detection system (NPOPR), based on the dual-reaction chamber technique, was used to monitor the $P(O_3)_{net}$ and OFS. This system has been successfully applied in multiple field observation campaigns (Hao et al., 2023; Zhou et al., 2024a; 2024b). The detection system consists of a sampling unit, a monitoring unit, and a data acquisition unit. Ambient air passes through a Teflon particulate filter (7592-104, Whatman, UK) to remove particles larger than 2 μm before entering the dual chambers. The reaction chamber and the reference chamber are made of two identical quartz tubes (inner diameter: 190.5 mm, length: 700 mm, wall thickness: 5 mm). Unlike the reaction chamber, which allows ultraviolet light to penetrate and initiate photochemical reactions, the reference chamber is covered with an ultraviolet protective film (SH2CLAR, 3M, Japan) to block light with wavelengths below 390 nm, thereby preventing $O_3$ formation in the reference chamber. A custom circuit control system alternates the gas flow between the reaction chamber and the reference chamber into the NO reaction tube every 2 minutes, where the $O_3$ is converted to $NO_2$, which is then introduced into a Cavity Attenuated Phase Shift (CAPS)-$NO_2$ analyzer (Aerodyne Research, Inc., Billerica MA, USA). The gas not introduced into the NO reaction tube is expelled through an auxiliary pump. The data acquisition system detects $NO_2$, including both ambient $NO_2$ and $NO_2$ converted from $O_3$. By combining the average residence time ($\tau$) of the gas in the chambers and the difference in $O_X$($\triangle O_X=\triangle(O_3+NO_2)$) between the two chambers, the $P(O_3)_{net\_Mea}$ can be calculated. The mean residence time in the reaction chamber is 0.15 h at the air flow rate of 2.1 L min$^{-1}$. Detailed characterization parameters of the detection system can be found in Hao et al. (2023).

$$P(O_3)_{net}\_Mea = P(O_X) = \frac{\Delta O_X}{\tau} = \frac{[O_X]_{reaction} - [O_X]_{reference}}{\tau} \tag{5}$$

An additional system for the addition of NO or VOCs was added to the NPOPR sampling unit to assess OFS. The OFS was assessed by measuring the changes in $P(O_3)_{net}$ induced by the addition of NO or VOCs, enabling the direct measurement of OFS. A schematic diagram of this principle is shown in Fig. S2. In the experiments for determining OFS through direct measurements (conducted daily from 8:00-18:00), each cycle lasted 1 hour. The first 20 minutes involved the addition of NO (denoted as $P(O_3)_{net}\_Mea^{+NO}$), the next 20 minutes measured the ambient baseline ($P(O_3)_{net}\_Mea$), and the final 20 minutes involved the addition of VOCs (denoted as $P(O_3)_{net}\_Mea^{+VOCs}$). The concentration of NO added was 20 % of the ambient $NO_X$ concentration at the site, while the concentration of VOCs indicators added was 20 % of the ambient VOCs reactivity. During the observation period, from 4–11 October, the VOCs indicators included isopentane as the representative alkane, ethylene and isoprene as the representative alkenes, and toluene as the representative aromatic hydrocarbon. From 13–26 October, ethylene was used as the representative non-methane hydrocarbon (NMHC) and formaldehyde as the representative oxygenated volatile organic compound (OVOC). The average concentrations of added NO and VOCs during the experiments were 2.8 ppbv and





7.0 ppbv, respectively. The sensitivity of $O_3$ production to precursor changes was quantified using the measured OFS, derived from the incremental reactivity (IR) index. IR is defined as the change in $P(O_3)_{net}$ per unit change in precursor concentration ($\Delta S(X)$): a negative IR value indicates that reducing the precursor concentration increases $O_3$ production (e.g., NO titration effect), while a larger absolute IR value suggests higher sensitivity of $O_3$ production to changes in the precursor. The IR was calculated as:

$$IR = \frac{P(O_3)_{net}\_Mea^{+X} - P(O_3)_{net}\_Mea}{\Delta S(X)} = \frac{\Delta P(O_3)_{net}\_Mea^{+X}}{\Delta S(X)} \tag{6}$$

where X represents VOCs or NO, $\Delta P(O_3)_{net}^{+X}$ represents the $P(O_3)_{net}$ values measured during the NO or VOCs addition period minus the $P(O_3)_{net}$ values measured when only injecting ambient air. $\Delta S(X)$ represents the concentration of the NO or VOCs precursor added during the corresponding measurement period.

In addition to $P(O_3)_{net}$ and OFS, hourly data such as $PM_{2.5}$, $O_3$, NO, $NO_2$, $SO_2$, carbon monoxide (CO), photolysis rates ($j_{O^1D}, j_{NO_2}, j_{H_2O_2}, j_{NO_3\_M}, j_{NO_3\_R}, j_{HONO}, j_{HCHO\_M}, j_{HCHO\_R}$), HONO, and VOCs concentrations were monitored (more details about the measurements are shown in Table S1). Hourly observations of conventional meteorological parameters, such as temperature, pressure, relative humidity, wind direction, and wind speed, were sourced from the European Centre for Medium-Range Weather Forecasts (ECMWF).

## 2.2 Box model simulation

This study employed an observation-constrained zero-dimensional photochemical reaction model (Observed 0-D box model) to simulate atmospheric photochemical processes. The chemical mechanism module is the core of the box model, and most mainstream studies use the Master Chemical Mechanism (MCM) nested within the model, incorporating processes such as solar radiation, boundary layer height, atmospheric photochemistry, and dry deposition (Zhang et al., 2022). The OBM model used in this study is AtChem2 (https://atchem.leeds.ac.uk/webapp/), which is equipped with the Master Chemical Mechanism (MCM v3.3.1: https://mcm.york.ac.uk/MCM) to simulate $O_3$ and radical chemistry and analyze their budgets (Wang et al., 2022a; Sommariva et al., 2020). The model includes approximately 143 VOCs, 6,700 chemical species, and over 17,000 reactions. Hourly resolution observational data of $O_3$, NO, $NO_2$, CO, $SO_2$, HONO, VOCs (in total 82 species), meteorological parameters (e.g., temperature, relative humidity, pressure, and boundary layer height), and photolysis rates were used as model constraints. Photolysis rates for unmeasured species were calculated using the Tropospheric Ultraviolet and Visible Radiation Model (TUV v5.3) (Table S2). Additionally, to avoid unreasonable increases in the concentrations of constrained species, a dilution rate of $1/86400$ $s^{-1}$ was applied. Before the simulation, the model underwent a 48-hour pre-run to stabilize unmeasured species (e.g., radicals).

The configuration of model mechanisms was informed by previous research, with a particular focus on the dry deposition processes of key species (e.g., $O_3$, $NO_2$, $SO_2$, $H_2O_2$, $HNO_3$, PAN, and HCHO), the heterogeneous uptake reactions of $HO_2$ and $N_2O_5$, and the Cl· chemistry mechanism. Dry deposition is a critical pathway for the transfer of atmospheric pollutants from



the gas phase to the Earth's surface, significantly influencing the concentration distribution and removal of regional pollutants. Many models have already incorporated this atmospheric physical process (Ma et al., 2022; Chen et al., 2020a). The heterogeneous uptake of $HO_2$, as an important sink, contributes to approximately 10 %–40 % of the global $HO_2$ concentration (Li et al., 2019). Studies have shown that including the heterogeneous uptake mechanism of $HO_2$ in simulations reduces $P(O_3)_{net}$ concentration and alters the sensitivity to VOCs (Zhou et al., 2021; Dyson et al., 2022). Additionally, Cl· enhances atmospheric oxidation, accelerating the $OH$-$HO_2$-$RO_2$ reaction cycle (Ma et al., 2023). By incorporating these mechanisms, this study aims to more accurately simulate the atmospheric chemical processes and their impacts on pollutant concentrations in the PRD region (Zhou et al., 2024a). The configurations of each scenario are as follows: Case A considers only the simplified chemical reaction mechanism from the MCM, excluding dry deposition and heterogeneous reactions; Case B incorporates the $HO_2$ uptake by ambient aerosols mechanism based on Case A; Case C further includes the dry deposition processes of key species on top of Case B; and Case $D_1$ extends Case C by adding the $N_2O_5$ uptake mechanism and Cl· related heterogeneous reaction mechanisms. Detailed simulation parameter settings are provided in the supplementary information (Table S3).

## 2.3 Model performance evaluation

The Index of Agreement (IOA) was used to evaluate the simulation performance (Li et al., 2021).

$$IOA = 1 - \frac{\sum_{i=1}^{n}(O_i - S_i)^2}{\sum_{i=1}^{n}(|O_i - \overline{O}| - |S_i - \overline{O}|)^2} \tag{7}$$

where $O_i$ and $S_i$ represent the observed and simulated values, respectively, $\overline{O}$ denote the mean of the observed values, and n is the number of samples. The IOA ranges from 0 to 1, with higher values indicating better agreement between observed and simulated values. In addition to the IOA, the Pearson correlation coefficient (R), mean bias (MB), normalized mean bias (NMB), root mean square error (RMSE), mean fractional bias (MFB) and mean fractional error (MFE) were used to evaluate the consistency between observed and simulated values (Table S7).

## 2.4 $k_{OH}$

Total OH reactivity ($k_{OH}$) is a crucial indicator of atmospheric chemical cycling and oxidative capacity (Gilman et al., 2009). $k_{OH}$ is defined as the sum of the products of the concentrations of all reactive species $X_i$ that can react with OH radicals and their respective reaction rate constants, calculated as follows:

$$k_{OH} = \sum_i k_{OH+X_i} \cdot [X]_i \tag{8}$$

where $X_i$ includes CO, $NO_X$, and VOCs, among others, and $k_{OH+Xi}$ is the reaction rate constant ($s^{-1}$) between reactive species $X_i$ and OH radicals.

## 2.5 OFP

$O_3$ Formation Potential (OFP) is an indicator used to measure the relative contribution of different VOC species to ground-





level O$_3$ formation (Wu et al., 2020). The formula for OFP is as follows:

$$\text{OFP} = [\text{VOCs}]_i \times \text{MIR}_i \tag{9}$$

where [VOCs]$_i$ represents the concentration of a specific VOC species $i$ (μg m$^{-3}$), and MIR represents the maximum incremental reactivity of the VOC species $i$ (g$_{O_3}$/g$_{VOC}$). MIR is used to characterize the increase in O$_3$ production per unit increase in VOCs under conditions where O$_3$ formation is most sensitive to VOCs.

**2.6 Absolute $P$(O$_3$)$_{net}$ sensitivity**

We calculated the modelled OFS using the absolute $P$(O$_3$)$_{net}$ sensitivity method from Sakamoto et al. (2019). It is defined as the change in $P$(O$_3$)$_{net}$ induced by a percentage increase in O$_3$ precursors. This method facilitates the quantitative assessment of how reductions in O$_3$ precursors contribute to the overall reduction of $P$(O$_3$)$_{net}$ over a period or within a region. The formula is as follows:

$$\text{Absolute } P(\text{O}_3)_{net} = \frac{\delta P(\text{O}_3)}{\delta \ln[X]} = P(\text{O}_3) \frac{\delta P(\text{O}_3)}{\delta \ln[X]} \tag{10}$$

In the equation, [$X$] represents NO$_x$ or VOCs. A positive absolute $P$(O$_3$)$_{net}$ sensitivity indicates that reducing the precursor will lead to a decrease in the $P$(O$_3$)$_{net}$. In contrast, a negative value indicates that reducing the precursor will lead to an increase in the $P$(O$_3$)$_{net}$ (Dyson et al., 2022). In this study, the analysis of absolute $P$(O$_3$)$_{net}$ sensitivity was conducted using the box model. Specifically, the model was used to calculate $P$(O$_3$)$_{net}$ and simulate the input precursor concentrations. The above formula was then applied to compute the simulated absolute $P$(O$_3$)$_{net}$ sensitivity, which was subsequently compared with the measured OFS results.

**3 Results and discussion**

**3.1 Overview of observation campaign**

The supplementary materials (Fig. S3, S4, and Table S4, S5) provide the time series plots, diurnal variation, and daytime averages (daytime: 6:00-18:00) of meteorological parameters, conventional pollutants, photolysis rate constants, NO, $P$(O$_3$)$_{net}$ and hourly VOCs concentrations from 4–26 October 2023, at the Guangdong Atmospheric Supersite of China. The site was located downwind of the Guangzhou-Foshan area, with atmospheric pollutants primarily originating from the northeast. To access daily O$_3$ pollution levels, the maximum daily 8-hour average O$_3$ concentration (MDA8) was employed, in accordance with the Technical Specification for Ambient Air Quality Evaluation (Trial) (HJ 663-2013). In this study, days with MDA8-O$_3$ concentration exceeding the Class II limit stipulated by the Ambient Air Quality Standards (GB3095-2012) were defined as O$_3$ pollution days (with MDA8-O$_3$ concentration limit of 160 μg m$^{-3}$ (equivalent to approximately 81.6 ppbv at 25°C), while others were defined as normal days.

During the whole observation period, there were 6 O$_3$ pollution days (15–17 and 24–26 October 2023). The maximum O$_3$ mixing ratio (136.5 ppbv) occurred at 15:00 on 25 October 2023, while the maximum $P$(O$_3$)$_{net}$ (53.7 ppbv h$^{-1}$) occurred at





10:00 on 24 October 2023. Diurnal variation plots show that $O_3$ and $P(O_3)_{net}$ exhibited single-peak patterns, with $O_3$ peaking at 15:00 and $P(O_3)_{net}$ peaking between 9:00-10:00. On $O_3$ pollution days, the daytime average concentrations of $O_3$ and $P(O_3)_{net}$ during the observation period were $63.2\pm37.6$ ppbv and $14.4\pm13.8$ ppbv h$^{-1}$, respectively, both approximately twice as high

225 as on normal days (daytime average $O_3$: $30.9\pm22.9$ ppbv; daytime average $P(O_3)_{net}$:$7.2\pm9.4$ ppbv h$^{-1}$). The maximum values of directly measured $P(O_3)_{net}$ in different ambient environments in previous studies are listed in Table S6, ranging from 10.5 to 100 ppbv h$^{-1}$, and the measured $P(O_3)_{net}$ values in this study fall within this range, demonstrating the reasonableness of the values measured in this study.

As shown in Fig. S4, the diurnal variation of parameters on $O_3$ pollution days and normal days indicates that the nighttime

230 background concentrations of $O_3$ precursors (TVOC and $NO_X$) are higher on $O_3$ pollution days. However, during the period of strongest sunlight (11:00-14:00), the concentrations of TVOC and $NO_X$ on $O_3$ pollution days are lower than those on normal days. Specifically, on $O_3$ pollution days, the TVOC concentration is 11.4 μg m$^{-3}$, and the $NO_X$ concentration is 13.5 ppbv, while on normal days, the TVOC concentration is 13.7 μg m$^{-3}$, and the $NO_X$ concentration is 14.8 ppbv. This suggests that stronger photochemical reactions occur on $O_3$ pollution days, leading to lower daytime concentrations of precursors compared to normal

235 days. The diurnal variation of NO concentration on $O_3$ pollution days showed an early morning peak at 8:00, rising to 12.2 ppbv and then decreasing to 1.6 ppbv. By comparing the diurnal variation data between $O_3$ pollution days and normal days, we found that both $O_3$ concentrations and $P(O_3)_{net}$ values were significantly higher on $O_3$ pollution days, particularly during the daytime (6:00-18:00). This phenomenon aligns with the conclusion that high temperatures, low humidity, strong radiation, and stable weather conditions favor $O_3$ pollution formation.

240 **3.2 Characteristics of VOC concentrations and composition**

This study analyzed 110 VOC species, examining the contributions of different categories to TVOC concentrations, $k_{OH}$, and daily OFP. We also identified the top 10 VOC species contributing to these three indicators (Fig. S5), aiming to explore the atmospheric presence, chemical reactivity, and environmental impact of VOCs. Additionally, this study used two classification methods to group VOC species. Method 1 divided VOCs into alkynes (1 species), alkanes (27 species), alkenes

245 (11 species), aromatic hydrocarbons (17 species), OVOCs (20 species), halogenated hydrocarbons (33 species), and sulfur-containing VOCs (1 species). Method 2 categorized VOCs into BVOC (Biogenic Volatile Organic Compounds), OVOCs (Oxygenated Volatile Organic Compounds), and AVOCs/NMHC (Anthropogenic Volatile Organic Compounds), with specific classifications shown in Table S4.

During the observation period, the daily average TVOC concentration ranged from 7.2 to 28.9 μg m$^{-3}$. OVOCs

250 contributed the most (40.8 %), followed by halogenated hydrocarbons (20.8 %), aromatic hydrocarbons (18.3 %), alkanes (17.9 %), alkenes (1.7 %), and alkynes (0.5 %). The $k_{OH}$ average value was $12.1\pm3.9$ s$^{-1}$, primarily contributed by OVOCs (62.9 %), followed by halogenated hydrocarbons (10.8 %), alkenes (10.4 %), aromatic hydrocarbons (9.8 %), alkanes (6.0 %), and alkynes (0.1 %). Among the alkenes in the known MCM mechanism, ethylene, as an indicator of VOCs, had the highest





proportion, accounting for 10.7 % of alkenes $k_{OH}$ and 2.8 % of NMHC $k_{OH}$. Formaldehyde, another VOCs indicator, was the most dominant species in OVOCs $k_{OH}$, contributing about 13.3 %. Among VOC species, OVOCs contributed the most to OFP (51.6 %), followed by aromatic hydrocarbons (32.9 %), alkenes (8.0 %), alkanes (6.9 %), halogenated hydrocarbons (0.5 %), and alkynes (0.2 %). The analysis results show that although halogenated hydrocarbons dominate VOCs concentration emissions, their contribution to $O_3$ pollution is low. In contrast, alkenes, despite their lower contribution to VOCs concentration emissions, are important precursors for $O_3$ formation. Based on the comprehensive analysis of VOCs concentration, $k_{OH,}$ and OFP, OVOCs and aromatic hydrocarbons significantly contribute to $O_3$ formation and should be prioritized as key VOC species for $O_3$ pollution control in the PRD region. This result aligns with other related studies in the PRD, such as those in Shenzhen (Yu et al., 2020; Guanghe et al., 2022), Guangzhou (Pei et al., 2022), and Jiangmen (Jing et al., 2024), which indicate that OVOCs and aromatic hydrocarbons are key VOC species for $O_3$ formation.

Overall, toluene, m/p-xylene, formaldehyde, 2-hexanone, ethyl acetate, and tetrahydrofuran consistently ranked in the top 10 VOC species in terms of concentration, $k_{OH}$ and OFP contribution. These VOC species mainly originate from human activities, such as industrial production, solvent use, traffic emissions, and fuel combustion, highlighting the significant impact of anthropogenic sources on $O_3$ pollution (Cai et al., 2010; Yang et al., 2023; Zheng et al., 2019).

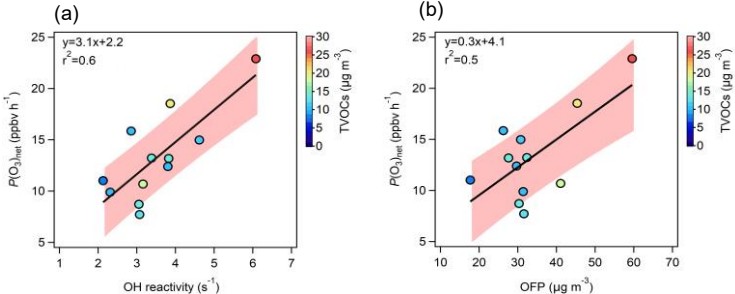

**Figure 1: Correlation between measured $P(O_3)_{net}$ ($P(O_3)_{net}$_Mea) and (a) total OH reactivity ($k_{OH}$) and (b) $O_3$ Formation Potential (OFP).**

Figure 1 shows the correlation between $P(O_3)_{net}$_Mea and $k_{OH}$ and OFP (calculated using the daytime average data during the observation period). The shaded area in the figure represents the confidence interval (68.3 %) of the fitting line between $P(O_3)_{net}$ and $k_{OH}$, and between $P(O_3)_{net}$ and OFP. Data outside the confidence interval may be due to the fact that the calculation of $k_{OH}$ and OFP did not fully consider the environmental conditions and atmospheric chemistry complexity at the observation site (Zhang et al., 2024; Yadav et al., 2024). The color of the scatter points represents the TVOC concentration. The $r^2$ values between $P(O_3)_{net}$ measurements and $k_{OH}$ and OFP are 0.6 and 0.5, respectively, indicating that VOCs with higher $k_{OH}$ and OFP significantly enhance the $P(O_3)_{net}$.

### 3.3 Comparison and optimization of simulated and measured $P(O_3)_{net}$ values

Based on our previous research (Zhou et al., 2024a), we named the scenario considering only the current chemical reaction mechanism from the MCM v3.3.1 in the box model as Case A. Subsequently, we gradually incorporated the $HO_2$ uptake by




ambient aerosols, dry deposition, $N_2O_5$ uptake, and $ClNO_2$ photolysis mechanisms into the MCM mechanism in the box model, implemented as modelling scenarios labeled Case B, Case C, and Case $D_1$. The specific parameter settings for each scenario are shown in Table S3. The time series and diurnal variations of the $P(O_3)_{net}$_Mea and $P(O_3)_{net}$_Mod for Cases A–$D_1$ are shown

in Fig. S7. To evaluate the model's performance, $P(O_3)_{net}$_Mea and $P(O_3)_{net}$_Mod data were used to calculate the IOA, R, MB, NMB, RMSE, MFB and MFE values to access the model's performance under different scenarios (Table S7). The IOA values between $P(O_3)_{net}$_Mod and $P(O_3)_{net}$_Mea were all > 0.86, and R ranged from 0.84–0.98, indicating that the model can reasonably reproduce the changes in $P(O_3)_{net}$. However, the MB and NMB were -3.0–-2.4 ppbv h$^{-1}$ and -30.5 %–24.9 %, respectively, indicating a systematic underestimation of $P(O_3)_{net}$. The RMSE ranged from 7.0 to 7.2 ppbv h$^{-1}$, while the MFB and MFE were

-3.1 %–1.7 % and 53.8 %–55.5 %, respectively. These results suggest that while the model captures the overall trends well, there is room for improvement in reducing simulation biases.

In all modelling scenarios from Case A–Case $D_1$, $P(O_3)_{net}$_Mod values were generally lower than $P(O_3)_{net}$_Mea. Although the correlation between $P(O_3)_{net}$_Mea and $P(O_3)_{net}$_Mod was good (Fig. S9), even after incorporating mechanisms that may compensate for $O_3$ production simulation biases into to the box model (labeled as Case $D_1$), the simulated daytime average

$P(O_3)_{net}$_Mod was still 3.4 ppbv h$^{-1}$ lower than $P(O_3)_{net}$_Mea (26.3 % bias), with a peak deviation of up to 13.3 ppbv h$^{-1}$ (44.8 %), as shown in Fig. 2.we defined the difference between $P(O_3)_{net}$_Mea and $P(O_3)_{net}$_Mod as $P(O_3)_{net}$_Missing, and its time series is shown in Fig. S10. During the observation period, 7–10 October and 18–22 October were rainy days, with a median $P(O_3)_{net}$_Missing < 1.1 ppbv h$^{-1}$; therefore, these days were excluded when calculating the diurnal variations of different $O_3$ production and consumption pathways. On non-rainy days, the median $P(O_3)_{net}$_Missing was 4.9±4.1ppbv h$^{-1}$, with the peak

median $P(O_3)_{net}$_Missing reaching 17.5 ppbv h$^{-1}$ on 15 October 2023. The median $P(O_3)_{net}$_Missing values on $O_3$ pollution days were statistically higher than those on normal days (*t-test*, P<0.5), indicating that the supplementary mechanisms explored in the model, as mentioned above, are not the main cause of the $P(O_3)_{net}$_Missing.

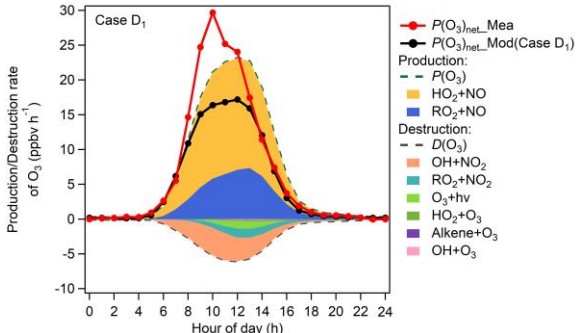

**Figure 2: Diurnal variations (excludes rainy days) of O₃ production and destruction rates modelled in Case D₁, and measured ($P(O_3)_{net}$_Mea) and modelled ($P(O_3)_{net}$_Mod) $P(O_3)_{net}$.**

We further explore the possible reasons for the discrepancies between $P(O_3)_{net}$_Mea and $P(O_3)_{net}$_Mod using the modelling results of Case $D_1$. The ratio of cumulative $P(O_3)_{net}$_Mea and $P(O_3)_{net}$_Mod derived from Case $D_1$ was 1.4, calculated by

summing the daytime data with 1 h resolution during the observation period. This result is close to the previous research





findings, such as Cazorla et al. reporting a ratio of 1.3 (Cazorla and Brune, 2009), Ren et al. (2013) and Hao et al. (2023) reporting a ratio of 1.4. As shown in Fig. 2, we found that the $HO_2+NO$ reaction dominated the $O_3$ production process, accounting for 71.4 % of total $O_3$ production pathways. In contrast, the main pathways for $O_3$ consumption were $OH+NO_2$ and $RO_2+NO_2$, accounting for 67.9 % and 16.5 % of total $O_3$ consumption pathways, respectively. The importance of the $HO_2/RO_2$

reaction pathways indicates that simulation biases in $HO_2/RO_2$ can further affect the accuracy of $P(O_3)_{net}$_Mod.

To further investigate the causes of $P(O_3)_{net}$_Missing, we conducted correlation analyses between $P(O_3)_{net}$_Missing and parameters such as TVOC, $NO_X$, $J_{O1D}$, $T$ and $O_X$ under both $O_3$ pollution days and normal days; the results are shown in Fig. S11. Under $O_3$ pollution days, $P(O_3)_{net}$_Missing showed a positive correlation with VOCs and $NO_X$, with $r^2$ values of 0.4 and 0.5, respectively, indicating that $P(O_3)_{net}$_Missing increases significantly at higher $O_3$ precursor concentrations. This

phenomenon is consistent with previous studies (Whalley et al., 2021; Ren et al., 2013; Zhou et al., 2024a). Meanwhile, $P(O_3)_{net}$_Missing exhibited a positive correlation with $J_{O1D}$ on both $O_3$ pollution days and normal days, with $r^2$ values of 0.5 and 0.4, respectively. In contrast, $P(O_3)_{net}$_Missing showed no significant correlation with TVOC, $NO_X$, $T$, or $O_X$ under normal days ($r^2 < 0.2$). This suggests that $P(O_3)_{net}$_Missing may be related to the missing mechanisms for photolyzable unknown VOCs in the box model. Research by Wang et al. (2022b) indicates that constraining OVOCs in the model is crucial for the accuracy

of $P(O_3)_{net}$_Mod, and photochemical models without OVOCs constraints significantly underestimate $P(O_3)_{net}$. In our previous study on the industrial city of Dongguan (Zhou et al., 2024b), we used parameter equations developed by Wang et al. (2024a; 2024b) to quantify the impact of missing $k_{OH}$ on $P(O_3)_{net}$_Missing and qualitatively tested the potential compensating effects of unmeasured acetaldehyde, acrolein, acetone, butanone, and branched alkenes in Dongguan on $P(O_3)_{net}$_Missing. Compared to the campaign conducted in Dongguan (Zhou et al., 2024a), this study measured more VOC species (Table S4). Therefore,

we further compensate for the Case $D_1$ scenario by constraining more measured VOC species compared to the study in Dongguan (e.g., OVOCs, halogenated hydrocarbons) to explore their impact on $P(O_3)_{net}$_Mod. The specific simulation scenario settings are described in Table S3.

Figure 3 shows the time series and diurnal variations of $P(O_3)_{net}$_Mea and $P(O_3)_{net}$_Mod (under Case $D_1$–$D_4$) during the observation period. Specifically, we added constraints for measured acetaldehyde, acrolein, acetone, and butanone (OVOCs,

which were considered as potential contributors for $P(O_3)_{net}$_Missing in Dongguan in our previous study, Zhou et al., 2024) to the model based on Case $D_1$, which is labeled as Case $D_2$. However, the daytime average value of $P(O_3)_{net}$_Mod from Case $D_2$ increased by only 0.5 % compared to Case $D_1$, this indicates that the dominant OVOCs species that causes $P(O_3)_{net}$_Missing may vary between Heshan and Dongguan. We further constrained all measured OVOC species in Heshan (which included additional OVOCs species compared to that added to Case $D_2$, such as propionaldehyde, butyraldehyde, and valeraldehyde)

that could be input into the box model in the Case $D_3$ simulation scenario (more details can be found in Table S8). The results showed that the averaged daytime $P(O_3)_{net}$_Mod from Case $D_3$ increased by 4.4 % compared to that in Case $D_2$. Notably, in Case $D_3$, constraining all OVOC species significantly improved $P(O_3)_{net}$_Mod during the morning period (8:00-9:00), with an increasing rate of approximately 10.2 % (~1.3 ppbv h$^{-1}$). Additionally, Case $D_4$ scenario added constraints for chlorine-containing VOCs (i.e., all measured VOC species listed in Table S8 that could be input into the OBM model were constrained).



The daytime average $P(O_3)_{net}\_Mod$ values from Case $D_4$ changed by only 1.1 % compared to those derived from Case $D_3$, indicating that the potential contribution of OVOCs to compensating $P(O_3)_{net}\_Missing$ is greater than that of chlorine-containing VOCs. However, in modelling scenario Case $D_4$, the daytime average $P(O_3)_{net}\_Mod$ still showed a 22.2 % underestimation compared to the measured values.

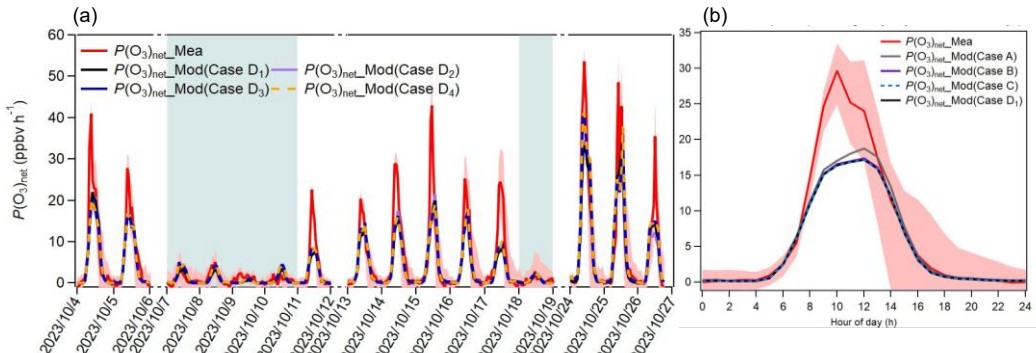

**Figure 3: The time series and diurnal variations of $P(O_3)_{net}\_Mea$ and $P(O_3)_{net}\_Mod$ (Case $D_1$–$D_4$) during the observation period. The shaded areas in (a) represent rainy days.**

The diurnal variations of $O_3$ production pathways in Case $D_4$ are shown in Fig. S12. Compared to Case $D_1$, the $RO_2+NO$ reaction rate in Case $D_4$ was higher by 0–2.1 ppbv h$^{-1}$ in the diurnal variations during the whole measurement period (excluded the rainy days). The $RO_2$ species with higher contributions to this pathway included $CH_3O_2$, $HO_2C_4O_2$, $HO_{13}C_4O_2$, $HOCH_2CH_2O_2$, $HO_3C_4O_2$, $CH_3COCH_2O_2$, and $COCCOH_2CO_2$. This indicates that the constraints on additional OVOCs in Case $D_4$ (such as aldehyde and ketone compounds with specific functional groups, e.g., carbonyl and hydroxyl) increased the intermediate $RO_2$ products, leading to a significant enhancement in the $RO_2+NO$ reaction rate. This suggests their large potential to contribute to $P(O_3)_{net}\_Missing$.

The modelling results of scenarios Case $D_1$–$D_4$ show that although constraining the measured VOC species in the box model mechanism can reduce $P(O_3)_{net}\_Missing$ to some extent, there is still a significant gap between the simulated and measured $P(O_3)_{net}$ values. Therefore, the potential contribution of unmeasured VOC species to compensating $P(O_3)_{net}\_Missing$ in the box model mechanism cannot be ignored. Yang et al. (2017) and Tan et al. (2019) conducted radical measurements at the Guangdong Atmospheric Supersite of China in autumn of 2014, revealing missing $k_{OH}$ contributions of approximately 32 % and 50 %, respectively. Yang et al. (2017) pointed out that the missing $k_{OH}$ contributions in the Heshan region may originate from OVOCs such as aldehydes, acids, and dicarbonyls. Tan et al. (2019) indicated that about 60 % of the $O_3$ produced in the Heshan region was contributed by unmeasured VOCs. We hypothesize that the remaining $P(O_3)_{net}\_Missing$ is caused by unknown VOCs that are not constrained in the box model. By quantifying the relationship between $k_{OH}$ and $P(O_3)_{net}$, the contribution of missing $k_{OH}$ ($k_{OH}\_Missing$) to $P(O_3)_{net}\_Missing$ can be assessed, and compensating for $k_{OH}\_Missing$ in the box model can help reduce $P(O_3)_{net}\_Missing$. Figure 4 shows the relationship between $k_{OH}$ and $P(O_3)_{net}\_Mod$ calculated under the Case $D_1$ scenario, which can be expressed as:

$$P(O_3)_{net}\_Missing = 3.4 \times k_{OH}\_Missing - 2.7 \tag{11}$$



where $P(O_3)_{net}$_Missing and $k_{OH\_}$ Missing in the equation represent the daytime averaged values for each day. Based on

this relationship, we calculated $k_{OH\_}$ Missing according to calculated $P(O_3)_{net}$_Missing for each day. This value was then used

to compensate for the unmeasured VOCs in the model (with a daytime $k_{OH}$ compensation range of 1.2–2.4 s⁻¹). Based on the

significant contribution of OVOCs to $P(O_3)_{net}$_Missing mentioned earlier, we designed three modelling scenarios to

compensate for $k_{OH\_}$ Missing, with the specific multiples varying each day: (1) Case $E_1$: by expanding the constrained overall

VOCs concentrations in Case $D_1$, the daily TVOC concentration was increased by 1.1 to 1.7 times; (2) Case $E_2$: according to

$k_{OH}$ ratio of NMHC to OVOCs in the constrained VOCs of Case $D_1$, the concentrations of ethylene (a representative NMHC

species) and formaldehyde (OVOCs indicator) were expanded separately. The ethylene concentration was increased by 5.9 to

85.6 times, and the formaldehyde concentration was increased by 1.4 to 2.0 times; (3) Case $E_3$: by expanding only the

formaldehyde concentration to compensate for $k_{OH\_}$ Missing, in this case, the daily formaldehyde concentration was increased

by 1.8 to 9.2 times, to verify the role of OVOCs in compensating for $P(O_3)_{net}$ Missing.



**Figure 4: The relationship between $k_{OH}$ and $P(O_3)_{net}$_Mod calculated under the Case $D_1$ scenario (using the daily daytime average values during the observation period).**

In Case $E_1$, where the overall TVOC concentration was increased to compensate for $k_{OH}$_Missing without distinguishing

VOCs categories, the compensation effect was limited due to the dilution effect of low-reactivity VOCs, resulting in a reduction

of the daytime average $P(O_3)_{net}$_Missing proportion from 26.3 % (calculated as $P(O_3)_{net}$_Missing/$P(O_3)_{net}$_Mea) to 10.3 %. In

Case $E_2$, where the concentrations of ethylene and formaldehyde were expanded, the daytime average $P(O_3)_{net}$_Missing

proportion reduced from 26.3 % to 17.2 %. This proportion is higher than that obtained from Case E1, which may be due to

the relatively low reactivity of ethylene limited the overall compensation effect. In contrast, Case $E_3$ compensated for $k_{OH\_}$

Missing solely by expanding the formaldehyde concentration. Since formaldehyde, as a representative high-reactivity OVOC

species, contributes more directly and significantly to $O_3$ generation through photochemical pathways (Mousavinezhad et al.,

2021), it achieved the best compensation effect, reducing the daytime average of $P(O_3)_{net}$_Missing from 26.3 % to 5.1 %.

However, $P(O_3)_{net}$_Missing during the peak period of diurnal variation remained at 9.0 ppbv h⁻¹. This result confirms the critical

role of high-reactivity OVOCs (especially those with the same photochemical reaction characteristics as formaldehyde) in

compensating for $P(O_3)_{net}$_Missing. Further, it suggests the potential presence of other unmeasured high-reactivity VOC species

in the ambient atmosphere. Constraining these species could help further improve the model's simulation accuracy (Lyu et al.,





2024; Wang et al., 2024b). Overall, the degree of compensation for $P(O_3)_{net}\_$Missing follows the order Case $E_3$>Case $E_1$>Case $E_2$, which may be related to the reactivity of the selected VOCs.

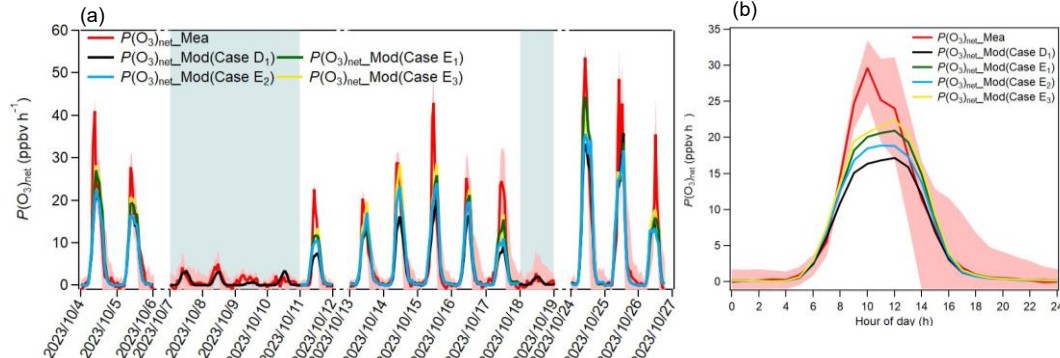

**Figure 5: (a) Time series and (b) diurnal variations of $P(O_3)_{net}\_$Mea and $P(O_3)_{net}\_$Mod (Case $D_1$–$E_3$) during the observation period; (b) Diurnal variations excluding rainy days. The shaded areas in (a) represent rainy days.**

## 4. OFS assessment based on measurements and simulations

This study systematically estimated OFS during the observation period (4–5, 11, 13–17, and 24–26 October 2023) using measured OFS (more details see Sect. 2.1) and modelled OFS (more details are shown in Sect. 2.6). Fig. S14 presents the diurnal variations of the directly measured IR index from OFS experiments, as well as the absolute $P(O_3)_{net}$ sensitivity of $NO_X$ and VOCs calculated based on the box model (Case $D_1$, Eq. (9)). We see from Fig. S14 that both measured OFS and modelled OFS captured the same diurnal OFS trend: an early morning (8:00-12:00) VOCs-limited/transition regime shifting to a $NO_X$-limited regime around midday (13:00), followed by a return to VOCs-limited/transition conditions in the afternoon (14:00-18:00). This midday transition to $NO_X$-limited conditions is chemically reasonable, where intensified $NO_2$ photolysis boosts $O_X$ production while concurrently diminished NO titration and declining VOCs emissions collectively favor $NO_X$-sensitive chemistry during peak sunlight hours (Wang et al., 2023). The overall OFS classification (mainly VOCs-limited and transition regimes) aligns with previous studies in Guangdong in autumn (Song et al., 2022; Chen et al., 2020b; Wu et al., 2020; Jing et al., 2024). However, the OFS assessment results from measured and modelling methods showed only 60 % agreement in hourly OFS variations (see Fig. S14).

In order to gain a deeper understanding of the similarities and differences between the direct measurement and the model simulation methods in diagnosing OFS, we divided the daytime observation period into three characteristic phases: the $P(O_3)_{net}$ rising phase (8:00-9:00), the $P(O_3)_{net}$ stable phase (10:00-12:00), and the $P(O_3)_{net}$ declining phase (13:00-17:00). Fig. 6 (a), (c), and (e) present the diurnal cumulative average results of IR derived from direct measurements of $\Delta P(O_3)_{net}^{+NO}$ and $\Delta P(O_3)_{net}^{+VOCs}$ using the NPOPR detection system for each phase. Fig. 6 (b), (d), and (f) show the diurnal cumulative average results of the absolute $P(O_3)_{net}$ sensitivity calculated from the box model (Case $D_1$) for each phase. We found that during $P(O_3)_{net}$ rising phase, both the direct measurement method and the model simulation method identified the OFS as being in the transition regime or VOCs-limited regime. However, the agreement between these two methods was only 63.6 %. This low consistency may be related to the rapid changes in precursor concentrations in the morning: the concentrations of VOCs and





$NO_X$ change quickly during this period, particularly due to traffic emissions and industrial activities, and these rapid variations in precursor concentrations make it challenging for the model to accurately capture the instantaneous reaction dynamics (Cao et al., 2021). During the $P(O_3)_{net}$ stable phase, the consistency between these two methods improved significantly, reaching an agreement of 72.7 %, with the OFS mostly located in the transition regime. This higher consistency occurred during periods of higher solar radiation intensity, when photochemical reactions were more stable. This stability resulted in improved model simulation accuracy. In the $P(O_3)_{net}$ declining phase, the OFS assessment results between these two methods researching an agreement of 72.7 %, both methods predominantly identified the OFS as either the transition regime or $NO_X$-limited regime. This relatively high agreement may be attributed to the reduced intensity of solar radiation and the decreased complexity of photochemical reactions in the afternoon, as Chen et al. (2025) showed that lower solar radiation simplifies reaction pathways, thereby enhancing model simulation accuracy.

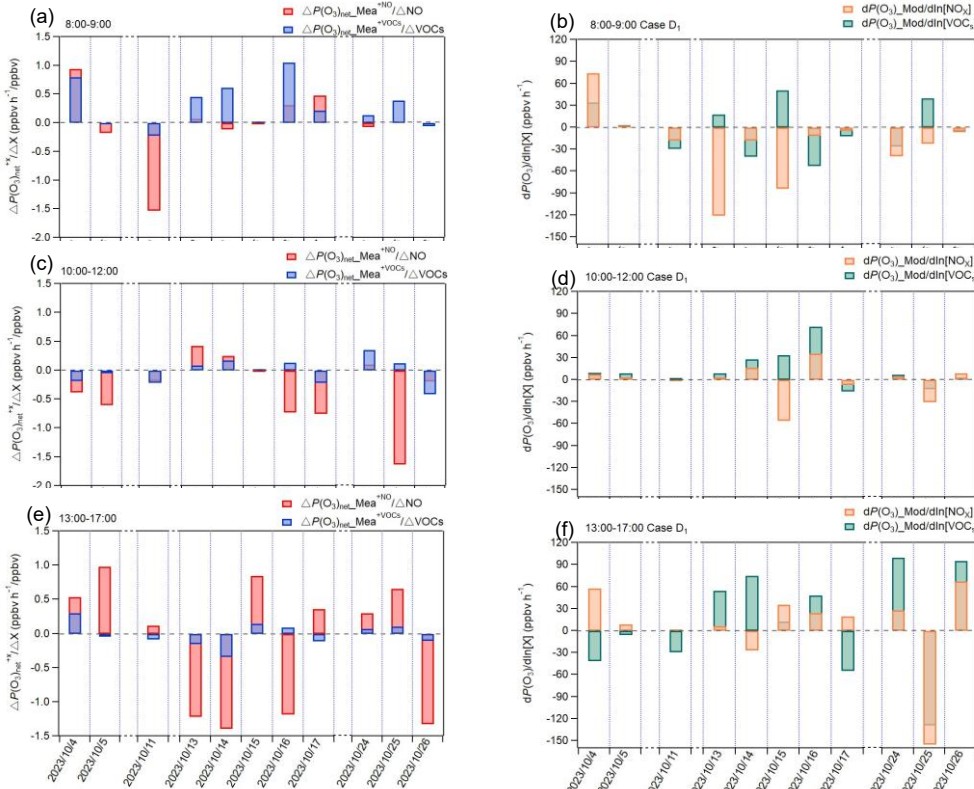

**Figure 6: Average values of IR derived from the direct measurement data using the NPOPR detection system (e.g., $\Delta P(O_3)_{net}^{+NO}$ and $\Delta P(O_3)_{net}^{+VOCs}$ ) and absolute $P(O_3)_{net}$ sensitivity from the box model during (a)–(b) $P(O_3)_{net}$ rising phase (8:00-9:00) ; (c)–(d) $P(O_3)_{net}$ stable phase (10:00-12:00) (e)–(f) $P(O_3)_{net}$ declining phase (13:00-17:00).**

The absolute $P(O_3)_{net}$ sensitivity for scenarios Case $E_1$–Case $E_3$ are shown in Fig. S15. The agreement between these scenarios and the direct measurement results changes across different periods, with consistency levels of 54.5–63.6 %, 45.5–





72.7 %, and 63.6–72.7 % during $P(O_3)_{net}$ rising phase, $P(O_3)_{net}$ stable phase, and $P(O_3)_{net}$ declining phase, respectively. In cases where $P(O_3)_{net}$_Missing was reduced (Case $E_1$–Case $E_3$), the OFS sometimes shifted to $NO_X$-limited conditions during certain periods, such as in Case $E_2$ during the $P(O_3)_{net}$ rising phase and Case E3 during the $P(O_3)_{net}$ stable phase on October 4, 2023. This contradictory phenomenon may be related to the model's incomplete representation of unknown high-reactivity VOCs

chemical mechanisms (e.g., aldehyde and ketone). Additionally, previous studies have pointed out that the diagnostic method based on the box model tends to overestimate the sensitivity to VOCs in certain regions of China due to neglecting the reactivity of unidentified VOCs in anthropogenic emissions (Xu et al., 2022; Lu et al., 2010). To more accurately simulate $O_3$ formation and precursor sensitivity, Xu et al. (2022) incorporated formaldehyde as input data in the box model, and found that this improvement significantly reduced the model's bias in diagnosing OFS, particularly in misjudging the VOCs-limited regime.

It is noteworthy that there are differences in the precursor sensitivity response mechanisms between the absolute $P(O_3)_{net}$ sensitivity assessment method based on the box model and the direct measurement method. For example, during the $P(O_3)_{net}$ stable phase (10:00-12:00 period) on 4–5 October, although both methods identified the OFS as being in the transition regime, the direct measurement showed that an increase in precursor concentrations suppressed $P(O_3)_{net}$, while the model simulations indicated that a reduction in precursor concentrations led to a decrease in $P(O_3)_{net}$. However, these findings only explain

regional differences in sensitivity determinations, and the underlying reasons for the differing precursor sensitivity response mechanisms between the two methods may require further investigation.

## 5 Conclusions

Understanding ozone ($O_3$) production mechanisms is critical for accurate $O_3$ pollution assessment and control, as photochemical production directly effects $O_3$ concentration levels. Due to the absence of certain mechanisms in conventional

models, particularly the kinetics from missing reactive volatile organic compounds (VOCs) species, the reliability of net photochemical $O_3$ production rates ($P(O_3)_{net}$) and $O_3$ formation sensitivity (OFS) evaluation is compromised. To address this issue, we employed the developed $P(O_3)_{net}$ (NPOPR) detection system based on the dual-reaction chamber technique to measure the $P(O_3)_{ne}$ and OFS. The system was applied in field observations at the Guangdong Atmospheric Supersite of China in Heshan, Pearl River Delta during the autumn of 2023. By combining the NPOPR detection system and the box model, a

systematic investigation of $P(O_3)_{net}$ and OFS was carried out.

During the observation period (4–26 October 2023), a total of 6 $O_3$ pollution days were recorded, with the maximum $O_3$ mixing ratio reaching 136.5 ppbv. The $P(O_3)_{net}$ levels on $O_3$ pollution days were significantly higher than those on normal days, indicating that high temperatures, low humidity, strong solar radiation, and stagnant weather conditions favor the $O_3$ pollution formation (see Sect. 3.1). The observational results show that oxygenated volatile organic compounds (OVOCs) and aromatic

hydrocarbons contributing 51.6 % and 32.9 % to OFP, respectively, which are the primary contributors to $O_3$ formation. These species are mainly emitted by anthropogenic emissions.

Systematic underestimation in the modelled $P(O_3)_{net}$ ($P(O_3)_{net}$_Mod) was found when compared to the measured ($P(O_3)_{net}$_Mea), which is defined as $P(O_3)_{net}$_Missing. In the Case A modelling, only the simplified chemical reaction



mechanism from the MCM was considered, and we see the daytime average $P(O_3)_{net}\_$Missing was around 2.6 ppbv h$^{-1}$ (20.3 %),

After gradually incorporating mechanisms such as HO$_2$ uptake by ambient aerosols, dry deposition, N$_2$O$_5$ uptake, and ClNO$_2$ photolysis (Case D$_1$), these gaps didn't get filled. The daytime average $P(O_3)_{net}\_$Missing was 3.4 ppbv h$^{-1}$ (26.3 % underestimation), with the highest daily median (50th percentile) attaining 17.5 ppbv h$^{-1}$on O$_3$ pollution days. After adding constraints for VOC species such as acetaldehyde, acrolein, acetone, and butanone compared to Case D$_1$ (defined as Case D$_2$), the $P(O_3)_{net}\_$Mod values increased by only 0.5 % compared to the modelling scenario Case D$_1$. However, after further

constraining all measurable OVOC species (Case D$_3$), the $P(O_3)_{net}\_$Mod values increased by 4.4 % compared to those obtained from Case D$_2$, with a notable improvement of 10.2 % (approximately 1.3 ppbv h$^{-1}$) during the $P(O_3)_{net}$ rising phase (8:00-9:00). This indicates that OVOCs play a particularly significant role in O$_3$ formation during the morning. Additionally, after adding chlorine-containing VOCs (in Case D$_4$), the $P(O_3)_{net}\_$Mod values increased by only 1.1 % compared to those obtained from Case D$_3$, further confirming the dominant role of OVOCs in compensating for $P(O_3)_{net}\_$Missing. These results also demonstrate

that by adding some missing mechanisms and measured VOCs species cannot fully eliminate simulation biases.

   Further analysis revealed that unmeasured VOCs, especially OVOCs, are the primary cause of these deviations. To quantify this effect, we developed a compensation approach based on the observed relationship between daytime averages $P(O_3)_{net}\_$Missing and $k_{OH\_}$Missing. This approach hypothesized that upscaling measured VOCs could compensate for $k_{OH\_}$Missing attributed to unmeasured species, thereby reducing $P(O_3)_{net}\_$Missing. Building upon Case D$_1$, designed three

modelling scenarios (Case E$_1$: expanded TVOC; Case E$_2$: expanded ethylene and formaldehyde; Case E$_3$: expanded formaldehyde) to compensate for $P(O_3)_{net}\_$Missing. Among these modelling scenarios, the daytime average $P(O_3)_{net}\_$Missing of Case E$_1$, Case E$_2$, and Case E$_3$ reduced to 10.3 %, 17.2 %, and 5.1 %, respectively. Notably, Case E$_3$ achieved the most significant reduction by solely increasing formaldehyde concentrations, validating the critical role of highly reactive OVOCs (particularly formaldehyde) in compensating for $P(O_3)_{net}\_$Missing. This suggests that there may be other unmeasured highly

reactive VOC species in the ambient atmosphere, and constraining these species could further improve the model's simulation accuracy.

   Additionally, the sensitivity assessment results derived from the different measured and modelled OFS approaches were compared: (1) in direct measurement using the NPOPR detection system, NO or VOCs were added to quantify changes in $P(O_3)_{net}$, with OFS determined through the incremental reactivity (IR) index (IR=$\Delta P(O_3)_{net}^{+x}$/$\Delta$S(x), where X = NOx or VOCs

and $\Delta$S(x) represents the added concentration); (2)in model simulations, where the box model calculated $P(O_3)_{net}$ and derived absolute $P(O_3)_{net}$ sensitivity ($\delta P(O_3)_{net}$/$\delta$[X], where X = NOx or VOCs). Meanwhile we found that the agreement of OFS assessment results between the direct measurements and the model results was lower in the $P(O_3)_{net}$ rising phase (8:00-9:00, 63.6 %) than those in the $P(O_3)_{net}$ stable phase (10:00-12:00, 72.7 %) and $P(O_3)_{net}$ declining phase (13:00-17:00, 72.7 %). This again highlights the importance of highly reactive OVOCs in improving the accuracy of OFS assessment. These results indicate

that reducing $P(O_3)_{net}\_$Missing can enhance the accuracy of OFS assessment to some extent, but fully eliminating the discrepancies still requires further constraints on unmeasured VOC species and further research.

   In conclusion, improving the model's accuracy requires further expansion of the measurement of VOC species,



particularly OVOCs, and the incorporation of relevant chemical mechanisms into the model. In future studies, continuing field observations based on direct measurement of $P(O_3)_{net}$ and accumulating more data will contribute to a better understanding of

$O_3$ pollution formation mechanisms and make effective $O_3$ pollution control strategies.

**Date availability.** The datasets supporting this research are included in this manuscript and its supplementary information files. The data for this study are also publicly available at https://doi.org/10.5281/zenodo.15052519. Meteorological data were sourced from the European Centre for Medium-Range Weather Forecasts (ECMWF, https://www.ecmwf.int/). Box model simulations were conducted using the AtChem2 model (https://atchem.leeds.ac.uk/webapp/) with the Master Chemical

Mechanism (MCM v3.3.1, https://mcm.york.ac.uk/MCM). Figures in this study were created using Igor Pro 6.7. Additional data or materials related to this study can be made available upon reasonable request to the corresponding author (junzhou@jnu.edu.cn), subject to restrictions on data resources.

**Author contributions.** Author contributions. JZ and MS designed this study. BZ and JZ wrote the manuscript with

contributions from all co-authors. BZ, JZ, TZ, DC, BJ, YZ, J. Li, MD, MX, JHJ, and J. Luo collected and analyzed the data. All authors reviewed and revised the manuscript.

**Competing interests.** The contact author has declared that none of the authors has any competing interests.

**Acknowledgments.** Many thanks to the Guangdong Ecological and Environmental Monitoring Center.

**Financial support.** This work was funded by the National Natural Science Foundation of China (No. 42305096), the Special

Support Plan for High-Level Talents of Guangdong Province (No. 2023JC07L057), the Natural Science Foundation of Guangdong Province (No. 2024A1515011494), the National Key Research and Development Program of China (No. 2023YFC3706204), and the Guangdong Provincial Basic and Applied Basic Research Fund (the Youth Doctoral "Launch" Project) for the Year 2025 (No. SL2024A04J00396). Jianhui Jiang was supported by the National Natural Science Foundation of China (No. 42207122).

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
