# Peer review of "Why observed and modelled ozone production rates and sensitivities differ, a case study at rural site in CHINA"

_EGUsphere, 2025_

## Author Comment (AC1)

We thank the referees for their insightful questions and comments, which helped to improve the quality of this paper. Our answers to all the concerns are listed in the following in red, after the reviewer's comments, which are in black. The changed parts in the modified manuscript are marked in yellow.

**Anonymous Referee #1**

This manuscript presents a comprehensive investigation of net ozone production rate  $(P(O_3)_{net})$  and ozone formation sensitivity (OFS) through the integration of in situ field observations using a novel dual-channel reaction chamber system (NPOPR) and detailed box model simulations based on MCM v3.3.1. The study is of high relevance and scientific value, particularly in addressing long-standing issues of underestimation in modeled ozone production. The work also has practical implications for improving model-based OFS diagnosis and VOC pollution control strategies. However, several major issues must be addressed in the manuscript.

In your study, observed OVOC concentrations are used to constrain the box model. However, many OVOCs (e.g., formaldehyde, acetaldehyde, ketones) are not only emitted directly but also formed via secondary photochemical reactions from VOC precursors. Directly constraining their concentrations may mask deficiencies in the model's chemical mechanism and artificially suppress diagnostic signals of missing secondary formation pathways.

We fully appreciate your concern and recognize the potential issues that may arise from directly constraining OVOC concentrations. To investigate whether the model can reproduce secondary formation on its own, we added a modelling scenario without OVOC constraints based on Case D1 and output key OVOC species (as shown in Fig. S19). Results show that, Without additional constraints, the model overestimates some OVOCs (e.g., HCHO and CH3CHO) yet adequately reproduces their secondaryformation pathways; in contrast, the observed diurnal cycle of CH3COCH3 shows no evident signature of secondary production. These results demonstrate that directly constraining the OVOC concentrations may mask deficiencies in the model's chemcial mechanism and artificially suppress diagnostic signals of missing secondary formation pathways. However, refraining from any constraint would also falsely amplify the primary-source signal, especially those lacking clear secondary-generation signatures. Applying a constraint can better capture the influence of primary OVOCs. Furthermore, our analysis indicates that the P(O3)net missing is not likely caused by unaccounted secondary production (see Sect. 3.3). Until such mechanistic gaps are resolved, observational nudging of OVOCs remains a pragmatic compromise: it preserves

concentration accuracy while curbing spurious chemical feedbacks. We have added such kind of discussion in lines 401-406 of the modified mansucript:

"The negligible (or even negative) change in  $P(O_3)_{net}$  Mod when OVOCs are constrained in Cases D1-D4 may arise because the OVOC constraint masks deficiencies in the model's chemcial mechanism and artificially suppresses diagnostic signals of missing secondary formation pathways. Until the underlying chemical mechanisms are improved, observational nudging of OVOCs offers a practical compromise—it helps maintain concentration accuracy while limiting unrealistic chemical feedbacks (more details can be found in Supplementary Materials S5)."

And S5 in the Supplementary Materials:

**"S5. Impacts of OVOCs constraints in the model**

To explore the impact of OVOCs constraint in the model, we further added a modelling scenario without OVOC constraints based on Case D1 and output key OVOC species (see Fig. S19). From Fig. S19, the model tends to overestimate some OVOC concentrations (i.e., HCHO, CH3CHO), and their secondary-formation pathways are adequately captured, while the observed diurnal variation of CH3COCH3 does not exhibit clear secondary formation characteristics. These results show that directly constraining OVOC concentrations can fill the concentration gap in the model to match observed OVOC levels, but may mask deficiencies in the model's chemical mechanism and artificially suppress diagnostic signals of missing secondary formation pathways (i.e., the RO2-to-OVOC reaction pathways). This will lead to the underestimation of the entire HOx-cycle oxidation rate, lowers the budgets of OH, O3, and  $NO_3$ , and subsequently the  $P(O_3)_{net}$  Mod. However, without any constraint, the model may overestimate the contribution from primary sources. Furthermore, our analysis indicates that the  $P(O_3)_{net}$  missing is not likely caused by unaccounted secondary production (see Sect. 3.3). Until such mechanistic gaps are resolved, observational nudging of OVOCs remains a pragmatic compromise: it preserves concentration accuracy while curbing spurious chemical feedbacks.

Figure S19: Comparison of measured OVOCs with modeled values from a no-constraint OVOC scenario based on Case D1."

Why were NO and NOx changed between the two methods to diagnose  $O_3$  sensitivity, respectively?

We measured ozone (O3) formation sensitivity (IR index) directly using the NPOPR instrument, with NO as the NOx indicator. This approach aligns with previous studies(Sklaveniti et al., 2018; Morino et al., 2023), which also used NO in their  $P(O_3)_{net}$  measurement systems. By adding NO instead of NOx, the CAPS-NO2 instrument could eliminate NO2 interference and directly quantify NO's contribution to O3 production. When O3 formation sensitivity is in the NOx-limited regime, increasing NO enhances  $P(O_3)_{net}$  because HO2/RO2 reacts more efficiently with NO to produce NO2. Conversely, in the VOCs-limited regime, adding NO may reduce  $P(O_3)_{net}$  because NO consumes OH to form HONO, weakening VOC oxidation chain reactions.

Our model's  $O_3$  formation sensitivity (OFS) analysis is based on the absolute  $P(O_3)_{net}$  sensitivity calculation, which directly evaluates how changes in  $NO_X$  (NO +  $NO_2$ ) affect  $P(O_3)_{net}$  while considering both the NO titration effect and the role of  $NO_2$  photolysis in  $O_3$  production.

The manuscript attributes the model - measurement discrepancy (P(O3)net\_Missing) entirely to missing reactive VOCs or underrepresented chemical pathways. However, box models by design do not account for horizontal or vertical

transport, which may play a significant role in shaping the measured ozone production rate—especially during periods with strong advection or mixing layer evolution, such as early morning or late afternoon. You should clarify why transport processes are neglected and whether their influence is truly negligible.

We acknowledge that transport processes may influence  $O_3$  distribution under specific conditions. However, this study primarily focuses on local photochemical  $O_3$  production. By employing the NPOPR detection system with an extremely short residence time (0.15 h), we aimed to capture instantaneous in-situ photochemical reactions rather than  $O_3$  accumulation effects over time and space. The modeling incorporated real-time meteorological conditions and pollutant concentration data. On such short timescales, the impacts of vertical mixing and horizontal advection become relatively minor. Moreover, this study compares measured and simulated net  $O_3$  production rates  $(P(O_3)_{\text{net}})$  rather than  $O_3$  concentrations themselves. Our previous study also demonstrated that  $P(O_3)_{\text{net}}$  more directly reflects the photochemical  $O_3$  formation potential from local precursors and is less affected by transport processes compared to  $O_3$  concentrations (Zhou et al., 2024b).

We added "The time resolution of the  $P(O_3)_{net}$  measurement is 4 min. Our previous study demonstrated that  $P(O_3)_{net}$  more directly reflects the photochemical  $O_3$  formation potential from local precursors and is less affected by transport processes compared to  $O_3$  concentrations (Zhou et al., 2024b)." in line 129-131 of the modified manuscript.

Although the manuscript includes substantial observation – model comparisons and compensatory mechanisms for missing reactivity, the concluding section does not clearly state what is new in this work compared to existing studies, please clearly emphasize the innovation points and boundaries of this study in the conclusion section and explain its promoting role in the research of the formation mechanism of ozone pollution.

We appreciate the reviewer's valuable suggestion to clarify the novel contributions of our study. The main innovation of this work is that we have successfully applied the NPOPR detection system to directly measure OFS in field observations. This approach enables us to quantitatively assess the impact of  $P(O_3)_{net}$  simulation deficits on OFS determination. Previous studies in this field predominantly relied on model simulations without sufficient observational validation. While our earlier research (Zhou et al., 2024a) qualitatively analyzed the effects of different VOC species on  $P(O_3)_{net}$  through model-observation comparisons, quantitative analysis was still lacking. This study makes significant advancements by employing direct measurement methods to quantitatively investigate the contributions of different VOC species to  $P(O_3)_{net}$

simulation deficits. Our findings address the critical scientific issue of model underestimation of  $P(O_3)_{net}$  and quantify its impact on OFS determination. These results provide both a robust database and theoretical foundation for improving the accuracy of model-based OFS assessment and developing more effective  $O_3$  pollution control strategies.

We have changed the sentence "In conclusion, improving the model's accuracy requires further expansion of the measurement of VOC species, particularly OVOCs, and the incorporation of relevant chemical mechanisms into the model. In future studies, continuing field observations based on direct measurement of P(O3)net and accumulating more data will contribute to a better understanding of O3 pollution formation mechanisms and make effective O3 pollution control strategies." to "In conclusion, we quantitatively assessed the  $P(O_3)_{net}$  simulation deficits and their impact on OFS diagnosis by comparing the measured and modelled P(O3)net, and found that the unmeasured VOCs —rather than the secondary atmospheric formation are the primary causative factor of  $P(O_3)_{net}$  Missing. Furthermore, both direct measurements and model results reveal a diurnal OFS shift dominated by the morning regime; transition and VOC-limited conditions prevailed, so prioritizing VOCs while co-controlling NOX is the most effective approach to O3 pollution control in PRD region. Our results also demonstrate that the persistent model biases risk under-estimating the local photochemical formation contribution to O3 pollution, thereby has weakening its perceived impact relative to physical transportation. Future studies should expanded VOCs measurements and combine direct P(O3)net observations with regional transport model to separate local production from up-wind advection." in lines 610-617 in the modified manuscript.

Lines 29-31: You mentioned "the only approach to fill the gag was to add unmeasured VOCs" appears too strong. It implies that no other explanations or methods could be relevant, which may not be justified. Consider softening this to reflect that unmeasured OVOCs were the most effective compensating factor in this study, rather than the only one possible.

Thanks for your suggestion. We have changed the sentence "The only approach to fill the gap between observation and computation was to add possible unmeasured reactive VOCs, ... " to "The results in this study reflected that unmeasured oxygenated VOCs (OVOCs) were the most effective compensating factor for the discrepancies between observed and computed  $P(O_3)_{net}$  and OFS, ..." in lines 29-30 in the modified manuscript.

```
Line 43: "precursor" should be "precursors".
```

We changed "precursor" to "precursors" in line 43 of the modified manuscript.

```
Line 46: "equation" should be "equations".
```

We changed "equation" to "equations" in line 50 of the modified manuscript.

Lines 205-206: It is unclear what magnitude of precursor perturbation was applied, please provide a more explicit description of the model configuration used for the sensitivity analysis.

We thank the reviewer for this valuable comment. Regarding the calculation method for absolute  $P(O_3)_{\text{net}}$  sensitivity in our model, we employed the analytical approach proposed by Sakamoto et al. (2019). This method does not require artificial perturbation of precursor concentrations in the simulations. Instead, it is defined as the change in  $P(O_3)_{\text{net}}$  values caused by a percentage change in either [NOX] or [VOCs] concentrations. To better clarify this methodology, we have revised the manuscript to explicitly state: "In this study, the analysis of absolute  $P(O_3)_{\text{net}}$  sensitivity was conducted using the box model through an analytical calculation approach that does not involve artificial perturbation of precursor concentrations." in line 233 of the modified manuscript.

Lines 278-281: The confidence interval of 68.3% is relatively conservative, please provide additional analysis.

Regarding the confidence interval selection, we initially chose the 68.3% interval as it corresponds to the  $\pm 1\sigma$  range in Gaussian distribution, which represents a standard statistical measure. However, as suggested by the reviewer, we have conducted additional sensitivity analysis using a 90% confidence interval. The results show similar trends in both cases, confirming the robustness of our findings.

Figure 1: Correlation between measured  $P(O_3)_{net}$  ( $P(O_3)_{net}$  Mea) and (a) total OH reactivity ( $k_{OH}$ ) and (b)  $O_3$  Formation Potential (OFP). The shaded area in the figure represents the confidence interval (90 %) of the fitting line between  $P(O_3)_{net}$  and  $k_{OH}$ , and between  $P(O_3)_{net}$  and OFP."

We have changed the sentence "The shaded area in the figure represents the confidence interval (68.3%) of the fitting line between  $P(O_3)_{net}$  and  $k_{OH}$ , and between  $P(O_3)_{net}$  and OFP." to "The shaded area in the figure represents the confidence interval (90 %) of the fitting line between  $P(O_3)_{net}$  and  $k_{OH}$ , and between  $P(O_3)_{net}$  and OFP." in line 309 in the modified manuscript.

Line 306: The reported p-value (P < 0.5) does not indicate statistical significance, and the analysis doesn't hold.

Thank you for catching this typographical error. We have carefully re-examined the statistical analysis comparing  $P(O_3)_{\text{net}}$ \_Missing between  $O_3$  pollution days and normal days. The corrected results show a significant difference with p = 0.03 (< 0.05). We have changed the sentence The median  $P(O_3)_{\text{net}}$ \_Missing values on  $O_3$  pollution days were statistically higher than those on normal days (*t-test*, P<0.5), ... To The  $P(O_3)_{\text{net}}$ \_Missing values on  $O_3$  pollution days were statistically higher than those on normal days (*t-test*, p<0.05), ... in line 343 of the modified manuscript.

Lines 305-307: The statement that the mechanisms added in Case D1 "are not the main cause" of the bias may overstate the conclusion. The remaining discrepancy could still be partly due to uncertainties in those mechanisms, parameterization, site-specific variability and transportation etc. A more cautious wording would improve clarity and avoid giving a false sense of certainty.

Thanks for your suggestion. We have changed the sentence "The median  $P(O_3)_{\text{net}}$  Missing values on  $O_3$  pollution days were statistically higher than those on normal days (t-test, P<0.5), indicating that the supplementary mechanisms explored in the model, as mentioned above, are not the main cause of the  $P(O_3)_{\text{net}}$  Missing." to "The  $P(O_3)_{\text{net}}$  Missing values on  $O_3$  pollution days were statistically higher than those on normal days (t-test, p<0.05), suggesting that while the supplementary mechanisms explored in the model may contribute to some extent, they are unlikely to be the dominant cause of the  $P(O_3)_{\text{net}}$  Missing." in lines 343-344 in the modified manuscript.

Lines 328-329: The statement that " $P(O_3)_{net}$ \_Missing increases significantly at higher  $O_3$  precursor concentrations" (based on  $r^2 = 0.4$ -0.5) may overstate the strength of the relationship. A moderate correlation should not be equated with a strong or significant increase unless supported by statistical testing.

We appreciate this constructive comment regarding the interpretation of our correlation results. Our additional analyses, including both Pearson correlation and t-tests, and found that r=0.24 and 0.20 for TVOCs and NOx, respectively. And the correlations are very weak, with t=1.6 and 1.3 for VOCs and NOX (t-tests), respectively. These t-test values for correlation significants are lower than the critical value of t=2.0, confirming

that  $P(O_3)_{net}$  Missing shows weak relationships with VOCs and NOX precursor concentrations.

We have changed the sentence "Under  $O_3$  pollution days,  $P(O_3)_{\text{net}}$  Missing showed a positive correlation with VOCs and NOX, with  $r^2$  values of 0.4 and 0.5, respectively, indicating that  $P(O_3)_{\text{net}}$  Missing increases significantly at higher  $O_3$  precursor concentrations. This phenomenon is consistent with previous studies (Whalley et al., 2021; Ren et al., 2013; Zhou et al., 2024a)." to "On  $O_3$  pollution days,  $P(O_3)_{\text{net}}$  Missing exhibited a moderate positive correlation with VOCs ( $r^2$  =0.4, R=0.2, t=2.9) and NOX ( $r^2$ =0.5, R=0.2, t=3.8), confirming that the  $P(O_3)_{\text{net}}$  Missing is larger at higher precursor concentrations/mixing ratios (both t > critical 2.0, p < 0.05), consistent with earlier box-model studies (Whalley et al., 2021; Ren et al., 2013; Zhou et al., 2024a). A moderate positive correlation is also found with  $J_{O1D}$  on both  $O_3$  pollution days and normal days, with  $r^2$  values of 0.5 and 0.4, respectively. On normal days all correlations collapse ( $r^2$  < 0.2, p > 0.1), implying that the model deficit is not tied to the measured precursors under low-NOx conditions and may instead related to the missing mechanisms for unmeasured photolabile VOCs." in lines 368-375 in the modified manuscript.

Lines 389-391: You use an empirical relationship between kOH and  $P(O_3)$ net to get kOH\_Missing, and then adjust VOC concentrations to match this value. However, this method assumes a direct linear relationship without showing how real chemical reactions support this assumption. Please explain why this approach is reasonable, and whether it reflects actual atmospheric chemistry.

The method of estimating missing VOC concentrations through the empirical linear relationship between OH reactivity ( $k_{\rm OH}$ ) and  $P({\rm O_3})_{\rm net}$  is used in this study, fundamentally based on the OH-driven nature of O3 production. The scientific basis lies in the fact that  $P({\rm O_3})_{\rm net}$  is closely related to the production rate of ROX radicals ( $P({\rm RO_X})$ ), which are primarily formed through the reaction of OH with VOCs. Since  $P({\rm RO_X})$  is directly influenced by the OH reactivity ( $k_{\rm OH}$ ),  $P({\rm O_3})_{\rm net}$  is consequently correlated with  $k_{\rm OH}$ . Furthermore, previous study have shown that  $P({\rm O_3})_{\rm net}$  exhibits a linear relationship with both  $P({\rm HO_X})$  and  $k_{\rm OH}$  when O3 formation is located in VOCs-limited regime (Baier et al., 2017), and this approach reflects nearly actual atmospheric chemistry if  $P({\rm O_3})_{\rm net}$  missing is driven by VOCs reactivity missing (Wang et al., 2024).

To explore the influence of unconstrained secondary products to  $P(O_3)_{net}$  missing (which may influence the linear relationship between  $P(O_3)_{net}$  missing and  $k_{OH}$ ), we checked the dependence of  $P(O_3)_{net}$  missing on the ethylbenzene/m,p-xylene ratio. As the m,p-xylene has a larger reaction rate constant  $(18.9 \times 10^{-12} \text{ cm}^3 \text{ molecule}^{-1} \text{ s}^{-1})$  than ethylbenzene  $(7.0 \times 10^{-12} \text{ cm}^3 \text{ molecule}^{-1} \text{ s}^{-1})$

molecule-1 s-1) when reacting with OH radicals, the ratio of ethylbenzene to m,p-xylene was used to characterize the degree of air mass aging (de Gouw et al., 2005; Yuan et al., 2013), a higher ratio of ethylbenzene to m,p-xylene corresponds to a higher degree of air mass aging. We see that the  $P(O_3)_{net}$  missing decreases with the increasing ratio of ethylbenzene to m,p-xylene (as added in Fig. S11f), which indicates that the  $P(O_3)_{net}$  missing was not caused by unconstrained secondary products.

To make the description clearer, we have changed the sentence "We hypothesize that the remaining  $P(O_3)_{net}$  Missing is caused by unknown VOCs that are not constrained in the box model. By quantifying the relationship between  $k_{\rm OH}$  and  $P({\rm O_3})_{\rm net}$ ..." to "We hypothesize that the remaining  $P(O_3)_{net}$  Missing is caused by unknown VOCs that are not constrained in the box model. The method of estimating missing VOC concentrations through the empirical linear relationship between OH reactivity ( $k_{OH}$ ) and  $P(O_3)_{net}$  is used in this study, the scientific basis lies in the fact that  $P(O_3)_{net}$  is closely related to the production rate of ROX radicals  $(P(RO_X))$ , which are primarily formed through the reaction of OH with VOCs. Since  $P(RO_X)$  is directly influenced by the OH reactivity  $(k_{\text{OH}})$ ,  $P(O_3)_{\text{net}}$  is consequently correlated with  $k_{\text{OH}}$ . Previous study have shown that  $P(O_3)_{net}$  exhibits a linear relationship with both P(HOx) and  $k_{OH}$  when O3 formation is located in VOCs-limited regime (Baier et al., 2017), and this approach reflects nearly actual atmospheric chemistry if  $P(O_3)_{net}$  missing is driven by VOCs reactivity missing (Wang et al., 2024b). Furthermore, we examined whether unconstrained secondary products affect  $P(O_3)_{\text{net}}$  missing —and thus the linear relationship between  $P(O_3)_{\text{net}}$  missing and  $k_{\text{OH}}$  by analysing its dependence on the ethylbenzene / m,p-xylene ratio. Because this ratio increases with the degree of air-mass aging (de Gouw et al., 2005; Yuan et al., 2013), the observed decrease in the  $P(O_3)_{net}$  missing with increasing ratio (Fig. S11f) indicates that the  $P(O_3)_{net}$ missing is not caused by unaccounted secondary production." In lines 428-438 in the modified manuscript.

The corresponding references are also added in the reference list:

"de Gouw, J., Middlebrook, A., Warneke, C., Goldan, P., Kuster, W., Roberts, J., Fehsenfeld, F., Worsnop, D., Canagaratna, M., and Pszenny, A.: Budget of organic carbon in a polluted atmosphere: Results from the New England Air Quality Study in 2002, Journal of Geophysical Research-Atmospheres, 110, D16305, https://doi.org/10.1029/2004JD005623, 2005.

Yuan, B., Hu, W. W., Shao, M., Wang, M., Chen, W. T., Lu, S. H., Zeng, L. M., and Hu, M.: VOC emissions, evolutions and contributions to SOA formation at a receptor site in eastern China, Atmospheric Chemistry and Physics, 13, 8815–8832, https://doi.org/10.5194/acp-13-8815-2013, 2013."

The corresponding figure which plots the dependence of  $P(O_3)_{net}$  missing on the ethylbenzene/m,p-Xylene ratio is added in Fig. S11f:

Figure S11: Correlations between  $P(O_3)_{net}$  missing and TVOCs, NOx,  $J_{O1D}$ , T, Ox (a–e), and the ethylbenzene/m,p-Xylene ratio (f, representing the air mass aging). Circles represent  $O_3$  pollution days, triangles represent normal days, and the shaded area indicates the 68.3 % confidence interval of the fitting line.

Lines 396-397: In Case E2, ethylene was amplified to 5.9-85.6 times the original concentration, far exceeding the limit emission levels in the conventional urban atmosphere. The lack of emission inventories or observational data support may cause the simulation results to deviate from reality. More discussion on the rationality of these magnifications is required.

The purpose of these scenarios is to demonstrate the potential impact of this type of VOCs on  $P(O_3)_{net}$ \_Missing. We have changed the sentence "...This value was then used to compensate for the unmeasured VOCs in the model (with a daytime  $k_{OH}$  compensation range of 1.2–2.4 s-1). Based on the significant contribution of OVOCs to  $P(O_3)_{net}$ \_Missing mentioned earlier, we designed three modelling scenarios to compensate for  $k_{OH}$ \_ Missing, with the specific multiples varying each day: (1) Case  $E_1$ : by expanding the constrained overall VOCs concentrations in

Case D1, the daily TVOC concentration was increased by 1.1 to 1.7 times; (2) Case E2: according to  $k_{\rm OH}$ ratio of NMHC to OVOCs in the constrained VOCs of Case D1, the concentrations of ethylene (a representative NMHC species) and formaldehyde (OVOCs indicator) were expanded separately. The ethylene concentration was increased by 5.9 to 85.6 times, and the formaldehyde concentration was increased by 1.4 to 2.0 times; (3) Case E3: by expanding only the formaldehyde concentration to compensate for  $k_{\text{OH}}$  Missing, in this case, the daily formaldehyde concentration was increased by 1.8 to 9.2 times, to verify the role of OVOCs in compensating for  $P(O_3)_{net}$  Missing" to "... This value was then used to compensate for the unmeasured VOCs in the model (with a daytime  $k_{\rm OH}$  compensation range of 1.2–2.4 s-1, approximately 27.6–45.1% of missing values). Based on the significant contribution of OVOCs to  $P(O_3)_{net}$  Missing mentioned earlier, we designed three modelling scenarios to compensate for  $k_{\rm OH}$  Missing, with the specific multiples varying each day. We note that these scenarios are idealized sensitivity tests to explore potential bounds of OVOCs' contribution to  $P(O_3)_{net}$  Missing compensation, rather than realistic emission assumptions. Specifically, we tested how much the P(O3)net Missing could be accounted for if the  $k_{OH}$  were attributed to different VOCs categories. The specific scenarios include: (1) Case E1: by expanding the constrained overall VOCs concentrations in Case D1 (daily mean compensation range for TVOCs: 0.5–2.8 µg m-3), the daily TVOC concentration was increased by 1.1 to 1.7 times; (2) Case E2: according to kOH ratio of NMHC to OVOCs in the constrained VOCs of Case D1, the concentrations of ethylene (a representative NMHC species) and formaldehyde (OVOCs indicator) were expanded separately. The ethylene concentration (daily mean compensation range for TVOCs: 0.5– 2.8 µg m-3) was increased by 5.9 to 85.6 times, and the formaldehyde concentration (daily mean compensation range for TVOCs: 0.0–0.5 μg m-3) was increased by 1.4 to 2.0 times; (3) Case E3: by expanding only the formaldehyde concentration to compensate for  $k_{\text{OH}}$  Missing, in this case, the daily formaldehyde concentration (daily mean compensation range for TVOCs: 0.6–1.4 µg m-3) was increased by 1.8 to 9.2 times, to verify the role of OVOCs in compensating for  $P(O_3)_{net}$  Missing." in lines 445-459 in the modified manuscript.

Line 485: It is not necessary to add legends to every subgraph in Fig. 6. Simplification can be considered.

Okay, we have removed the redundant legends in Fig. 6. See lines 517–520 in the modified manuscript:

Figure 6: Average values of IR derived from the direct measurement data using the NPOPR detection system (e.g.,  $\Delta P(\mathrm{O_3})_{\mathrm{net}}^{+\mathrm{NO}}$  and  $\Delta P(\mathrm{O_3})_{\mathrm{net}}^{+\mathrm{VOCs}}$ ) and absolute  $P(\mathrm{O_3})_{\mathrm{net}}$  sensitivity from the box model during (a)–(b)  $P(\mathrm{O_3})_{\mathrm{net}}$  rising phase (8:00-9:00); (c)–(d)  $P(\mathrm{O_3})_{\mathrm{net}}$  stable phase (10:00-12:00) (e)–(f)  $P(\mathrm{O_3})_{\mathrm{net}}$  declining phase (13:00-17:00)."

Lines 519-521: The conclusion repeatedly emphasizes the role of OVOCs in  $O_3$  formation and compensation, yet it assumes these are mainly anthropogenic in origin. As noted earlier, many OVOCs are also formed secondarily. It is recommended that you should distinguish between primary and secondary OVOC contributions or clearly state the limitation of their current attribution.

Thank you for your suggestion. We agree with the reviewer that it is not appripriate to say "These species are mainly emitted by anthropogenic emissions", as the OVOCs are also formed secondarily from both anthropogenic and natural emissions. Therefore, we have deleted this sentence and added the related discussion in lines 289-293 in the modified manuscript:

"As OVOCs arise from both direct (anthropogenic and natural) emissions and secondary atmospheric formation (Lyu et al., 2024; Yuan et al., 2012), precluding a direct quantification of their respective contributions to O3 formation. Nevertheless, our previous work showed that

anthropogenic primary VOCs correlate most closely with instantaneous  $P(O_3)_{net}$  on  $O_3$  pollution days, and urban anthropogenic OVOC emissions markedly enhance both oxidative capacity and  $O_3$  production (Qian et al., 2025; Wang et al., 2024b)."

**Appendix:**

(1) We detected an error in Fig. 3b, therefore, we changed the diurnal variations of  $P(O_3)_{\text{net}}$ Mea and  $P(O_3)_{\text{net}}$ Mod (Case A–D1) to diurnal variations of  $P(O_3)_{\text{net}}$ Mea and  $P(O_3)_{\text{net}}$ Mod (Case D1–E3) in Fig. 3b.

(2) The calculation method of the absolute  $P(O_3)_{net}$  is adapted from the logarithmic derivative approach of Sakamoto et al. (2019). Therefore, We have changed the sentence "We calculated the modelled OFS using the absolute  $P(O_3)_{net}$  sensitivity method from Sakamoto et al. (2019). It is defined as the change in  $P(O_3)_{net}$  induced by a percentage increase in  $O_3$  precursors. This method facilitates the quantitative assessment of how reductions in  $O_3$  precursors contribute to the overall reduction of  $P(O_3)_{net}$  over a period or within a region. The formula is as follows:

Absolute
$$P(O_3)_{net} = \frac{\delta P(O_3)}{\delta \ln[X]} = P(O_3) \frac{\delta P(O_3)}{\delta \ln[X]}$$
 (10)" to

"We calculated the modelled OFS using the absolute  $P(O_3)_{net}$  sensitivity adapted from the logarithmic derivative approach of Sakamoto et al. (2019). It is defined as the change in  $P(O_3)_{net}$  for the natural logarithm of  $O_3$  precursor concentrations. This method facilitates the quantitative assessment of how reductions in  $O_3$  precursors contribute to the overall reduction of  $P(O_3)_{net}$  over a period. The formula is as follows:

Absolute
$$P(O_3)_{\text{net}} = \frac{dP(O_3)_{\text{net}}}{d \ln[X]}$$
 (10)"

in lines 224-228 in the modified manuscript.

**References**

Baier, B. C., Brune, W. H., Miller, D. O., Blake, D., Long, R., Wisthaler, A., Cantrell, C., Fried, A., Heikes, B., and Brown, S.: Higher measured than modeled ozone production at increased NOX levels in the Colorado Front Range, Atmospheric Chemistry and Physics, 17, 11273-11292,http://doi.org/10.5194/acp-17-11273-2017, 2017. de Gouw, J. A., Warneke, C., Parrish, D. D., Holloway, J. S., Trainer, M., and Fehsenfeld, F. C.: Emission sources and ocean uptake of acetonitrile (CH3CN) in the atmosphere, Journal of Geophysical Research-Atmospheres, 108, 4329, https://doi.org/10.1029/2002JD002897, 2003.

de Gouw, J., Middlebrook, A., Warneke, C., Goldan, P., Kuster, W., Roberts, J., Fehsenfeld, F., Worsnop, D., Canagaratna, M., and Pszenny, A.: Budget of organic carbon in a polluted atmosphere: Results from the New England Air Quality Study in 2002, Journal of Geophysical Research-Atmospheres, 110, D16305, https://doi.org/10.1029/2004JD005623, 2005.

Lyu, X., Li, H., Lee, S.-C., Xiong, E., Guo, H., Wang, T., and de Gouw, J.: Significant Biogenic Source of Oxygenated Volatile Organic Compounds and the Impacts on Photochemistry at a Regional Background Site in South China, Environmental Science & Technology, 58, 20081-20090,http://doi.org/10.1021/acs.est.4c05656, 2024.

Morino, Y., Sadanaga, Y., Sato, K., Sakamoto, Y., Muraoka, T., Miyatake, K., Li, J., and Kajii, Y.: Direct evaluation of the ozone production regime in smog chamber experiments,

Atmospheric Environment, 309, 119889,http://doi.org/10.1016/j.atmosenv.2023.119889, 2023.

Qian, H., Xu, B., Xu, Z., Zou, Q., Zi, Q., Zuo, H., Zhang, F., Wei, J., Pei, X., and Zhou, W.: Anthropogenic Oxygenated Volatile Organic Compounds Dominate Atmospheric Oxidation Capacity and Ozone Production via Secondary Formation of Formaldehyde in the Urban Atmosphere, ACS ES&T Air,http://doi.org/10.1021/acsestair.4c00317, 2025.

Sakamoto, Y., Sadanaga, Y., Li, J., Matsuoka, K., Takemura, M., Fujii, T., Nakagawa, M., Kohno, N., Nakashima, Y., and Sato, K.: Relative and absolute sensitivity analysis on ozone production in Tsukuba, a city in Japan, Environmental science & technology, 53, 13629-13635,http://doi.org/10.1021/acs.est.9b03542, 2019.

Sklaveniti, S., Locoge, N., Stevens, P. S., Wood, E., Kundu, S., and Dusanter, S.: Development of an instrument for direct ozone production rate measurements: Measurement reliability and current limitations, Atmospheric Measurement Techniques, 11, 741-761,http://doi.org/10.1029/98jd00349, 2018.

Wang, W., Yuan, B., Peng, Y., Su, H., Cheng, Y., Yang, S., Wu, C., Qi, J., Bao, F., and Huangfu, Y.: Direct observations indicate photodegradable oxygenated volatile organic compounds (OVOCs) as larger contributors to radicals and ozone production in the

atmosphere, Atmospheric Chemistry and Physics, 22, 4117-4128,http://doi.org/10.5194/acp-22-4117-2022, 2022.

Wang, W., Yuan, B., Su, H., Cheng, Y., Qi, J., Wang, S., Song, W., Wang, X., Xue, C., and Ma, C.: A large role of missing volatile organic compound reactivity from anthropogenic emissions in ozone pollution regulation, Atmospheric Chemistry and Physics, 24, 4017-4027,http://doi.org/10.5194/acp-24-4017-2024, 2024.

Yuan, B., Shao, M., De Gouw, J., Parrish, D. D., Lu, S., Wang, M., Zeng, L., Zhang, Q., Song, Y., and Zhang, J.: Volatile organic compounds (VOCs) in urban air: How chemistry affects the interpretation of positive matrix factorization (PMF) analysis, Journal of Geophysical Research: Atmospheres, 117,http://doi.org/10.1029/2012jd018236, 2012.

Yuan, B., Hu, W. W., Shao, M., Wang, M., Chen, W. T., Lu, S. H., Zeng, L. M., and Hu, M.: VOC emissions, evolutions and contributions to SOA formation at a receptor site in eastern China, Atmospheric Chemistry and Physics, 13, 8815–8832, https://doi.org/10.5194/acp-13-8815-2013, 2013.

Zhou, J., Wang, W., Wang, Y., Zhou, Z., Lv, X., Zhong, M., Zhong, B., Deng, M., Jiang, B., and Luo, J.: Intercomparison of measured and modelled photochemical ozone production rates: Suggestion of chemistry hypothesis regarding unmeasured VOCs, Science of The Total Environment, 951, 175290, http://doi.org/10.1016/j.scitotenv.2024.175290, 2024a.

Zhou, J., Zhang, C., Liu, A., Yuan, B., Wang, Y., Wang, W., Zhou, J.-P., Hao, Y., Li, X.-B., and He, X.: Measurement report: Vertical and temporal variability in the near-surface ozone production rate and sensitivity in an urban area in the Pearl River Delta region, China, Atmospheric Chemistry and Physics, 24, 9805-9826, doi: 10.5194/acp-24-9805-2024, 2024b.

---

## Author Comment (AC2)

We extend our sincere gratitude to the reviewer's valuable guidance provided throughout the review process, which have significantly contributed to the paper's quality. Our responses are listed below, presented in red, following the reviewers' comments, which are in black. The revisions made to the manuscript are highlighted in yellow.

Zhong and colleagues present measurements and detailed analysis using constrained box model approaches of in-situ ozone formation at a field site in the PRD, using a newly-developed direct measurement of ozone production rates, alongside measurements of various atmospheric chemical species/photochemical parameters.

The paper presents an extensive exploration of the measurements, assessing the NOx- and VOC-dependence of the measured and modelled ozone formation, and relating this to (e.g.) missing VOC species. The approach is logical and largely well described (although I have some significant suggestions for clarifications — below), and the work represents a good advance in approaches to analysis of these new measurement approaches/data and could make a valuable contribution; in particular the assessment of ozone production "gaps" vs model with co-reactant concentrations/conditions (NOx/VOC sensitivity)

My principal concern is that the degree of accuracy (and maybe precision) of the measurements may be overestimated, and that the analysis of these – still relatively new – measurements is taken further than the data uncertainties really justify; that the data are over-interpreted.

Thank you for your suggestion. We have thoroughly discussed the measurement accuracy and uncertainties of the custom-made net O3 production rate (NPOPR) detection system in our previous studies (Hao et al., 2023; Zhou et al., 2024b), and found that the measurement accuracy of the NPOPR detection system is determined as 13.9 %, this is estimated from the systematic errors inherent in the system representing the maximum systematic error resulting from photochemical O3 production in the reference chamber.

These errors arise from photochemical O3 productions in the reference chamber, because of the UV protection Ultem film that only filters out the sunlight with wavelengths less than 390 nm. Consequently, photochemical O3 production from sunlight wavelengths between 390 nm and 790 nm still exist in the reference chamber and causes the systemic errors mentioned above.

Furthermore, according to the  $P(O_3)_{\text{net}}$  evaluation method listed in Eq. (5) in the main text, the measurement error of  $P(O_3)_{\text{net}}$  depends on the estimation error of Ox in the reaction and reference chambers, which includes the measurement error of  $O_X$  of CAPS-NO2 monitor and the error caused by the light-enhanced loss of O3. These collective measurement error of  $P(O_3)_{\text{net}}$  is referred to as the measurement precision of the NPOPR detection system, which is different with the measurement accuracy described above. This error refers to the degree of consistency or repeatability observed in a set of measurements by the NPOPR detection system. To make the description clearer, we have added this explanation in lines 127- 137 in the main text:

"The mean residence time in the reaction chamber is 0.15 h at the air flow rate of 2.1 L min-1, and the limit of detection (LOD) of the NPOPR detection system is 0.86 ppbv h-1 at the sampling air flow rate of 2.1 L min-1, which is obtained as three times the measurement error of  $P(O_3)_{net}$  (Hao et al., 2023). The measurement error of  $P(O_3)_{net}$  is determined by the uncertainty in the Ox mixing ratio estimated for both the reaction and reference chambers. This uncertainty combines (i) the measurement uncertainty of the CAPS-NO2 monitor used to derive Ox and (ii) the error induced by light-enhanced O3 loss inside the chambers. Taken together, these contributions define the measurement precision of the NPOPR detection system. In addition, the measurement accuracy of the NPOPR detection system is 13.9 %, corresponding to the maximum systematic error arising from photochemical O3 production in the reference chamber (Hao et al., 2023; Zhou et al., 2024b); details are given in Sect. S1 in the supplementary materials."

And S1 in the supplementary materials:

**"S1. Measurement error of P(O3)net of the NPOPR detection system**

We have thoroughly described the measurement error of  $P(O_3)_{net}$  of the NPOPR detection system in our previous study (Hao et al., 2023; Zhou et al., 2024b). The measurement error of  $P(O_3)_{net}$  depends on the estimation error of Ox in the reaction and reference chambers, which includes the measurement error of  $O_X$  of CAPS-NO2 monitor and the error caused by the light-enhanced loss coefficient of  $O_3(\gamma)$ ,

which can be calculated as follows:

$$\left(O_{X}\right)_{\text{error}} = \sqrt{\left(O_{X\gamma}\right)_{\text{error}}^{2} + \left(O_{X_{CAPS}}\right)_{\text{error}}^{2}}$$
(S1)

where  $(O_X)_{error}$  represents the absolute error in the estimated  $O_X$  concentration in the reaction and reference chambers, which results from the quadratic propogation of the absolute errors  $(O_{X_{\gamma}})_{error}$  and  $(O_{X_{CAPS}})_{error}$ . Here,  $(O_{X_{CAPS}})_{error}$  signifies the measurement error of the  $O_X$  measured by the CAPS-NO2 monitor, while  $(O_{X_{\gamma}})_{error}$  denotes the error associated with the  $\gamma$ -corrected Ox of the chambers, where  $\gamma$  represent the light-enhanced  $O_3$  loss coefficient.

To get  $(O_{X_{CAPS}})_{error}$ , we calibrated the CAPS-NO2 monitor as follows: a. injected ~10–100 ppbv of NO2 for 30 minutes to passivate the surfaces of the monitor and then injecting ultrapure air for ~ 10 minutes to ensure the zero point did not drift, according to the ultrapure air condition, the LOD of CAPS was 0.88 and 0.02 ppbv (3  $\sigma$ ) at an integration time of 35 and 100 s, respectively; b. injected a wide range of NO2 concentration (from 0–160 ppbv) prepared from a NO2 standard gas (with the original concentration of 2.08 ppmv) mixed with ultrapure air into the CAPS-NO2 monitor, repeated the experiments for three times at each NO2 concentration, the final results are shown in Fig. S16.

Figure S16: Calibration results of the CAPS NO2 monitor at different NO2 mixing ratios. The y-axis represents the NO2 mixing ratios measured by the CAPS NO2 monitor, and the x-axis represents the prepared NO2 mixing ratios prepared from the diluted NO2 standard gas.

We fitted the calibration results with a 68.3 % confidence level, and the blue line in Fig. S16 represents the maximum fluctuation range under this confidence level,  $(O_{X_{CAPS}})_{error}$  was then calculated from the fluctuation range of the 68.3 % confidence interval of the calibration curve, the relationship between the  $(O_{X_{CAPS}})_{error}$  and the measured Ox value ([Ox]measured) can be expressed as a power function curve, as shown in Eq. (S2):

$$(O_{X_{CAPS}})_{error} = 9.72 \times [0x]_{measured}^{-1.0024}$$
 (S2)

We acknowledge that this power function has been derived from calibration data of the OX concentrations

ranged from 20 ppbv to 160 ppbv. Utilizing this function outside this calibrated range, especially at very low  $O_X$  concentrations, may result in errors that are disproportionately large and may not accurately capture the true variability of the measurement errors. In this study, the  $O_X$  concentrations ranged from 18 to 148 ppbv, which falls into the calibration range. Consequently, this power function is deemed appropriate for estimating the  $(O_{X_{CAPS}})_{error}$  throughout the whole measurement period.

 $(O_{X\gamma})_{error}$  was derived from the light-enhanced loss of  $O_3$  in the reaction and reference chambers at 2.1 L min-1, the flow rate used during the observation campaign. To establish the calibration curve, we performed an outdoor experiment:  $O_3$  (~ 130 ppbv), produced by an  $O_3$  generator (P/N 97-0067-02, Analytic Jena US, USA), was induced into the two chambers. Zero air was co-injected with the  $O_3$  to suppress any photochemical  $O_3$  production outdoors. This setup allowed us to monitor daytime changes in the photolysis frequencies of various species. We simultaneously recorded  $J(O^1D)$ , T, RH, P and  $O_3$  mixing ratios at the inlets and outlets of both chambers. T and RH were measured with a thermometer (Vaisala, HMP110, USA). The light-enhanced  $O_3$  loss coefficient ( $\gamma$ ) was then calculated using Eq. (S3):

$$\gamma = \frac{d[O_3] \times D}{\omega \times [O_3] \times \tau} \tag{S3}$$

where  $d[O_3]$  represents the difference between the  $O_3$  mixing ratios at the inlets and outlets of both chambers (i.e., the light-enhanced  $O_3$  loss); D is the diameter of the chambers;  $\omega$  is the average velocity of  $O_3$  molecules;  $[O_3]$  is the injected  $O_3$  mixing ratio at the inlet;  $\tau$  is the average residence time of the air in the reaction and reference chambers. The relationship between  $J(O^1D)$  and  $\gamma$  is shown in Fig. S18, the obtained  $\gamma$ - $J(O^1D)$  equation was used to correct  $d[O_3]$  in both chambers during the daytime, thereby eliminating the influence of light-enhanced loss. Our previous study has shown that after this correction,  $d[O_3]$  showed no clear correlation with RH for either chamber (Hao et al., 2023), indicating that RH did not affect the  $O_3$  mixing ratio during the observation period. When quantifying  $d[O_3]$  from ambient air measurements, we first calculate  $\gamma$  from the measured  $J(O^1D)$  using the  $\gamma$ - $J(O^1D)$  equations listed in Fig. S17 for each chamber, then compute  $d[O_3]$  from the measured  $[O_3]$  and Eq. (S3).

Figure S17: The relationship between  $\gamma$  and  $J(O^1D)$  in the reaction and reference chambers, the shaded areas represent the maximum range of fluctuation under this confidence level.

When injecting ambient air into the NPOPR system, the error of  $P(O_3)_{net}$  with a residence time of  $\tau$  can be calculated using Eq. (S4):

$$P(O_3)_{\text{net\_error}} = \frac{(O_{X\gamma})_{\text{rea\_error}}^2 + ((9.72 \times [(O_X]_{\text{rea\_measured}})_{\text{rea\_std}}^2) + (O_{X\gamma})_{\text{ref\_error}}^2 + ((9.72 \times [(O_X]_{\text{ref\_measured}})_{\text{ref\_std}}^2))^2}{\tau}$$

$$(S4)$$

where  $(O_{X_{\gamma}})_{\text{rea error}}$  and  $(O_{X_{\gamma}})_{\text{ref error}}$  represent the measurement error due to light-enhanced loss of  $O_3$  in the reaction and reference chambers, respectively, and  $(9.72 \times [O_X]_{\text{measured}}^{-1.0024})_{\text{rea_std}}$  and  $(9.72 \times [O_X]_{\text{measured}}^{-1.0024})_{\text{ref_std}}$  represent the standard deviation of  $O_X$  in the reaction and reference chambers, respectively, caused by the CAPS  $NO_2$  monitor with an integration time period of 100 s. Combined with the associated residence time  $\langle \tau \rangle$  under different flow rates, i.e.,  $\langle \tau \rangle$  was 0.16 h at a flow rate of  $2.1 \text{ L min}^{-1}$ . In our previous research (Hao et al., 2023), we evaluated the residence time error and determined it to be approximately 0.0015, when we considered this error in the calculation of  ${}^{\circ}P(O_3)_{\text{net}}$  error', we observed a minimal reduction in the  ${}^{\circ}P(O_3)_{\text{net}}$  error' values, ranging from 0 to 4% [0.25-0.75 percentile]. This impact is considered negligible in relation to the overall  ${}^{\circ}P(O_3)_{\text{net}}$  error' as presented in Eq. S4. Consequently, we did not consider the uncertainty associated with the residence time in our calculations. We note that this collective measurement error of  $P(O_3)_{\text{net}}$  is referred to as the measurement precision of the NPOPR detection system, which is different with the measurement accuracy of the NPOPR detection system described above."

I think this is certainly the case for the model, considering the VOC coverage (correctly) identified and uncertainties in e.g. knowledge of HONO (not measured directly), and I would ask the authors to consider carefully if the measured  $P(O_3)$  is really good to 10% accuracy – and hence if quite such an extensive set of analysis is warranted. I'm conscious that lots of things will more-or-less co-vary diurnally within the measurement uncertainty (concentrations, j, T,  $O_3...$ ) – is it really possible to extract missing reactants at a few % accuracy from within the combined measurement and model uncertainties?

Thank you for your insightful comment. We have thoroughly discussed the measurement error (precision) and accuracy of the custom-made NPOPR detection system, as described above. The evaluated measurement uncertainty of the  $P(O_3)_{net}$  during the observation period are shown in Fig. S13. We see that the measurement uncertainty decreased with increasing  $P(O_3)_{net}$  values, which ranges from 0-23%.

"

Figure S13: (a) Time series of measured  $P(O_3)_{\text{net}}$ \_Mea,  $P(O_3)_{\text{net}}$ ^{+NO} and  $P(O_3)_{\text{net}}$ ^{+VOCs} based on sensitivity experiments using the NPOPR detection system, with an enlarged view for an  $O_3$  pollution day (October 26, 2023) and a normal ( $O_3$  non-pollution) day (October 13, 2023). The shaded areas represent the errors of each measured term, calculated from the instrument measurement uncertainties given in Hao et al. (2023). (b) Relative errors of measured  $P(O_3)_{\text{net}}$ \_Mea,  $P(O_3)_{\text{net}}$ ^{+NO}, and  $P(O_3)_{\text{net}}$ ^{+VOCs} as a function of their measured values, ..."

More details concerning the measurement uncertainty are added in lines 503-506 in the modified manuscript: "The time series of measured  $P(O_3)_{net}$  Mea,  $P(O_3)_{net}$  had  $P(O_3)_{net}$  based on sensitivity experiments using the NPOPR detection system are shown in Fig. S13. We see the measurement uncertainty decreased with increasing  $P(O_3)_{net}$  values: it reaches approximately 23% when  $P(O_3)_{net}$  is around 0 ppbv h-1, but falls below 3% when  $P(O_3)_{net}$  is around 50 ppbv h-1."

Fig. S13b shows that the higher measurement uncertainty usually appears when the  $P(O_3)_{\text{net}}$  is relatively low (e.g., in the early morning, evening, or late afternoon), whereas the relatively large  $P(O_3)_{\text{net}}$  missing occur mainly around noon. The different diurnal patterns of measurement uncertainty (caused by concentrations, j, T,  $O_3...$ ) and the modelling bias responsible for the  $P(O_3)_{\text{net}}$  missing indicate that the two do not co-vary on a diurnal basis. The  $P(O_3)_{\text{net}}$  missing may be small when averaged over daytime; however, it can reach  $\sim$  33% around noon. This value is much higher than the measurement uncertainty

at that time (

Figure 3: The time series and diurnal variations of  $P(O_3)_{net}$  Mea and  $P(O_3)_{net}$  Mod (Case  $D_1$ - $D_4$ ) during the observation period, with an enlarged view for an  $O_3$  pollution day

(October 26, 2023) and a normal (O3 non-pollution) day (October 14, 2023); The shaded areas in (a) represent rainy days.

Figure 5: (a) Time series and (b) diurnal variations of  $P(O_3)_{net}$  Mea and  $P(O_3)_{net}$  Mod (Case  $D_1$ – $E_3$ ) during the observation period, with an enlarged view for an  $O_3$  pollution day (October 26, 2023) and a normal ( $O_3$  non-pollution) day (October 14, 2023); (b) Diurnal variations excluding rainy days. The shaded areas in (a) represent rainy days.

From Fig. 3a, we see the modelled results on individual days varies, which make the  $P(O_3)_{\text{net}}$  missing different with the overall diurnal variation (shown in Fig. 3b). For example,  $P(O_3)_{\text{net}}$  Mod of Case D2 increased more on 17 October 2023 than that on October 14 2023, while  $P(O_3)_{\text{net}}$  Mod of Cases D3-D4 increased more on October 17 2023 than that on October 14 2023. These results indicated that the overall increasing or decreasing trends can't represent all individual days during the observation period. On the other hand, the constraining of more OVOCs only corrects the outcome, not the related reaction processes. For example, the RO2 radicals, which are the main intermediates that drive the oxidation chain during photochemical O3 formation, may vary day-to-day. We added the related description for Fig. 3 in lines 339-341 of the modified manuscript:

"Furthermore, the enlarged days in Fig. 2 reveal day-to-day variations in  $P(O_3)_{net}$  Mod across the different cases, underscoring that the overall diurnal pattern described above does not resolve this variability."

We added the related description for Fig. 5 in lines 492-494 of the modified manuscript:

"However, we observe a slight difference in the diurnal trends of  $P(O_3)_{net}$  across different days (enlarged view in Fig. 5); this depicts the overall pattern for the observation period described above does not capture day-to-day variability."

We detected an error when check the  $P(O_3)_{net}$ \_Mod data from Case  $D_2$  and Case D1, the overall daytime  $P(O_3)_{net}$ \_Mod from Case D2 showed a slightly decreasing trend (0.5%) compared to  $P(O_3)_{net}$ \_Mod from Case D1, therefore, we have changed the sentence "However, the daytime average value of  $P(O_3)_{net}$ \_Mod from Case  $D_2$  increased by only 0.5% compared to Case  $D_1$ , this indicates that the dominant OVOCs species that causes  $P(O_3)_{net}$ \_Missing may vary between Heshan and Dongguan." to "However, the daytime mean  $P(O_3)_{net}$ \_Mod in Case  $D_2$  decreased by 0.5% compared with Case  $D_1$ , indicating that the dominant OVOC species responsible for  $P(O_3)_{net}$ \_Missing may differ between Heshan and Dongguan." in lines 390-391 of the modified manuscript.

I recognise the method is published in the Hao et al (2023) paper, but would encourage the authors to include more discussion of measurement uncertainties here, and at the least a detailed justification of the statement "...measurement uncertainties around 10%" (L79). this should include, separately, accuracy, precision, and selectivity/bias – the impact of wall artefacts on the measured P(O3), which may vary with conditions (j, RH, VOC/NOx levels), and would best appear around L120.

Thank you for your suggestion. The relevant discussion and data analysis of measurement uncertainty of the custom-made NPOPR detection system are provided above. In summary, our measurement error is calculated in real time based on the light intensity and the Ox concentrations of the ambient air. As mentioned above, the related experiments and data analysis are shown in the Supplementary Materials S1. The related discussion has been added in lines 127-137 of the modified manuscript:

"The mean residence time in the reaction chamber is 0.15 h at the air flow rate of 2.1 L min-1, and the limit of detection (LOD) of the NPOPR detection system is 0.86 ppbv h-1 at the sampling air flow rate of 2.1 L min-1, which is obtained as three times the measurement error of  $P(O_3)_{net}$  (Hao et al., 2023). The measurement error of  $P(O_3)_{net}$  is determined by the uncertainty in the Ox mixing ratio estimated for both the reaction and reference chambers. This uncertainty combines (i) the measurement uncertainty of the CAPS-NO2 monitor used to derive Ox and (ii) the error induced by light-enhanced O3 loss inside the chambers. Taken together, these contributions define the measurement precision of the NPOPR detection system. In addition, the measurement accuracy of the NPOPR detection system is 13.9 %, corresponding to the maximum systematic error arising from photochemical O3 production in the reference chamber (Hao et al., 2023; Zhou et al., 2024b); details are given in Sect. S1 in the supplementary materials."

Furthermore, we checked the measurement uncertainties of different O3 production sensors worldwide and confirmed that the uncertainties ranged from 10-30%. Therefore, we have changed the sentence to "Through practical applications in field observations, scholars generally agree that these detection systems offer rapid stability and high precision, with measurement uncertainties ranged from 10-30 %." in line 84 of the modified manuscript.

**Corrections / Comments**

L47 and following – it would be useful to distinguish between NO titration of O3 – ie NOx/O3 PSS shifts – and net production of Ox (which is what we really mean by ozone production). Several points in the text later (eg L134) there is reference to titration reducing ozone production – I'd argue that this is PSS shift, not a change in the ozone production chemistry, and a different terminology might help.

We apologize for the ambiguous description. We have changed the sentence to: "IR is defined as the change in  $P(O_3)_{net}$  per unit change in precursor concentration ( $\Delta S(X)$ ): a negative IR value indicates that reducing the precursor concentration increases  $O_3$  production (e.g., decrease NOx would increase  $O_3$  through OH mediate effect), ..." in line 156 of the modified manuscript.

Furthermore, we changed "This midday transition to NOX-limited conditions is chemically reasonable, where intensified NO2 photolysis boosts OX production while

concurrently diminished NO titration and declining VOCs emissions collectively favor NOX-sensitive chemistry during peak sunlight hours (Wang et al., 2023)." to "This midday transition to NOX-limited conditions is chemically reasonable, where intensified NO2 photolysis boosts OX production while persistent photochemistry consumption without replenishment (Wang et al., 2023)." In lines 512-513 of the modified manuscript to make the description clearer.

L77 please acknowledge / include the pioneering work of Brune and colleagues (Cazorla et al., 2012) as the first "modern" MOPS system developers (I realise this is referenced later).

Thank you for your reminder. We have added the pioneering work of Brune and Cazorla as references (Cazorla and Brune, 2009; Cazorla et al., 2012) in line 81 of the modified manuscript:

"To date, several  $P(O_3)_{net}$  detection systems based on the dual-reaction chamber technique have been developed, referred to as measurement of  $O_3$  production sensor (MOPS),  $O_3$  production rate measurement system (O3PR),  $O_3$  production rates instrument (OPRs), net photochemical  $O_3$  production rate detection system (NPOPR), Mea-OPR, or  $O_3$  production rate-cavity ringdown spectroscopy system (OPR-CRDS) (Baier et al., 2015; Cazorla and Brune, 2009; Cazorla et al., 2012; Sadanaga et al., 2017; Sklaveniti et al., 2018; Hao et al., 2023; Wang et al., 2024c; Tong et al., 2025)."

L79 Measurement uncertainties – I do not think 10% is realistic – see general comments above

We apologize that we made a mistake here. As mentioned above, we checked the measurement uncertainties of different O3 production sensors worldwide and confirmed that the uncertainties ranged from 10-30%. Therefore, we have changed the sentence to "Through practical applications in field observations, scholars generally agree that these detection systems offer rapid stability and high precision, with measurement uncertainties ranged from 10-30 %." in the modified manuscript.

L100 is there much emission / chemical heterogeneity around the site? e.g. on the timescale of NOx PSS (1 min+) or HONO PSS (10-15min+)?

Thank you for your thoughtful consideration. The observation site, the Guangdong Atmospheric Supersite of China, is located in a farmland conservation zone and forested region at the suburban area of Heshan City. There are no major local industrial emission sources and the motorcycles dominate urban transport in Heshan City. However, the supersite experiences minimal spatial heterogeneity in either primary emissions or chemical composition as it is located on a small mountain approximately 3 km from the nearest area with heavy traffic emissions. With a mean wind speed is 2.8 m s-1 during the observation period, the air mass originating from the traffic corridor requires ~ 17 min to reach the supersite; consequently, rapid dilution and initial photochemical processing of exhaust plumes occur before they reach the supersite.

We changed the sentence "The supersite is situated in the downwind area of Guangzhou and Foshan and is characterized by active secondary reactions. It lies at the intersection of forest-agricultural and urban systems, representing a typical rural station. The surrounding area primarily consists of farmland conservation zones and forested areas, with no significant industrial emissions. It is suitable for comprehensive monitoring and research on regional atmospheric complex pollution in the PRD (Mazaheri et al., 2019)." to "The supersite is situated in the downwind area of Guangzhou, Foshan, and Dongguan, a region characterized by active secondary reactions and serving as a receptor for pollution transported from the industrial and urban centers (Luo et al., 2025; Huang et al., 2020). The surrounding area is primarily composed of farmland conservation zones and forested regions, with no major industrial sources. The supersite sits on a small mountain ~ 3 km from the nearest area heavy traffic corridor; at the observed mean wind speed of 2.8 m s-1, the air mass from the corridor takes ~ 17 min to arrive. This separation limits spatial heterogeneity in both emissions and chemical composition, making the site well-suited for comprehensive monitoring and research on complex regional air pollution in the PRD (Mazaheri et al., 2019)." in lines 104-111 of the modified manuscript.

L131 Explain how the VOC addition amounts were determined/apportioned between the two species

According to the previous study, the selection of the VOCs indicator for the O3 formation sensitivity measurement can be determined using the VOCs measured previous to the O3 formation sensitivity (OFS) measurements (Carter

et al., 1995; Wu et al., 2022). We first calculated the total VOCs reactivity using the daytime VOCs measured from 20 September to 3 October and from 4-11 October 2023 at the observation site. For the OFS measurement from 4-11 October, we used the total VOCs reactivity measured from 20 September to 3 October 2023, VOCs indicators included isopentane as the representative alkane, ethylene and isoprene as the representative alkenes, and toluene as the representative aromatic hydrocarbon. From 13-26 October, we used the averaged daytime total VOCs reactivity measured from 4 to 11 October 2023, VOCs indicators included Ethylene was used as the representative nonmethane hydrocarbon (NMHC) indicator and formaldehyde as the representative oxygenated volatile organic compound (OVOC) indicator. The related description is added in lines 143-151 of the modified manuscript:

"Following Carter et al. (1995) and Wu et al. (2022), we select VOCs surrogates for the OFS measurement on the basis of ambient measurements previous to the measurements. From 4–11 October, the tracer mixture was formulated from the average daytime total VOC reactivity measured during 20 September–3 October 2023, and isopentane served as the alkane surrogate, ethylene and isoprene as the alkene surrogates, and toluene as the aromatic surrogate. For 13–26 October 2023, we used the average daytime total VOC reactivity obtained during 4–11 October 2023; ethylene represented non-methane hydrocarbons (NMHCs) and formaldehyde represented oxygenated VOCs (OVOCs). Each surrogate was mixed in proportion to its category's share of the ambient reactivity, and the effective precursor strength (NO or VOCs) should increase by 20 % relative to the original ambient level."

L202 E10 – I do not follow how the net P(O3) is equal to P(O3) multiplied by the change in P(O3) divided by the (natural log of) change in X.

We apologize for the error in the equation expression. The description has been revised in lines 223-227 of the modified manuscript:

"We calculated the modelled OFS using the absolute  $P(O_3)_{net}$  sensitivity, adapted from the logarithmic derivative approach of Sakamoto et al. (2019). It is defined as the change in  $P(O_3)_{net}$  with respect to the natural logarithm of  $O_3$  precursor concentrations. This method facilitates the quantitative assessment of how reductions in  $O_3$  precursors contribute to the overall reduction of  $P(O_3)_{net}$  over a period or within a region. The formula is as follows:

Absolute
$$P(O_3)_{\text{net}} = \frac{dP(O_3)_{\text{net}}}{d \ln[X]}$$
 (10)

L234 not sure "stronger" photochemical reactions is right word – do you mean higher photolysis rates? An alternative explanation for the (slightly) lower concs might be greater solar heating/higher BLH/more dilution on the hotter/sunnier/higher P(O3) days – evidence in the BLH data?

Thank you for your suggestion. Yes, we mean higher photolysis rates occur on  $O_3$  pollution days (see Fig. S4a). We further checked the planetary boundary-layer height (PBLH) on both  $O_3$  pollution days and normal days. We found that in the morning the PBLH was higher on  $O_3$  pollution days, whereas in the afternoon it became lower than on normal days (see Fig. S4k). During the period of strongest sunlight (11:00-14:00), the PBLH on  $O_3$  pollution days and normal days does not appear statistically different (*t-test*, p=0.45). Therefore, the lower concentrations of TVOC and  $NO_X$  on  $O_3$  pollution days compared with normal are not likely due to changes in PBLH change or increased dilution.

Accordingly, we have changed the description from "This suggests that stronger photochemical reactions occur on O3 pollution days, leading to lower daytime concentrations of precursors compared to normal days." to "As the PBLH on O3 pollution days and normal days does not differ statistically during the period of strongest solar radiation (11:00-14:00, *t*-test, *p*=0.45, see Fig. S4k), the lower daytime concentrations/mixing ratios of O3 precursors on O3 pollution days than on normal days may be due to higher photolysis rates on O3 pollution days (see Fig. S4a)." in lines 257-259 in the modified manuscript.

L305 give the missing P(O3) as a % also – maybe 24 hour mean. Statistical test – I assume *P* level of 0.05 not 0.5?

Sorry for the confusion description. We have modified the sentence to make it clearer, as shown in lines 338-343 in the modified manuscript:

"On non-rainy days, the averaged daytime  $P(O_3)_{net}$  Missing reached 4.5±7.6 ppbv h-1, accounting for 31% of the total measured  $P(O_3)_{net}$ . The averaged daytime  $P(O_3)_{net}$  Missing values on  $O_3$  pollution days were statistically higher than those on normal days (*t-test*, p<0.05), suggesting that while the supplementary mechanisms explored in the model may contribute to

some extent, they are unlikely to be the dominant cause of the  $P(O_3)_{net}$  Missing."

L325+: Is there really sensitivity in the correlations to identify particular causes?

Thank you for your question. The correlation analysis is only a preliminary examination of the factors that may be related to the  $P(O_3)_{\text{net}}$ \_Missing in the model, aimed at guiding further investigation; it therefore does not allow us to identify specific causes. We have softened the wording in the sentences in lines 367 to 374in the modified manuscript:

"To explore the possible drivers of  $P(O_3)_{net}$  Missing, we correlated it with TVOC, NOX,  $J_{O1D}$ , T, and OX separately for O3 pollution days and normal days (Fig. S11). On O3 pollution days,  $P(O_3)_{net}$  Missing exhibited a moderate positive correlation with VOCs ( $r^2 = 0.4$ , R = 0.2, t = 2.9) and NOX ( $r^2 = 0.5$ , R = 0.2, t = 3.8), confirming that the  $P(O_3)_{net}$  Missing is larger at higher precursor concentrations/mixing ratios (both t > critical 2.0, p < 0.05), consistent with earlier box-model studies (Whalley et al., 2021; Ren et al., 2013; Zhou et al., 2024a). A moderate positive correlation is also found with  $J_{O1D}$  on both O3 pollution days and normal days, with  $r^2$  values of 0.5 and 0.4, respectively. On normal days all correlations collapse ( $r^2 < 0.2$ , p > 0.1), implying that the model deficit is not tied to the measured precursors under low-NOx conditions and may instead related to the missing mechanisms for unmeasured photolabile VOCs."

**L345 is it valuable to include all of D1-D4 – cut straight to the final case, D4?**

By setting different simulated scenarios from Cases D1–D4, we primarily wanted to check quantitively whether the additional mechanisms and the measured OVOCs have a significant influence on  $P(O_3)_{net}$ \_Missing. The configurations of each scenario are as follows: Case A considers only the simplified chemical reaction mechanism from MCM v3.3.1; Case B incorporates the HO2 uptake by ambient aerosols mechanism based on Case A; Case C further includes the dry deposition processes of key species on top of Case B; Case D1 extends Case C by adding the N2O5 uptake mechanism and CI-related heterogeneous reaction mechanisms. Case D2 includes the measured OVOCs based on Case D1— namely, acetaldehyde, acrolein, acetone, and butanone—which were considered qualitatively as potential contributors to  $P(O_3)_{net}$ \_Missing in Dongguan in our previous study (Zhou et al., 2024); Case D3 constrained all measured OVOC species in Heshan based on Case D2;

Case D4 constrained chlorine-containing VOCs (i.e., all measured VOC species listed in Table S8 that could be input into the OBM model). Detailed simulation parameter settings are provided in the main text and the Supplementary Materials (Table S3). We believe that this step-by-step simulation process is necessary for a better understanding of the different mechanisms and the impact of OVOCs on  $P(O_3)_{net}$ \_Missing.

Fig S13 – It is very hard to see the change in *P*O3 from the added NO/added VOCs – suggest show a zoom in on a polluted/non-polluted day in addition so the data can be seen. The uncertainty ranges look very small on this figure?

Thank you for your suggestion. We have added a zoom in on a  $O_3$  pollution day (October 26, 2023) and a normal ( $O_3$  non-pollution) day (October 13, 2023) in Fig. S13 to make it clearer.

"

Figure S13: (a) Time series of measured  $P(O_3)_{\text{net}}$ \_Mea,  $P(O_3)_{\text{net}}^{+NO}$  and  $P(O_3)_{\text{net}}^{+VOCs}$  based on sensitivity experiments using the NPOPR detection system, with an enlarged view for an  $O_3$  pollution day (October 26, 2023) and a normal (O3 non-pollution) day (October 14, 2023). The shaded areas represent the errors of each measured term, calculated from the instrument measurement uncertainties given in Hao et al. (2023); (b) Relative errors of measured  $P(O_3)_{\text{net}}$ \_Mea,  $P(O_3)_{\text{net}}^{+NO}$ , and  $P(O_3)_{\text{net}}^{+VOCs}$  as a function of their measured values; ...

The measurement uncertainties, shown as shaded areas in Fig. S13, are calculated as the measurement error described in Hao et al. (2023) and Zhou et al. (2024): when injecting ambient air into the NPOPR system, the error of  $P(O_3)_{net}$  with a residence time of  $\tau$  can be calculated using this equation:

$$P(O_3)_{\text{net\_error}} = \sqrt{(O_{X\gamma})_{\text{rea\_error}}^2 + ((9.72 \times [(O_X]_{\text{rea\_measured}})_{\text{rea\_std}}^{-1.0024})_{\text{rea\_std}}^2 + ((9.72 \times [(O_X]_{\text{ref\_measured}})_{\text{ref\_std}}^{-1.0024})_{\text{ref\_error}}^2 + ((9.72 \times [(O_X]_{\text{ref\_measured}})_{\text{ref\_std}}^{-1.0024})_{\text{ref\_std}}^2 + ((9.72 \times [(O_X]_{\text{ref\_measured}})_{\text{ref\_std}}^2 + ((9.72 \times [(O_X]_{\text{ref\_std}})_{\text{ref\_std}}^2 + ((9.72 \times$$

 $(O_{X\gamma})_{rea\_error}$  and  $(O_{X\gamma})_{ref\_error}$  represent the measurement error due to light-enhanced loss of O3 in the reaction and reference chambers, respectively, and  $(9.72\times[O_X]_{measured}^{-1.0024})_{rea\_std}$  and  $(9.72\times[O_X]_{measured}^{-1.0024})_{ref\_std}$  represent the standard deviation of OX in the reaction and reference chambers, respectively, caused by the CAPS NO2 monitor with an integration time period of 100 s. Combined with the associated residence time  $\langle \tau \rangle$  under different flow rates, i.e.,  $\langle \tau \rangle$  was 0.063 h at a flow rate of 5 L min-1. Therefore, the instrument measurement error is determined by the measurement error of OX in the reaction and reference chambers, which may also be influence by the lightenhanced loss of OX in the reaction and reference chambers under ambient conditions when the light intensity (especially J(O1D)) and O3 mixing ratios are high. The related description is added now in the Supplementary Materials S1.

To check the relative error of  $P(O_3)_{net}$ \_Mea,  $P(O_3)_{net}$ ^{NO} and  $P(O_3)_{net}$ ^{VOCs}, we plotted the measurement error as a function of their measured values (see Fig. S13b). We find that the uncertainty decreased with increasing data values: it reaches approximately 23% when  $P(O_3)_{net}$  is around 0 ppbv h-1, but falls below 3% when  $P(O_3)_{net}$  is around 50 ppbv h-1.

We have now added such kind of description in lines 503-506 in the modified manuscript: "The time series of measured  $P(O_3)_{\text{net}}$ \_Mea,  $P(O_3)_{\text{net}}$ \_{NO}^{+NO} and  $P(O_3)_{\text{net}}$ \_{VOCs}^{+VOCs} based on sensitivity experiments using the NPOPR detection system are shown in Fig. S13. We see the measurement uncertainty decreased with increasing  $P(O_3)_{\text{net}}$  values: it reaches approximately 23% when  $P(O_3)_{\text{net}}$  is around 0 ppbv h-1, but falls below 3% when  $P(O_3)_{\text{net}}$  is around 50 ppbv h-1."

L440ish: Ozone regime – there is only one data point in the afternoon showing a VOC limited regime (14:00). Is there really a shift from VOC to NOx to VOC to NOx limited through the day – can the data really show this? I am conscious that there are not many days going into these averages. How is "transition regime" defined for the measurements? The explanation (L442): P(O3) measurement is not affected by NOx/O3 titration (or PSS) – rather it measures change in the net Ox production.

Yes, we obtained only one data point showing a VOC-limited regime in the afternoon from the direct measurement. However, this diurnal profile shown in Fig. S14 is compiled from all days on which O3 formation sensitivity was directly measured; it represents the overall trend during the observation period and does not reflect the day-to-day variation. As described in the main text, in total 11 days were incorporated into this calculation during the observation period, which includes 4–5, 11, 13–17, and 24–26 October 2023. We added the related discussion in lines 507-509 of the modified manuscript:

"Fig. S14 shows the diurnal variation of the directly measured IR index compiled from all 11 days of OFS experiments, together with the absolute  $P(O_3)_{net}$  sensitivity to  $NO_X$  and VOCs calculated with the box model (Case  $D_1$ , Eq. (10)). It therefore depicts the overall trend across the observation period and does not reflect the day-to-day variability."

Here, we define the transition regime as the region over which the IR shows a simultaneous increase or decrease upon addition of both VOCs and NO. We added it in lines 161-162 in Sect. 2.1. "We define the transition regime as the region over which the IR shows a simultaneous increase or decrease upon addition of both VOCs and NO."

Since we are measuring  $P(O_3)_{net}$  in our NPOPR detection system, the measurement result is not affected by NOx/O3 titration (or PSS), we have modified the sentence "This midday transition to NOX-limited conditions is chemically reasonable, where intensified NO2 photolysis boosts OX production while concurrently diminished NO titration and declining VOCs emissions collectively favor NOX-sensitive chemistry during peak sunlight hours (Wang et al., 2023)." to "This midday transition to NOX-limited conditions is chemically reasonable, where intensified NO2 photolysis boosts OX

production while persistent photochemistry consumption without replenishment (Wang et al., 2023)." in lines 511-513 of the modified manuscript.

Fig 6 – please show the mean diurnals for PO3 (from the measurements) for the three regimes identified. Not sure that the rapid changes in emissions can be the explanation – the model is constrained to the observed concentrations, so it has this "built in".

Thank you for your suggestion. We have now conducted the mean diurnal cycles of  $P(O_3)_{net}$  for the three regimes identified from the direct measurements. The  $O_3$  formation sensitivity (OFS) of each day was diagnosed from its daily-integrated measurements; the mean diurnal variation for all days within the same OFS category was calculated. In total, eight days were classified as transition regime (4-5, 11, 14-15, 24-26 October 2023), two as VOC-limited regime (13 and 16 October 2023), and one as NOx-limited regime (17 October 2023). The resulting diurnal profiles are shown in Fig. S13c-d. Between 08:00 and 12:00 the mean diurnal profiles reveal a gradual shift from VOC-limited toward NOx-limited conditions within the VOC-limited category, and a similar progression from transition to NOx-limited within the transition category. We added the related description in lines 536-539 of the modified manuscript:

"To illustrate that the diurnal shift in OFS depicted in Fig. 6 is not random noise but reflects the general rule, we grouped the 11 days of direct measurements by their initial O3-formation regime, calculated their average diurnal variations, and thus reproduced the "morning-transition" phenomenon in Fig. S13c–d."

And added the mean diurnal variation of  $P(O_3)_{net}$  for the three regimes in Fig. S13c-d:

Figure S13: ... (c-e) Mean diurnal profiles of the three O3 formation regimes identified: eight days classified as transition regime (4-5, 11, 14-15, 24-26 October 2023, two as VOC-limited regime (13 and 16 October 2023), and one as NOx-limited regime (17 October 2023)."

We agree with the reviewer that the mean diurnal variations in O3 formation sensitivity (OFS) are not influenced by rapid emission changes. As mentioned above, the site is situated on a small mountain ~3 km from the nearest heavy-traffic corridor; at the observed mean wind speed of 2.8 m s-1, an air mass from the corridor takes ~17 min to arrive. This separation reduces spatial heterogeneity in both emissions and chemical composition at the observation site (see Sect. 2.1 of the revised manuscript).

A greater challenge in this regard may be the P(O3) measurements which average over an hour effectively?

We designed experiments to determine the OFS from direct measurements conducted daily from 8:00-18:00. Each measurement cycle lasted 1 hour: the first 20 min consisted of NO addition (denoted  $P(O_3)_{net}$ \_Mea+NO), the next 20 min of ambient baseline measurement ( $P(O_3)_{net}$ \_Mea), and the final 20 min of VOCs addition ( $P(O_3)_{net}$ \_Mea+VOCs). Therefore, we first interpolated  $P(O_3)_{net}$ \_Mea+NO,  $P(O_3)_{net}$ \_Mea, and  $P(O_3)_{net}$ \_Mea+VOCs to 4-min resolution and then averaged these values over 1 h to eliminate the influence of the data fluctuation. However, the 1-hour averaging may smooth out transient responses of the measured  $P(O_3)_{net}$ . The related description is now added in lines 151-153 in the modified manuscript:

"For data treatment, we first interpolated  $P(O_3)_{net}$ \_Mea+NO,  $P(O_3)_{net}$ \_Mea, and  $P(O_3)_{net}$ \_Mea+VOCs to 4-min resolution and then averaged them over 1 h to suppress data fluctuations. We caution that this 1-hour averaging may smooth out transient responses in the measured  $P(O_3)_{net}$ ."

Isnt it more that you have already shown that the model (not unexpectedly) has bias from missing VOCs and this is reflected in these analyses also?

Yes, we have shown that the bias in the model may be attributed to missing VOCs, as reflected by the comparison of OFS results obtained from direct

measurements and from modeling cases D1 and E1–E3 during the rising, stable, and declining phases of  $P(O_3)_{net}$  (as described in Sect. 4). Briefly, in modeling cases where  $P(O_3)_{net}$  Missing was reduced (Cases E1–E3), the simulated OFS occasionally shifted toward NOx-limited conditions during certain periods. This contradictory phenomenon may be related to the model's incomplete representation of the chemical mechanisms of unknown highly reactive VOCs (e.g., aldehydes and ketones), which is consistent with previous studies suggesting that diagnostic methods based on box models tend to overestimate VOC sensitivity due to the neglect of unidentified VOCs in anthropogenic emissions or their secondary products (Xu et al., 2022; Lu et al., 2010).

Therefore, we deleted the sentence "This low consistency may be related to rapid changes in precursor concentrations in the morning: the concentrations of VOCs and NOX concentrations change quickly during this period, particularly due to traffic emissions and industrial activities. These rapid variations make it challenging for the model to accurately capture the instantaneous reaction dynamics (Cao et al., 2021).", and added "These results demonstrate that the bias between measured and modeled OFS arises chiefly from missing VOCs or shortcomings in the model's chemical mechanism." in lines 551-552 to explain the reason for the inconsistency between OFS derived from the direct measurement and the model simulation methods.

L489/Fig S15 – what are cases E1-E3? Not mentioned previously and I cannot find a definition / description of these. I cannot follow L490-L505 as these cases/ scenarios are not defined

Sorry for the unclear description. In Case E1, the overall TVOC concentration was increased to compensate for  $k_{\text{OH}}$ \_Missing without distinguishing VOCs categories. In Case E2, ethylene and formaldehyde were increased to compensate for  $k_{\text{OH}}$ \_Missing. In Case E3, only formaldehyde concentration was expanded to compensate for  $k_{\text{OH}}$ \_Missing. The detailed settings of each simulation case are listed in Table S3, and the explanations concerning Cases E1-E3 are listed in lines 476-484 in the modified manuscript:

"In Case  $E_1$ , where the overall TVOC concentration was increased to compensate for  $k_{\rm OH}$ \_Missing without distinguishing VOCs categories, the compensation effect was limited due to the dilution effect of low-reactivity VOCs, resulting in a reduction of the daytime average  $P(O_3)_{\rm net}$ \_Missing proportion from 26.3 % (calculated as  $P(O_3)_{\rm net}$ \_Missing/ $P(O_3)_{\rm net}$ \_Mea) to 10.3 %. In Case  $E_2$ , where the concentrations of ethylene and formaldehyde were expanded to compensate for  $k_{\rm OH}$ \_Missing, the daytime average  $P(O_3)_{\rm net}$ \_Missing proportion reduced from 26.3 % to 17.2 %. This proportion is higher than that obtained from Case E1, which may be due to the relatively low reactivity of ethylene limited the overall compensation effect. In contrast, Case  $E_3$  compensated for  $k_{\rm OH}$ \_Missing solely by expanding the formaldehyde concentration. More details concerning the cases settings are shown in Table S3."

Conclusions – is there a comment to make on the impact of different chemical mechanisms (eg L65+) on the model/measurement agreement?

We have investigated the influence of some missing mechanisms in MCM v3.3.1, such as HO2 uptake by ambient aerosols, dry deposition, N2O5 uptake, and ClNO2 photolysis (Case D1), to the modelling results. However, some other reaction mechanisms, such as the RO2 isomerization (Crounse et al., 2012), autoxidation (Wang et al., 2017), and the accretion reactions (Berndt et al., 2018) can also effect modelled  $P(O_3)_{net}$ , but these processes have not been investigated in this study. Therefore, we have added a comment on the impact of different chemical mechanism as follows in lines 418-421 of the modified manuscript:

"Previous studies have shown that the  $RO_2$  isomerization (Crounse et al., 2012), autoxidation (Wang et al., 2017), and the accretion reactions (Berndt et al., 2018) can also effect modelled  $P(O_3)_{net}$ , but these processes have not been investigated here."

And lines 584-588 of the modified manuscript:

"These results also demonstrate that incorporating the aforementioned missing mechanisms and measured VOC species cannot fully eliminate simulation bias. Other processes, i.e., the RO2, autoxidation, and the accretion reactions can also affect modelled  $P(O_3)_{net}$ , but they have not been examined here. The negative correlation of  $P(O_3)_{net}$  Missing with the air mass aging indicates that the  $P(O_3)_{net}$  missing is not likely caused by unaccounted secondary production."

Conclusions – do you wish to add an overall comment vs the NOx- vs VOC-control on O3 observed at the site, and implications for policy vs reducing ozone

formation rates? this might usefully also remind the reader of the distinction between the in-situ formation rate (focus here) and the total level experienced (integration of local formation and upwind chemistry / advection).

Yes, we have now added an overarching statement in lines 609-616 of the modified manuscript:

"In conclusion, we quantitatively assessed the  $P(O_3)_{net}$  simulation deficits and their impact on OFS diagnosis by comparing the measured and modelled  $P(O_3)_{net}$ , and found that the unmeasured VOCs —rather than the secondary atmospheric formation —are the primary causative factor of  $P(O_3)_{net}$  Missing. Furthermore, both direct measurements and model results reveal a diurnal OFS shift dominated by the morning regime; transition and VOC-limited conditions prevailed, so prioritizing VOCs while co-controlling NOX is the most effective approach to  $O_3$  pollution control in PRD region. Our results also demonstrate that the persistent model biases risk under-estimating the local photochemical formation contribution to  $O_3$  pollution, thereby has weakening its perceived impact relative to physical transportation. Future studies should expanded VOCs measurements and combine direct  $P(O_3)_{net}$  observations with regional transport model to separate local production from up-wind advection."

**Model Approach Clarifications**

-A summary table of the different scenarios, A-E, would be very helpful – I think this is referred to but I cannot find in the SI?

There is a detailed description of different modelling scenarios from A-E, as shown in Table S3 in the SI:

Table S3. Description of different modelling scenarios and the parameter settings

| Case | Description                                                                                            | Parameter settings                 | references           |
|------|--------------------------------------------------------------------------------------------------------|------------------------------------|----------------------|
| A    | Ambient gases (NO, NO2, SO2, CO, O3),                                                                  | $O_3$ (0.27 cm s -1 )   | (Xue et al., 2014)   |
|      | HONO, 44 VOCs, meteorological                                                                          |                                    |                      |
|      | parameters (T, RH, P, BLH), photolysis                                                                 |                                    |                      |
|      | rates, and O 3 dry deposition                                                               |                                    |                      |
| В    | Case A with the addition of HO 2 uptake                                                     | $r_{\rm H2O} = 0.19$               | (Zhu et al., 2020;   |
|      |                                                                                                        |                                    | Zhou et al., 2021)   |
| C    | Case Bwith the addition of trace gases                                                                 | $NO_2 (0.6 \text{ cm s}^{-1})$     | (Zhang et al., 2003; |
|      | (NO 2 , SO 2 , H 2 O 2 , HNO 3 , PAN, HCHO) dry | $SO_2 (0.8 \text{ cm s}^{-1})$     | Xue et al., 2014)    |
|      | deposition                                                                                      | $H_2O_2$ (1.2 cm s -1 ) |                      |

Notes: Parameter values for modelling scenarios from Case A to Case D1 are set the same as those in Zhou et al. (2024a).

-Model observation constraint: It would be helpful to explain how the constraint to observations was implemented – we have model outputs on an hourly basis but how frequently where the constraints applied to the model and did concentrations evolve freely in between (shorter model integration timestep)? If the model species are not in balance a "saw tooth" effect can result in the simulated concs at the higher model time resolution between observation constraint points – was this the case and if so impacts on the P(O3) which is in effect averaged over this period?

Thank you for your insightful suggestion. In this study, the box model was constrained by on-site observations of VOCs, NO, CO, HONO, and meteorological parameters (i.e., photolysis rates, RH, *T*, and P), as described in Sect. 2.2. Constraints were applied every hour, with no free concentration evolution in between. We have added the relevant description in lines 181–182 of the revised manuscript.

"The constraints are applied to the model every 1 h, with no free concentration evolution in between."

From Figs.3, 5, and S13, we do not see the "saw tooth" effect. Unfortunately, the data obtained in this campaign were mostly in 1 hour time resolution, thus we cannot evaluate how the model behaves at sub-hourly resolution between constraint points. We regard this as a critical issue that could materially affect modelled  $P(O_3)_{net}$  and will therefore be addressed explicitly in our next study.

-Approach to HONO – tucked away in the SI, the use of MARGA measurements of soluble nitrite for gas-phase HONO is mentioned. I appreciate that some approach was needed, but the sensitivity of the model results to this assumption is needed – eg what shift in P(O3) does a 20% change in HONO concs (or whatever is reasonable) result in. If this is a large shift – are the subsequent analysis all valid, i.e. can you be confident in pushing the model explorations so far? What is the time resolution of the MARGA data vs the observed temporal variation of e.g. NOx?

Thank you for the suggestion. According to Xu et al. (2019), a large number of two-channel WD/IC instruments represented by the Monitor for AeRosols and Gases in ambient Air (MARGA) instruments was widely used to obtain aerosol composition information, as well as acid trace gas levels, including HONO (Stieger et al., 2018). However, the application of HONO data was limited because of the measurement uncertainty. The measurement uncertainty of HONO database obtained by MARGA was evaluated by Xu et al. (2019) and Spindler et al. (2003). For this purpose, Xu et al. (2019) used a MARGA and more accurate equipment (LOPAP) to simultaneously measure the HONO concentration at the Station for Observing Regional Processes of the Earth System (SORPES) in the YRD of east China; Spindler et al. (2003) performed the laboratory and field experiments as well as direct kinetic laboratory studies to quantify an artefact by the aqueous phase formation of HONO from dissolved NO2 and SO2 at wetted denuder walls. In this study, we used the method proposed by Spindler et al. (2003) to check the measurement error of HONO by MARGA, and then checked its influence to the modelled P(O3)net. More details are added in the Supplementary Materials S2.

"However, previous studies have shown the HONO may be overestimated by MARGA due to aqueous phase formation of HONO from dissolved NO2 and SO2 at wetted denuder walls

(Stieger et al., 2018; Spindler et al. 2003). The measurement error of HONO by MARGA was evaluated by Xu et al. (2019) and Spindler et al. (2003). In this study, we used the method proposed by Spindler et al. (2003) to evaluate measurement uncertainty of HONO database obtained by MARGA, and then checked its influence to the modelled  $P(O_3)_{net}$ . The overall artefact formation measurement error of HONO by MARGA is expressed as a sum in Eq. (S5):

$$[HNO_2]_{art} = 0.056[NO_2] + (0.0032/ppb)[NO_2][SO_2]$$
 (S5)

where 0.0032 is the reciprocal value of the slope of the straight line between the HNO2 concentration corrected for the HNO2 content in purified air, the mean NO2 artefact and the concentration product of NO2 and SO2. We further modelled  $P(Ox)_{net}$  Case D4 with the corrected HONO, and found that the corrected HONO could decrease the modelled  $P(Ox)_{net}$  Case D4 by 0-8%, as shown in Fig. S18. Therefore, we note that with the measurement error of HONO by MARGA, the modelling method may consistently underestimate the modelled  $P(Ox)_{net}$  in all cases, and the  $P(Ox)_{net}$  missing in our study should be regarded as the lower limit values.

Figure S18: The modelled  $P(Ox)_{net}$  Case D4 and the with and without the HONO correction."

Accordingly, we added the related description in lines 334-336 of the modified manuscript:

"Due to the measurement error of HONO by MARGA in this study, the modelled  $P(O_3)_{net}$  tends to be underestimated (as shown in SM: S2); thus, we define the  $P(O_3)_{net}$  Missing obtained from all simulation cases as the upper-limit values."

**And lines 574-576 of the modified manuscript:**

"Systematic underestimation of modelled  $P(O_3)_{net}$  ( $P(O_3)_{net}$  Mod) was found when compared to the measured  $P(O_3)_{net}$  ( $P(O_3)_{net}$  Mea); this difference is defined as upper-limit  $P(O_3)_{net}$  Missing due to the overestimation of HONO by MARGA in this study."

The nitrogen oxides analyzer (Fengyue Aorui-1014, China) provided NOx data at a temporal resolution of 1 min, while the NO2- data from the MARGA were recorded at 1 h resolution. Therefore, all data were averaged to 1 h for model input, as stated in lines 163–165 of the main text.

"In addition to  $P(O_3)_{net}$  and OFS, hourly data such as  $PM_{2.5}$ ,  $O_3$ , NO,  $NO_2$ ,  $SO_2$ , carbon monoxide (CO), photolysis rates  $(j_O^1D, j_{NO_2}, j_{H_2O_2}, j_{NO_3\_M}, j_{NO_3\_R}, j_{HONO}, j_{HCHO\_M}, j_{HCHO\_R})$ , HONO, and VOCs concentrations were monitored (more details about the measurements are shown in Table S1)."

-Lots on the deposition velocities but how well do we know the boundary layer height BLH?

The planetary boundary layer height (PBLH) is the reanalysis data obtained from the website of the NOAA Air Resources Laboratory (https://ready.arl.noaa.gov/READYamet.php). The diurnal changes of PBLH on O3 pollution days and normal days are added in Fig. S4 in the Supplementary Materials. We have provided an additional description on the source of the PBLH data in lines 167-169 in the revised manuscript:

"The PBLH data used in the model here was obtained from the web portal of the Real-time Environmental Applications and Display sYstem (READY) of the National Oceanic and Atmospheric Administration (NOAA) Air Resource Laboratory (https://ready.arl.noaa.gov/READYamet.php)."

L172 – is the HO2 uptake process an irreversible loss (in the model)?

Yes, we used the data obtained from Zhou et al. (2020, 2023) measured from the ambient air and assumed it is an irreversible loss in the model.

L174 – is the  $N_2O_5$  uptake an irreversible loss – if I follow some scenarios included recycling via  $CINO_2$  – not quite clear how the condensed phase processed were simulated?

Previous study has found that the absorption of  $N_2O_5$  on aerosols surfaces containing chloride ions leads to the formation of CINO2 (Finlayson-Pitts et al., 1989). This  $N_2O_5$  uptake process is an irreversible loss: it converts  $N_2O_5$  into stable soluble nitrate and potentially volatile CINO2, thereby permanently removing gaseous  $N_2O_5$ . CINO2 will be photolyzed into CI- and NO2 under sunlight (Chen et al., 2023; Ma et al., 2023; McNamara et al., 2020; Peng et al., 2021). The detailed reactions are as follows:

$$N_2O_5 + Cl^-(aq) \to \varphi ClNO_2 + (2 - \varphi)NO_3^-(aq)$$

$$ClNO_2 \stackrel{k_2}{\to} Cl \cdot + NO_2$$

$$k_2 = j_{ClNO_2}$$

Where  $\varphi$  is the production yield of  $ClNO_2$ ,  $k_2$  is the photolysis rate of ClNO2  $(j_{ClNO_2})$ . These details are shown in Supplementary Materials S3.

However, the current chemical mechanism, MCM v3.3.1, lacks these reaction processes, so we added the mechanism of N2O5 uptake by aerosols and photolysis of ClNO2 to MCM v3.3.1 to simulate ClNO2 and explored its impact on  $P(O_3)_{net}$ . We set the heterogeneous uptake coefficient of N2O5 as 0.02 ( $\gamma_{N_2O_5} = 0.02$ ), and the production yield of  $ClNO_2$  as 0.6 ( $\varphi = 0.6$ ). Here the ClNO2 was derived from model simulations and its related reactions are added into MCM v3.3.1 as:

"% ASA\*24175.8\*0.02/4:N2O5 = 0.6 CINO2+1.4 PNO3;"

"%J<45>: CINO2 = CI +NO2;"

Where ASA represent the surface area of the ambient aerosols, 24175.8 represent the mean molecular speed of  $N_2O_5$  (cm s-1), 0.6 represent the production yield of ClNO2, J<45> represent the photolysis rate of ClNO2. As our previous study has thoroughly described these mechanisms (Zhou et al., 2024a), we have modified the description in lines 199-200 of the modified manuscript:

"..., and Case  $D_1$  extends Case C by adding the  $N_2O_5$  uptake mechanism and Cl· related photochemical reactions. Detailed simulation parameter settings can be found in our previous study (Zhou et al., 2024a) and the supplementary information (Table S3)."

"Table S3. Description of different modelling scenarios and the parameter settings

| Case           | Description                                                                                                                                                        | Parameter settings                                                                                                                                                                                                     | references                                                                              |
|----------------|--------------------------------------------------------------------------------------------------------------------------------------------------------------------|------------------------------------------------------------------------------------------------------------------------------------------------------------------------------------------------------------------------|-----------------------------------------------------------------------------------------|
| A              | Ambient gases (NO, NO 2 , SO 2 , CO, O 3 ), HONO, 44 VOCs, meteorological parameters ( T , RH, P , BLH), photolysis | O 3 (0.27 cm s -1 )                                                                                                                                                                              | (Xue et al., 2014)                                                                      |
| В              | rates, and $O_3$ dry deposition
Case A with the addition of $HO_2$ uptake                                                                                       | γ HO2 =0.19                                                                                                                                                                                                 | (Zhu et al., 2020;
Zhou et al., 2021)                                                |
| С              | Case B with the addition of trace gases (NO 2 , SO 2 , H 2 O 2 , HNO 3 , PAN, HCHO) dry deposition          | $NO_2 \ (0.6 \text{ cm s}^{-1})$
$SO_2 \ (0.8 \text{ cm s}^{-1})$
$H_2O_2 \ (1.2 \text{ cm s}^{-1})$
$HNO_3 \ (4.7 \text{ cm s}^{-1})$
$PAN \ (0.4 \text{ cm s}^{-1})$
$HCHO \ (0.9 \text{ cm s}^{-1})$ | (Zhang et al.,
2003; Xue et al.,
2014)                                            |
| $D_1$          | Case C with the addition of $N_2O_5$ non-homogeneous absorption reactions and $CINO_2$ photolysis                                                                  | $\gamma_{\text{N2O5}}=0.02$ $\varphi_{\text{CINO}_2}=0.6$                                                                                                                                                              | (Xue et al., 2014;
Badger et al.,
2006; Xia et al.,
2019; Xia et al.,
2020) |
| $D_2$          | Case D 1 with increased constraints for acetaldehyde, acrolein, acetone, and                                                                            |                                                                                                                                                                                                                        |                                                                                         |
| $D_3$          | butanone Case D 1 with increased constraints for all measurable OVOCs                                                                                   | Constraints based on measurement data                                                                                                                                                                                  |                                                                                         |
| $D_4$          | Case D 3 with increased constraints for all measurable chlorinated VOCs                                                                                 |                                                                                                                                                                                                                        |                                                                                         |
| $E_1$          | Case D 1 with overall VOCs concentration in constraints increased                                                                                       |                                                                                                                                                                                                                        |                                                                                         |
| $E_2$          | Case $D_1$ with increased concentrations of ethylene and formaldehyde in constraints                                                                               | Increase based on the correlation between $P(O_3)_{\text{net}}$ Missing and $k_{\text{OH}}$ Missing                                                                                                                    |                                                                                         |
| E 3 | Case D 1 with increased formaldehyde concentration in constraints                                                                                       | NOn_IMISSING                                                                                                                                                                                                           |                                                                                         |

Notes: Parameter values for modelling scenarios from Case A to Case  $D_1$  are set the same as those in Zhou et al. (2024a)."

**Minor Points**

Please define OFS where first used (abstract, L61)

Thank you for your suggestion. We have added the definition of O₃ formation sensitivity (OFS) in lines 44-49 in the modified manuscript:

"The sensitivity of O3 formation to its precursors is defined as the O3 formation sensitivity (OFS), which can be classified into three regimes: NOx-limited, VOC-limited, or mixed sensitivity (Seinfeld & Pandis, 2016; Sillman, 1999). In an NOx-limited regime, the VOC/NOx ratio is high and O3 production is controlled primarily by changes in NOx. In a VOC-limited regime, the VOC/NOx ratio is low, so O3 decreases with additional NOx and increases with higher VOCs. In the mixed-sensitivity regime, O3 rises when either NOx or VOC emissions increase (Wang et al., 2019)."

Accordingly, we used OFS directly in lines 65 in the modified manuscript:

"These gaps lead to systematic biases in the simulated  $P(O_3)_{net}$  (Woodward-Massey et al., 2023; Tan et al., 2017; Tan et al., 2019), thereby affecting the accurate determination of OFS."

The related references are added in the references list:

"Seinfeld, J. H. and Pandis, S. N. (Eds. 3): Atmospheric Chemistry and Physics: From Air Pollution to Climate Change, John Wiley & Sons, Hoboken, ISBN 978-1-118-94740-1, 2016.

Sillman, S.: The relation between ozone, NOx and hydrocarbons in urban and polluted rural environments, Atmospheric Environment, 33, 1821–1845, https://doi.org/10.1016/S1352-2310(98)00345-8, 1999.

Wang, P., Chen, Y., Hu, J., Zhang, H., and Ying, Q.: Attribution of tropospheric ozone to NOx and VOC emissions: considering ozone formation in the transition regime, Environmental Science & Technology, 53, 1404–1412, https://doi.org/10.1021/acs.est.8b05981, 2019."

Various places – ppb etc are mixing ratios not concentrations

We changed "concentrations" to "mixing ratios" when describing "ppb" throughout the manuscript.

L165 het loss can be important for HO2 removal globally – but wont HO2 + NO dominate under the conditions of these BL measurements?

Yes, HO2+NO dominates under our field measurement conditions, as demonstrated in Fig. S8. However, this dominance refers only to the HO2 removal rate. For O3 production, HO2+NO is a positive source as it

simultaneously produces  $NO_2$ , while  $HO_2$  uptake act as a negative source, similar to other termination reactions (e.g.,  $HO_2+HO_2$ ,  $HO_2+RO_2$ ). Therefore, in terms of the negative impact on  $P(O_3)_{net}$ , the heterogeneous reaction of  $HO_2$  with ambient aerosols is more important than  $HO_2+NO$ .

Therefore, we modified the sentence "Although the heterogeneous uptake of HO2 is not the dominant loss pathway of HO2, it accounts for approximately 10–40 % of global HO2 loss (Li et al., 2019); as a termination reaction, its direct negative impact on photochemical O3 production is non-negligible." in lines 189-191 in the modified manuscript.

**References:**

Berndt, T., Mentler, B., Scholz, W., Fischer, L., Herrmann, H., Kulmala, M., Hansel A.: 2018. Accretion product formation from Ozonolysis and OH radical reaction of  $\alpha$ -Pinene: mechanistic insight and the influence of isoprene and ethylene, Environmental Science & Technology, 52, 11069–11077, doi: 10.1021/acs.est.8b02210, 2018.

Carter, W. P. L., A. Pierce J. A., Luo, D., Malkina, I. L.: Environmental chamber study of maximum incremental reactivities of volatile organic-compounds, Atmospheric Environment, 29, 2499, doi.org/10.1016/1352-2310(95)00149-S, 1995.

Chen, X., Xia, M., Wang, W., Yun, H., Yue, D., and Wang, T.: Fast near-surface CINO2 production and its impact on O3 formation during a heavy pollution event in South China, Science of The Total Environment, 858, 159998, doi: 10.1016/j.scitotenv.2022.159998, 2023.

Crounse, J. D., Knap, H. C., Ørnsø, K. B., Jørgensen, S., Paulot, F, Kjaergaard, H. G., and Wennberg, P. O.: Atmospheric Fate of Methacrolein. 1. Peroxy Radical Isomerization Following Addition of OH and O2. The Journal of Physical Chemistry A, 116, 5756-5762, doi.org/10.1021/jp211560u, 2012.

Finlayson-Pitts, BJ., Ezell, MJ., and Pitts, JN.: Formation of chemically active chlorine compounds by reactions of atmospheric NaCl particles with gaseous N2O5 and CIONO2, Nature, 337, 241-244, doi:10.1038/337241a0, 1989.

Hao, Y., Zhou, J., Zhou, J. P., Wang, Y., Yang, S., Huangfu, Y., Li, X. B., Zhang, C., Liu, A., Wu, Y., Zhou, Y., Yang, S., Peng, Y., Qi, J., He, X., Song, X., Chen, Y., Yuan, B., and Shao, M.: Measuring and modeling investigation of the net photochemical ozone production rate via an improved dual-channel reaction chamber technique, Atmospheric Chemistry and Physics, 23, 9891-9910, 10.5194/acp-23-9891-2023, 2023.

Ma, W., Feng, Z., Zhan, J., Liu, Y., Liu, P., Liu, C., Ma, Q., Yang, K., Wang, Y., He, H., Kulmala, M., Mu, Y., and Liu, J.: Influence of photochemical loss of volatile organic compounds on understanding ozone formation mechanism, Atmospheric Chemistry and Physics, 22, 4841–4851, doi:10.5194/acp.22.4841, 2022.

McNamara, SM., Kolesar, KR., Wang, S., Kirpes, RM., May, NW., Gunsch, MJ., Cook, RD., Fuentes, JD., Hornbrook, RS., Apel, EC., China, S., Laskin, A., Pratt, and Pratt KA.: Observation of road salt aerosol driving inland wintertime atmospheric chlorine chemistry, ACS Central Science, 6, 684-694, doi:10.1021/acscentsci.9b00994, 2020.

Peng, X., Wang, W., Xia, M., Chen, H., Ravishankara, AR., Li, Q., Saiz-Lopez, A., Liu, P., Zhang, F., Zhang, C., Xue, L., Wang, X., George, C., Wang, J., Mu, Y., Chen, J., and Wang, T.: An unexpected large continental source of reactive bromine and chlorine with significant impact on wintertime air quality, National Science Review, 8, 99-109, doi:10.1093/nsr/nwaa304, 2021.

Spindler, G., Hesper, J., Brüggemann, E., Dubois, R., Müller, T., and Herrmann, H.: Wet annular denuder measurements of nitrous acid: laboratory study of the artefact reaction of  $NO_2$  with S(IV) in aqueous solution and comparison with field measurements: Atmospheric Environment, 37, 2643, doi:10.1016/S1352.2310(03)00209.7, 2003.

Stieger, B., Spindler, G., Fahlbusch, B., Müller, K., Grüner, A., Poulain, L., Thöni, L., Seitler, E., Wallasch, M., and Herrmann, H.: Measurements of PM10 ions and trace gases with the online system MARGA at the research station Melpitz in Germany – A five-year study, Journal of Atmospheric Chemistry, 75, 33–70, doi.org/10.1007/s10874-017-9361-0, 2018.

Wang, S., Wu, R., Berndt, T., Ehn, M., and Wang, L.: Formation of highly oxidized radicals and multifunctional products from the atmospheric oxidation of Alkylbenzenes, Environmental Science & Technology, 51, 8442-8449, doi:10.1021/acs.est.7b02374, 2017.

Wu, S., Lee, H. J., Anderson, A., Liu, S., Kuwayama, T., Seinfeld, J. H., and Kleeman, M. J.: Direct measurements of ozone response to emissions perturbations in California, Atmospheric Chemistry and Physics, 22, 4929-4949, oi.org/10.5194/acp-22-4929-2022, 2022.

Xu, Z., Liu, Y., Nie, W., Sun, P., Chi, X., and Ding, A.: Evaluating the measurement interference of wet rotating-denuder – ion chromatography in measuring atmospheric HONO in a highly polluted area: Atmospheric Measurement Techniques, 12, 6737 – 6748, doi:/10.5194/amt.12.6737, 2019.

Zhou, J., Wang, W., Wang, Y., Zhou, Z., Lv, X., Zhong, M., Zhong, B., Deng, M., Jiang, B., and Luo, J.: Intercomparison of measured and modelled photochemical ozone production rates:

Suggestion of chemistry hypothesis regarding unmeasured VOCs, Science of The Total Environment, 951, 175290, doi:10.1016/j.scitotenv.2024.175290, 2024a.

Zhou, J., Wang, W., Wang, Y., Zhou, Z., Lv, X., Zhong, M., Zhong, B., Deng, M., Jiang, B., Luo, J., Cai, J., Li, X.-B., Yuan, B., and Shao, M.: Intercomparison of measured and modelled photochemical ozone production rates: Suggestion of chemistry hypothesis regarding unmeasured VOCs, Science of The Total Environment, 951, 175290, doi:/10.1016/j.scitotenv.2024.175290, 2024b.

---

## Author Response (AR1)

We thank the referees for their insightful questions and comments, which helped to improve the quality of this paper. Our answers to all the concerns are listed in the following in red, after the reviewer's comments, which are in black. The changed parts in the modified manuscript are marked in yellow.

**Anonymous Referee #1**

This manuscript presents a comprehensive investigation of net ozone production rate ($P(O_3)_{net}$) and ozone formation sensitivity (OFS) through the integration of in situ field observations using a novel dual-channel reaction chamber system (NPOPR) and detailed box model simulations based on MCM v3.3.1. The study is of high relevance and scientific value, particularly in addressing long-standing issues of underestimation in modeled ozone production. The work also has practical implications for improving model-based OFS diagnosis and VOC pollution control strategies. However, several major issues must be addressed in the manuscript.

In your study, observed OVOC concentrations are used to constrain the box model. However, many OVOCs (e.g., formaldehyde, acetaldehyde, ketones) are not only emitted directly but also formed via secondary photochemical reactions from VOC precursors. Directly constraining their concentrations may mask deficiencies in the model's chemical mechanism and artificially suppress diagnostic signals of missing secondary formation pathways.

We fully appreciate your concern and recognize the potential issues that may arise from directly constraining OVOC concentrations. To investigate whether the model can reproduce secondary formation on its own, we added a modelling scenario without OVOC constraints based on Case D1 and output key OVOC species (as shown in Fig. S19). Results show that, Without additional constraints, the model overestimates some OVOCs (e.g., HCHO and $CH_3CHO$) yet adequately reproduces their secondary-formation pathways; in contrast, the observed diurnal cycle of $CH_3COCH_3$ shows no evident signature of secondary production. These results demonstrate that directly constraining the OVOC concentrations may mask deficiencies in the model's chemcial mechanism and artificially suppress diagnostic signals of missing secondary formation pathways. However, refraining from any constraint would also falsely amplify the primary-source signal, especially those lacking clear secondary-generation signatures. Applying a constraint can better capture the influence of primary OVOCs. Furthermore, our analysis indicates that the $P(O_3)_{net}$ missing is not likely caused by unaccounted secondary production (see Sect. 3.3). Until such mechanistic gaps are resolved, observational nudging of OVOCs remains a pragmatic compromise: it preserves

concentration accuracy while curbing spurious chemical feedbacks. We have added such kind of discussion in lines 401-406 of the modified manscuript:

"The negligible (or even negative) change in $P(O_3)_{net\_}$Mod when OVOCs are constrained in Cases D1-D4 may arise because the OVOC constraint masks deficiencies in the model's chemcial mechanism and artificially suppresses diagnostic signals of missing secondary formation pathways. Until the underlying chemical mechanisms are improved, observational nudging of OVOCs offers a practical compromise—it helps maintain concentration accuracy while limiting unrealistic chemical feedbacks (more details can be found in Supplementary Materials S5)."

And S5 in the Supplementary Materials:

"**S5. Impacts of OVOCs constraints in the model**

 To explore the impact of OVOCs constraint in the model, we further added a modelling scenario without OVOC constraints based on Case D1 and output key OVOC species (see Fig. S19). From Fig. S19, the model tends to overestimate some OVOC concentrations (i.e., HCHO, $CH_3CHO$), and their secondary-formation pathways are adequately captured, while the observed diurnal variation of $CH_3COCH_3$ does not exhibit clear secondary formation characteristics. These results show that directly constraining OVOC concentrations can fill the concentration gap in the model to match observed OVOC levels, but may mask deficiencies in the model's chemical mechanism and artificially suppress diagnostic signals of missing secondary formation pathways (i.e., the $RO_2$-to-OVOC reaction pathways). This will lead to the underestimation of the entire HOx-cycle oxidation rate, lowers the budgets of OH, $O_3$, and $NO_3$, and subsequently the $P(O_3)_{net\_}$Mod. However, without any constraint, the model may overestimate the contribution from primary sources. Furthermore, our analysis indicates that the $P(O_3)_{net}$ missing is not likely caused by unaccounted secondary production (see Sect. 3.3). Until such mechanistic gaps are resolved, observational nudging of OVOCs remains a pragmatic compromise: it preserves concentration accuracy while curbing spurious chemical feedbacks.

[Figure]

**Figure S19: Comparison of measured OVOCs with modeled values from a no-constraint OVOC scenario based on Case D1.**"

Why were NO and NOx changed between the two methods to diagnose $O_3$ sensitivity, respectively?

We measured ozone ($O_3$) formation sensitivity (IR index) directly using the NPOPR instrument, with NO as the $NO_X$ indicator. This approach aligns with previous studies(Sklaveniti et al., 2018; Morino et al., 2023), which also used NO in their $P(O_3)_{net}$ measurement systems. By adding NO instead of $NO_X$, the CAPS-$NO_2$ instrument could eliminate $NO_2$ interference and directly quantify NO's contribution to $O_3$ production. When $O_3$ formation sensitivity is in the $NO_X$-limited regime, increasing NO enhances $P(O_3)_{net}$ because $HO_2/RO_2$ reacts more efficiently with NO to produce $NO_2$. Conversely, in the VOCs-limited regime, adding NO may reduce $P(O_3)_{net}$ because NO consumes OH to form HONO, weakening VOC oxidation chain reactions.

Our model's $O_3$ formation sensitivity (OFS) analysis is based on the absolute $P(O_3)_{net}$ sensitivity calculation, which directly evaluates how changes in $NO_X$ (NO + $NO_2$) affect $P(O_3)_{net}$ while considering both the NO titration effect and the role of $NO_2$ photolysis in $O_3$ production.

The manuscript attributes the model – measurement discrepancy ($P(O_3)_{net}$_Missing) entirely to missing reactive VOCs or underrepresented chemical pathways. However, box models by design do not account for horizontal or vertical

transport, which may play a significant role in shaping the measured ozone production rate—especially during periods with strong advection or mixing layer evolution, such as early morning or late afternoon. You should clarify why transport processes are neglected and whether their influence is truly negligible.

We acknowledge that transport processes may influence $O_3$ distribution under specific conditions. However, this study primarily focuses on local photochemical $O_3$ production. By employing the NPOPR detection system with an extremely short residence time (0.15 h), we aimed to capture instantaneous in-situ photochemical reactions rather than $O_3$ accumulation effects over time and space. The modeling incorporated real-time meteorological conditions and pollutant concentration data. On such short timescales, the impacts of vertical mixing and horizontal advection become relatively minor. Moreover, this study compares measured and simulated net $O_3$ production rates ($P(O_3)_{net}$) rather than $O_3$ concentrations themselves. Our previous study also demonstrated that $P(O_3)_{net}$ more directly reflects the photochemical $O_3$ formation potential from local precursors and is less affected by transport processes compared to $O_3$ concentrations (Zhou et al., 2024b).

We added "The time resolution of the $P(O_3)_{net}$ measurement is 4 min. Our previous study demonstrated that $P(O_3)_{net}$ more directly reflects the photochemical $O_3$ formation potential from local precursors and is less affected by transport processes compared to $O_3$ concentrations (Zhou et al., 2024b)." in line 129-131 of the modified manuscript.

Although the manuscript includes substantial observation–model comparisons and compensatory mechanisms for missing reactivity, the concluding section does not clearly state what is new in this work compared to existing studies, please clearly emphasize the innovation points and boundaries of this study in the conclusion section and explain its promoting role in the research of the formation mechanism of ozone pollution.

We appreciate the reviewer's valuable suggestion to clarify the novel contributions of our study. The main innovation of this work is that we have successfully applied the NPOPR detection system to directly measure OFS in field observations. This approach enables us to quantitatively assess the impact of $P(O_3)_{net}$ simulation deficits on OFS determination. Previous studies in this field predominantly relied on model simulations without sufficient observational validation. While our earlier research (Zhou et al., 2024a) qualitatively analyzed the effects of different VOC species on $P(O_3)_{net}$ through model-observation comparisons, quantitative analysis was still lacking. This study makes significant advancements by employing direct measurement methods to quantitatively investigate the contributions of different VOC species to $P(O_3)_{net}$

simulation deficits. Our findings address the critical scientific issue of model underestimation of $P(O_3)_{net}$ and quantify its impact on OFS determination. These results provide both a robust database and theoretical foundation for improving the accuracy of model-based OFS assessment and developing more effective $O_3$ pollution control strategies.

We have changed the sentence "In conclusion, improving the model's accuracy requires further expansion of the measurement of VOC species, particularly OVOCs, and the incorporation of relevant chemical mechanisms into the model. In future studies, continuing field observations based on direct measurement of $P(O_3)_{net}$ and accumulating more data will contribute to a better understanding of $O_3$ pollution formation mechanisms and make effective $O_3$ pollution control strategies." to "In conclusion, we quantitatively assessed the $P(O_3)_{net}$ simulation deficits and their impact on OFS diagnosis by comparing the measured and modelled $P(O_3)_{net}$, and found that the unmeasured VOCs —rather than the secondary atmospheric formation — are the primary causative factor of $P(O_3)_{net}$_Missing. Furthermore, both direct measurements and model results reveal a diurnal OFS shift dominated by the morning regime; transition and VOC-limited conditions prevailed, so prioritizing VOCs while co-controlling $NO_X$ is the most effective approach to $O_3$ pollution control in PRD region. Our results also demonstrate that the persistent model biases risk under-estimating the local photochemical formation contribution to $O_3$ pollution, thereby has weakening its perceived impact relative to physical transportation. Future studies should expanded VOCs measurements and combine direct $P(O_3)_{net}$ observations with regional transport model to separate local production from up-wind advection." in lines 610-617 in the modified manuscript.

Lines 29-31: You mentioned "the only approach to fill the gag was to add unmeasured VOCs" appears too strong. It implies that no other explanations or methods could be relevant, which may not be justified. Consider softening this to reflect that unmeasured OVOCs were the most effective compensating factor in this study, rather than the only one possible.

Thanks for your suggestion. We have changed the sentence "The only approach to fill the gap between observation and computation was to add possible unmeasured reactive VOCs, ... " to "The results in this study reflected that unmeasured oxygenated VOCs (OVOCs) were the most effective compensating factor for the discrepancies between observed and computed $P(O_3)_{net}$ and OFS, ..." in lines 29-30 in the modified manuscript.

Line 43: "precursor" should be "precursors".

We changed "precursor" to "precursors" in line 43 of the modified manuscript.

Line 46: "equation" should be "equations".

We changed "equation" to "equations" in line 50 of the modified manuscript.

Lines 205-206: It is unclear what magnitude of precursor perturbation was applied, please provide a more explicit description of the model configuration used for the sensitivity analysis.

We thank the reviewer for this valuable comment. Regarding the calculation method for absolute $P(O_3)_{net}$ sensitivity in our model, we employed the analytical approach proposed by Sakamoto et al. (2019). This method does not require artificial perturbation of precursor concentrations in the simulations. Instead, it is defined as the change in $P(O_3)_{net}$ values caused by a percentage change in either [NO$_X$] or [VOCs] concentrations. To better clarify this methodology, we have revised the manuscript to explicitly state: "In this study, the analysis of absolute $P(O_3)_{net}$ sensitivity was conducted using the box model through an analytical calculation approach that does not involve artificial perturbation of precursor concentrations." in line 233 of the modified manuscript.

Lines 278-281: The confidence interval of 68.3% is relatively conservative, please provide additional analysis.

Regarding the confidence interval selection, we initially chose the 68.3% interval as it corresponds to the ±1$\sigma$ range in Gaussian distribution, which represents a standard statistical measure. However, as suggested by the reviewer, we have conducted additional sensitivity analysis using a 90% confidence interval. The results show similar trends in both cases, confirming the robustness of our findings.

"

[Figure]

**Figure 1: Correlation between measured $P(O_3)_{net}$ ($P(O_3)_{net}$_Mea) and (a) total OH reactivity ($k_{OH}$) and (b) O$_3$ Formation Potential (OFP). The shaded area in the figure represents the confidence interval (90 %) of the fitting line between $P(O_3)_{net}$ and $k_{OH}$, and between $P(O_3)_{net}$ and OFP."**

We have changed the sentence "The shaded area in the figure represents the confidence interval (68.3%) of the fitting line between $P(O_3)_{net}$ and $k_{OH}$, and between $P(O_3)_{net}$ and OFP." to "The shaded area in the figure represents the confidence interval (90 %) of the fitting line between $P(O_3)_{net}$ and $k_{OH}$, and between $P(O_3)_{net}$ and OFP." in line 309 in the modified manuscript.

Line 306: The reported p-value (P < 0.5) does not indicate statistical significance, and the analysis doesn't hold.

Thank you for catching this typographical error. We have carefully re-examined the statistical analysis comparing $P(O_3)_{net\_}$Missing between $O_3$ pollution days and normal days. The corrected results show a significant difference with $p = 0.03$ (< 0.05). We have changed the sentence"The median $P(O_3)_{net\_}$Missing values on $O_3$ pollution days were statistically higher than those on normal days (*t-test*, P<0.5), ..." to "The $P(O_3)_{net\_}$Missing values on $O_3$ pollution days were statistically higher than those on normal days (*t-test, p<0.05*), ..." in line 343 of the modified manuscript.

Lines 305-307:The statement that the mechanisms added in Case D1 "are not the main cause" of the bias may overstate the conclusion. The remaining discrepancy could still be partly due to uncertainties in those mechanisms, parameterization, site-specific variability and transportation etc. A more cautious wording would improve clarity and avoid giving a false sense of certainty.

Thanks for your suggestion. We have changed the sentence "The median $P(O_3)_{net\_}$Missing values on $O_3$ pollution days were statistically higher than those on normal days (*t-test*, P<0.5), indicating that the supplementary mechanisms explored in the model, as mentioned above, are not the main cause of the $P(O_3)_{net\_}$Missing." to "The $P(O_3)_{net\_}$Missing values on $O_3$ pollution days were statistically higher than those on normal days (*t-test, p<0.05*), suggesting that while the supplementary mechanisms explored in the model may contribute to some extent, they are unlikely to be the dominant cause of the $P(O_3)_{net\_}$Missing." in lines 343-344 in the modified manuscript.

Lines 328-329: The statement that "$P(O_3)_{net\_}$Missing increases significantly at higher $O_3$ precursor concentrations" (based on $r^2$ = 0.4-0.5) may overstate the strength of the relationship. A moderate correlation should not be equated with a strong or significant increase unless supported by statistical testing.

We appreciate this constructive comment regarding the interpretation of our correlation results. Our additional analyses, including both Pearson correlation and t-tests, and found that r=0.24 and 0.20 for TVOCs and NOx, respectively. And the correlations are very weak, with t=1.6 and 1.3 for VOCs and $NO_X$ (*t-tests*), respectively. These t-test values for correlation significants are lower than the critical value of t=2.0, confirming

that $P(O_3)_{net\_Missing}$ shows weak relationships with VOCs and NO$_X$ precursor concentrations.

We have changed the sentence "Under O$_3$ pollution days, $P(O_3)_{net\_Missing}$ showed a positive correlation with VOCs and NO$_X$, with r$^2$ values of 0.4 and 0.5, respectively, indicating that $P(O_3)_{net\_Missing}$ increases significantly at higher O$_3$ precursor concentrations. This phenomenon is consistent with previous studies (Whalley et al., 2021; Ren et al., 2013; Zhou et al., 2024a)." to "On O$_3$ pollution days, $P(O_3)_{net\_Missing}$ exhibited a moderate positive correlation with VOCs (r$^2$ =0.4, R=0.2, t=2.9) and NO$_X$ (r$^2$=0.5, R=0.2, t=3.8), confirming that the $P(O_3)_{net\_Missing}$ is larger at higher precursor concentrations/mixing ratios (both t > critical 2.0, $p < 0.05$), consistent with earlier box-model studies (Whalley et al., 2021; Ren et al., 2013; Zhou et al., 2024a). A moderate positive correlation is also found with $J_{O1D}$ on both O$_3$ pollution days and normal days, with r$^2$ values of 0.5 and 0.4, respectively. On normal days all correlations collapse (r$^2 < 0.2$, $p > 0.1$), implying that the model deficit is not tied to the measured precursors under low-NO$_x$ conditions and may instead related to the missing mechanisms for unmeasured photolabile VOCs." in lines 368-375 in the modified manuscript.

Lines 389-391: You use an empirical relationship between kOH and P(O$_3$)net to get kOH_Missing, and then adjust VOC concentrations to match this value. However, this method assumes a direct linear relationship without showing how real chemical reactions support this assumption. Please explain why this approach is reasonable, and whether it reflects actual atmospheric chemistry.

The method of estimating missing VOC concentrations through the empirical linear relationship between OH reactivity ($k_{OH}$) and $P(O_3)_{net}$ is used in this study, fundamentally based on the OH-driven nature of O$_3$ production. The scientific basis lies in the fact that $P(O_3)_{net}$ is closely related to the production rate of RO$_X$ radicals ($P(RO_X)$), which are primarily formed through the reaction of OH with VOCs. Since $P(RO_X)$ is directly influenced by the OH reactivity ($k_{OH}$), $P(O_3)_{net}$ is consequently correlated with $k_{OH}$. Furthermore, previous study have shown that $P(O_3)_{net}$ exhibits a linear relationship with both $P(HOx)$ and $k_{OH}$ when O$_3$ formation is located in VOCs-limited regime (Baier et al., 2017), and this approach reflects nearly actual atmospheric chemistry if $P(O_3)_{net}$ missng is driven by VOCs reactivity missing (Wang et al., 2024).

To explore the influence of unconstrained secondary products to $P(O_3)_{net}$ missing (which may influence the linear relationship between $P(O_3)_{net}$ missing and $k_{OH}$), we checked the dependence of $P(O_3)_{net}$ missing on the ethylbenzene/m,p-xylene ratio. As the m,p-xylene has a larger reaction rate constant ($18.9 \times 10^{-12}$ cm$^3$ molecule$^{-1}$ s$^{-1}$) than ethylbenzene ($7.0 \times 10^{-12}$ cm$^3$

molecule$^{-1}$ s$^{-1}$) when reacting with OH radicals, the ratio of ethylbenzene to m,p-xylene was used to characterize the degree of air mass aging (de Gouw et al., 2005; Yuan et al., 2013), a higher ratio of ethylbenzene to m,p-xylene corresponds to a higher degree of air mass aging. We see that the $P(O_3)_{net}$ missing decreases with the increasing ratio of ethylbenzene to m,p-xylene (as added in Fig. S11f), which indicates that the $P(O_3)_{net}$ missing was not caused by unconstrained secondary products.

To make the description clearer, we have changed the sentence "We hypothesize that the remaining $P(O_3)_{net\_Missing}$ is caused by unknown VOCs that are not constrained in the box model. By quantifying the relationship between $k_{OH}$ and $P(O_3)_{net}$..." to "We hypothesize that the remaining $P(O_3)_{net\_Missing}$ is caused by unknown VOCs that are not constrained in the box model. The method of estimating missing VOC concentrations through the empirical linear relationship between OH reactivity ($k_{OH}$) and $P(O_3)_{net}$ is used in this study, the scientific basis lies in the fact that $P(O_3)_{net}$ is closely related to the production rate of $RO_X$ radicals ($P(RO_X)$), which are primarily formed through the reaction of OH with VOCs. Since $P(RO_X)$ is directly influenced by the OH reactivity ($k_{OH}$), $P(O_3)_{net}$ is consequently correlated with $k_{OH}$. Previous study have shown that $P(O_3)_{net}$ exhibits a linear relationship with both $P(HOx)$ and $k_{OH}$ when $O_3$ formation is located in VOCs-limited regime (Baier et al., 2017), and this approach reflects nearly actual atmospheric chemistry if $P(O_3)_{net}$ missing is driven by VOCs reactivity missing (Wang et al., 2024b). Furthermore, we examined whether unconstrained secondary products affect $P(O_3)_{net}$ missing —and thus the linear relationship between $P(O_3)_{net}$ missing and $k_{OH}$— by analysing its dependence on the ethylbenzene / m,p-xylene ratio. Because this ratio increases with the degree of air-mass aging (de Gouw et al., 2005; Yuan et al., 2013), the observed decrease in the $P(O_3)_{net}$ missing with increasing ratio (Fig. S11f) indicates that the $P(O_3)_{net}$ missing is not caused by unaccounted secondary production." In lines 428-438 in the modified manuscript.

The corresponding references are also added in the reference list:

"de Gouw, J., Middlebrook, A., Warneke, C., Goldan, P., Kuster, W., Roberts, J., Fehsenfeld, F., Worsnop, D., Canagaratna, M., and Pszenny, A.: Budget of organic carbon in a polluted atmosphere: Results from the New England Air Quality Study in 2002, Journal of Geophysical Research-Atmospheres, 110, D16305, https://doi.org/10.1029/2004JD005623, 2005.

Yuan, B., Hu, W. W., Shao, M., Wang, M., Chen, W. T., Lu, S. H., Zeng, L. M., and Hu, M.: VOC emissions, evolutions and contributions to SOA formation at a receptor site in eastern China, Atmospheric Chemistry and Physics, 13, 8815–8832, https://doi.org/10.5194/acp-13-8815-2013, 2013."

The corresponding figure which plots the dependence of $P(O_3)_{net\_missing}$ on the ethylbenzene/m,p-Xylene ratio is added in Fig. S11f:

[Figure]

**Figure S11: Correlations between $P(O_3)_{net\_}$missing and TVOCs, NOx, $J_{O1D}$, $T$, Ox (a–e),** **and the ethylbenzene/m,p-Xylene ratio (f, representing the air mass aging).** **Circles represent O₃ pollution days, triangles represent normal days, and the shaded area indicates the 68.3 % confidence interval of the fitting line.**

Lines 396-397: In Case E2, ethylene was amplified to 5.9-85.6 times the original concentration, far exceeding the limit emission levels in the conventional urban atmosphere. The lack of emission inventories or observational data support may cause the simulation results to deviate from reality. More discussion on the rationality of these magnifications is required.

The purpose of these scenarios is to demonstrate the potential impact of this type of VOCs on $P(O_3)_{net\_}$Missing. We have changed the sentence "...This value was then used to compensate for the unmeasured VOCs in the model (with a daytime $k_{OH}$ compensation range of 1.2–2.4 s$^{-1}$). Based on the significant contribution of OVOCs to $P(O_3)_{net\_}$Missing mentioned earlier, we designed three modelling scenarios to compensate for $k_{OH\_}$ Missing, with the specific multiples varying each day: (1) Case E₁: by expanding the constrained overall VOCs concentrations in

Case $D_1$, the daily TVOC concentration was increased by 1.1 to 1.7 times; (2) Case $E_2$: according to $k_{OH}$ ratio of NMHC to OVOCs in the constrained VOCs of Case $D_1$, the concentrations of ethylene (a representative NMHC species) and formaldehyde (OVOCs indicator) were expanded separately. The ethylene concentration was increased by 5.9 to 85.6 times, and the formaldehyde concentration was increased by 1.4 to 2.0 times; (3) Case $E_3$: by expanding only the formaldehyde concentration to compensate for $k_{OH}$_ Missing, in this case, the daily formaldehyde concentration was increased by 1.8 to 9.2 times, to verify the role of OVOCs in compensating for $P(O_3)_{net}$_Missing" to "...This value was then used to compensate for the unmeasured VOCs in the model (with a daytime $k_{OH}$ compensation range of 1.2–2.4 s$^{-1}$, approximately 27.6–45.1% of missing values). Based on the significant contribution of OVOCs to $P(O_3)_{net}$_Missing mentioned earlier, we designed three modelling scenarios to compensate for $k_{OH}$_Missing, with the specific multiples varying each day. We note that these scenarios are idealized sensitivity tests to explore potential bounds of OVOCs' contribution to $P(O_3)_{net}$_Missing compensation, rather than realistic emission assumptions. Specifically, we tested how much the $P(O_3)_{net}$_Missing could be accounted for if the $k_{OH}$ were attributed to different VOCs categories. The specific scenarios include: (1) Case $E_1$: by expanding the constrained overall VOCs concentrations in Case $D_1$ (daily mean compensation range for TVOCs: 0.5–2.8 µg m$^{-3}$), the daily TVOC concentration was increased by 1.1 to 1.7 times; (2) Case $E_2$: according to $k_{OH}$ ratio of NMHC to OVOCs in the constrained VOCs of Case $D_1$, the concentrations of ethylene (a representative NMHC species) and formaldehyde (OVOCs indicator) were expanded separately. The ethylene concentration (daily mean compensation range for TVOCs: 0.5– 2.8 µg m$^{-3}$) was increased by 5.9 to 85.6 times, and the formaldehyde concentration (daily mean compensation range for TVOCs: 0.0–0.5 µg m$^{-3}$) was increased by 1.4 to 2.0 times; (3) Case $E_3$: by expanding only the formaldehyde concentration to compensate for $k_{OH}$_Missing, in this case, the daily formaldehyde concentration (daily mean compensation range for TVOCs: 0.6–1.4 µg m$^{-3}$) was increased by 1.8 to 9.2 times, to verify the role of OVOCs in compensating for $P(O_3)_{net}$_Missing." in lines 445-459 in the modified manuscript.

Line 485: It is not necessary to add legends to every subgraph in Fig. 6. Simplification can be considered.

Okay, we have removed the redundant legends in Fig. 6. See lines 517–520 in the modified manuscript:

"

[Figure]

**Figure 6: Average values of IR derived from the direct measurement data using the NPOPR detection system (e.g., $\Delta P(O_3)_{net}^{+NO}$ and $\Delta P(O_3)_{net}^{+VOCs}$ ) and absolute $P(O_3)_{net}$ sensitivity from the box model during (a)–(b) $P(O_3)_{net}$ rising phase (8:00-9:00) ; (c)–(d) $P(O_3)_{net}$ stable phase (10:00-12:00) (e)–(f) $P(O_3)_{net}$ declining phase (13:00-17:00).**"

Lines 519-521: The conclusion repeatedly emphasizes the role of OVOCs in $O_3$ formation and compensation, yet it assumes these are mainly anthropogenic in origin. As noted earlier, many OVOCs are also formed secondarily. It is recommended that you should distinguish between primary and secondary OVOC contributions or clearly state the limitation of their current attribution.

Thank you for your suggestion. We agree with the reviewer that it is not appririate to say "These species are mainly emitted by anthropogenic emissions", as the OVOCs are also formed secondarily from both anthropogenic and natural emissions. Therefore, we have deleted this sentence and added the related discussion in lines 289-293 in the modified manuscript:

"As OVOCs arise from both direct (anthropogenic and natural) emissions and secondary atmospheric formation (Lyu et al., 2024; Yuan et al., 2012), precluding a direct quantification of their respective contributions to $O_3$ formation. Nevertheless, our previous work showed that

anthropogenic primary VOCs correlate most closely with instantaneous $P(O_3)_{net}$ on $O_3$ pollution days, and urban anthropogenic OVOC emissions markedly enhance both oxidative capacity and $O_3$ production (Qian et al., 2025; Wang et al., 2024b)."

Appendix:

(1) We detected an error in Fig. 3b, therefore, we changed the diurnal variations of $P(O_3)_{net}$_Mea and $P(O_3)_{net}$_Mod (Case A–$D_1$) to diurnal variations of $P(O_3)_{net}$_Mea and $P(O_3)_{net}$_Mod (Case $D_1$–$E_3$) in Fig. 3b.

[Figure]

(2) The calculation method of the absolute $P(O_3)_{net}$ is adapted from the logarithmic derivative approach of Sakamoto et al. (2019). Therefore, We have changed the sentence "We calculated the modelled OFS using the absolute $P(O_3)_{net}$ sensitivity method from Sakamoto et al. (2019). It is defined as the change in $P(O_3)_{net}$ induced by a percentage increase in $O_3$ precursors. This method facilitates the quantitative assessment of how reductions in $O_3$ precursors contribute to the overall reduction of $P(O_3)_{net}$ over a period or within a region. The formula is as follows:

$$\text{Absolute } P(O_3)_{net} = \frac{\delta P(O_3)}{\delta \ln[X]} = P(O_3)\frac{\delta P(O_3)}{\delta \ln[X]} \tag{10}$$" to

"We calculated the modelled OFS using the absolute $P(O_3)_{net}$ sensitivity adapted from the logarithmic derivative approach of Sakamoto et al. (2019). It is defined as the change in $P(O_3)_{net}$ for the natural logarithm of $O_3$ precursor concentrations. This method facilitates the quantitative assessment of how reductions in $O_3$ precursors contribute to the overall reduction of $P(O_3)_{net}$ over a period. The formula is as follows:

$$\text{Absolute } P(O_3)_{net} = \frac{dP(O_3)_{net}}{d \ln[X]} \tag{10}$$"

in lines 224-228 in the modified manuscript.

**References**

Baier, B. C., Brune, W. H., Miller, D. O., Blake, D., Long, R., Wisthaler, A., Cantrell, C., Fried, A., Heikes, B., and Brown, S.: Higher measured than modeled ozone production at increased $NO_X$ levels in the Colorado Front Range, Atmospheric Chemistry and Physics, 17, 11273-11292,http://doi.org/10.5194/acp-17-11273-2017, 2017.

de Gouw, J. A., Warneke, C., Parrish, D. D., Holloway, J. S., Trainer, M., and Fehsenfeld, F. C.: Emission sources and ocean uptake of acetonitrile ($CH_3CN$) in the atmosphere, Journal of Geophysical Research-Atmospheres, 108, 4329, https://doi.org/10.1029/2002JD002897, 2003.

de Gouw, J., Middlebrook, A., Warneke, C., Goldan, P., Kuster, W., Roberts, J., Fehsenfeld, F., Worsnop, D., Canagaratna, M., and Pszenny, A.: Budget of organic carbon in a polluted atmosphere: Results from the New England Air Quality Study in 2002, Journal of Geophysical Research-Atmospheres, 110, D16305, https://doi.org/10.1029/2004JD005623, 2005.

Lyu, X., Li, H., Lee, S.-C., Xiong, E., Guo, H., Wang, T., and de Gouw, J.: Significant Biogenic Source of Oxygenated Volatile Organic Compounds and the Impacts on Photochemistry at a Regional Background Site in South China, Environmental Science & Technology, 58, 20081-20090,http://doi.org/10.1021/acs.est.4c05656, 2024.

Morino, Y., Sadanaga, Y., Sato, K., Sakamoto, Y., Muraoka, T., Miyatake, K., Li, J., and Kajii, Y.: Direct evaluation of the ozone production regime in smog chamber experiments, Atmospheric Environment, 309, 119889,http://doi.org/10.1016/j.atmosenv.2023.119889, 2023.

Qian, H., Xu, B., Xu, Z., Zou, Q., Zi, Q., Zuo, H., Zhang, F., Wei, J., Pei, X., and Zhou, W.: Anthropogenic Oxygenated Volatile Organic Compounds Dominate Atmospheric Oxidation Capacity and Ozone Production via Secondary Formation of Formaldehyde in the Urban Atmosphere, ACS ES&T Air,http://doi.org/10.1021/acsestair.4c00317, 2025.

Sakamoto, Y., Sadanaga, Y., Li, J., Matsuoka, K., Takemura, M., Fujii, T., Nakagawa, M., Kohno, N., Nakashima, Y., and Sato, K.: Relative and absolute sensitivity analysis on ozone production in Tsukuba, a city in Japan, Environmental science & technology, 53, 13629-13635,http://doi.org/10.1021/acs.est.9b03542, 2019.

Sklaveniti, S., Locoge, N., Stevens, P. S., Wood, E., Kundu, S., and Dusanter, S.: Development of an instrument for direct ozone production rate measurements: Measurement reliability and current limitations, Atmospheric Measurement Techniques, 11, 741-761,http://doi.org/10.1029/98jd00349, 2018.

Wang, W., Yuan, B., Peng, Y., Su, H., Cheng, Y., Yang, S., Wu, C., Qi, J., Bao, F., and Huangfu, Y.: Direct observations indicate photodegradable oxygenated volatile organic compounds (OVOCs) as larger contributors to radicals and ozone production in the

atmosphere, Atmospheric Chemistry and Physics, 22, 4117-4128,http://doi.org/10.5194/acp-22-4117-2022, 2022.

Wang, W., Yuan, B., Su, H., Cheng, Y., Qi, J., Wang, S., Song, W., Wang, X., Xue, C., and Ma, C.: A large role of missing volatile organic compound reactivity from anthropogenic emissions in ozone pollution regulation, Atmospheric Chemistry and Physics, 24, 4017-4027,http://doi.org/10.5194/acp-24-4017-2024, 2024.

Yuan, B., Shao, M., De Gouw, J., Parrish, D. D., Lu, S., Wang, M., Zeng, L., Zhang, Q., Song, Y., and Zhang, J.: Volatile organic compounds (VOCs) in urban air: How chemistry affects the interpretation of positive matrix factorization (PMF) analysis, Journal of Geophysical Research: Atmospheres, 117,http://doi.org/10.1029/2012jd018236, 2012.

Yuan, B., Hu, W. W., Shao, M., Wang, M., Chen, W. T., Lu, S. H., Zeng, L. M., and Hu, M.: VOC emissions, evolutions and contributions to SOA formation at a receptor site in eastern China, Atmospheric Chemistry and Physics, 13, 8815–8832, https://doi.org/10.5194/acp-13-8815-2013, 2013.

Zhou, J., Wang, W., Wang, Y., Zhou, Z., Lv, X., Zhong, M., Zhong, B., Deng, M., Jiang, B., and Luo, J.: Intercomparison of measured and modelled photochemical ozone production rates: Suggestion of chemistry hypothesis regarding unmeasured VOCs, Science of The Total Environment, 951, 175290,http://doi.org/10.1016/j.scitotenv.2024.175290, 2024a.

Zhou, J., Zhang, C., Liu, A., Yuan, B., Wang, Y., Wang, W., Zhou, J.-P., Hao, Y., Li, X.-B., and He, X.: Measurement report: Vertical and temporal variability in the near-surface ozone production rate and sensitivity in an urban area in the Pearl River Delta region, China, Atmospheric Chemistry and Physics, 24, 9805-9826, doi: 10.5194/acp-24-9805-2024, 2024b.

**Anonymous Referee #2**

Zhong and colleagues present measurements and detailed analysis using constrained box model approaches of in-situ ozone formation at a field site in the PRD, using a newly-developed direct measurement of ozone production rates, alongside measurements of various atmospheric chemical species/ photochemical parameters.

The paper presents an extensive exploration of the measurements, assessing the NOx- and VOC-dependence of the measured and modelled ozone formation, and relating this to (e.g.) missing VOC species. The approach is logical and largely well described (although I have some significant suggestions for clarifications – below), and the work represents a good advance in approaches to analysis of these new measurement approaches/data and could make a valuable contribution; in particular the assessment of ozone production "gaps" vs model with co-reactant concentrations/conditions (NOx/VOC sensitivity)

My principal concern is that the degree of accuracy (and maybe precision) of the measurements may be overestimated, and that the analysis of these – still relatively new – measurements is taken further than the data uncertainties really justify; that the data are over-interpreted.

Thank you for your suggestion. We have thoroughly discussed the measurement accuracy and uncertainties of the custom-made net $O_3$ production rate (NPOPR) detection system in our previous studies (Hao et al., 2023; Zhou et al., 2024b), and found that the measurement accuracy of the NPOPR detection system is determined as 13.9 %, this is estimated from the systematic errors inherent in the system representing the maximum systematic error resulting from photochemical $O_3$ production in the reference chamber.

These errors arise from photochemical $O_3$ productions in the reference chamber, because of the UV protection Ultem film that only filters out the sunlight with wavelengths less than 390 nm. Consequently, photochemical $O_3$ production from sunlight wavelengths between 390 nm and 790 nm still exist in the reference chamber and causes the systemic errors mentioned above.

Furthermore, according to the $P(O_3)_{net}$ evaluation method listed in Eq. (5) in the main text, the measurement error of $P(O_3)_{net}$ depends on the estimation error of Ox in the reaction and reference chambers, which includes the measurement error of $O_X$ of CAPS-$NO_2$ monitor and the error caused by the light-enhanced loss of $O_3$. These collective measurement error of $P(O_3)_{net}$ is referred to as the measurement precision of the NPOPR detection system, which is different with the measurement accuracy described above. This error refers to the degree of consistency or repeatability observed in a set of measurements by the NPOPR detection system. To make the description clearer, we have added this explanation in lines 127- 137 in the main text:

"The mean residence time in the reaction chamber is 0.15 h at the air flow rate of 2.1 L min$^{-1}$, and the limit of detection (LOD) of the NPOPR detection system is 0.86 ppbv h$^{-1}$ at the sampling air flow rate of 2.1 L min$^{-1}$, which is obtained as three times the measurement error of $P(O_3)_{net}$(Hao et al., 2023). The measurement error of $P(O_3)_{net}$ is determined by the uncertainty in the Ox mixing ratio estimated for both the reaction and reference chambers. This uncertainty combines (i) the measurement uncertainty of the CAPS-$NO_2$ monitor used to derive Ox and (ii) the error induced by light-enhanced $O_3$ loss inside the chambers. Taken together, these contributions define the measurement precision of the NPOPR detection system. In addition, the measurement accuracy of the NPOPR detection system is 13.9 %, corresponding to the maximum systematic error arising from photochemical $O_3$ production in the reference chamber (Hao et al., 2023; Zhou et al., 2024b); details are given in Sect. S1 in the supplementary materials."

And S1 in the supplementary materials:

**"S1. Measurement error of $P(O_3)_{net}$ of the NPOPR detection system**

We have thoroughly described the measurement error of $P(O_3)_{net}$ of the NPOPR detection system in our previous study (Hao et al., 2023; Zhou et al., 2024b). The measurement error of $P(O_3)_{net}$ depends on the estimation error of Ox in the reaction and reference chambers, which includes the measurement error of $O_X$ of CAPS-NO$_2$ monitor and the error caused by the light-enhanced loss coefficient of O$_3$ ($\gamma$), which can be calculated as follows:

$$(O_X)_{error} = \sqrt{(O_{X_\gamma})_{error}^2 + (O_{X_{CAPS}})_{error}^2}$$

(S1)

where $(O_X)_{error}$ represents the absolute error in the estimated $O_X$ concentration in the reaction and reference chambers, which results from the quadratic propogation of the absolute errors $(O_{X_\gamma})_{error}$ and $(O_{X_{CAPS}})_{error}$. Here, $(O_{X_{CAPS}})_{error}$ signifies the measurement error of the $O_X$ measured by the CAPS-NO$_2$ monitor, while $(O_{X_\gamma})_{error}$ denotes the error associated with the $\gamma$-corrected Ox of the chambers, where $\gamma$ represent the light-enhanced O$_3$ loss coefficient.

To get $(O_{X_{CAPS}})_{error}$, we calibrated the CAPS-NO$_2$ monitor as follows: a. injected ~10–100 ppbv of NO$_2$ for 30 minutes to passivate the surfaces of the monitor and then injecting ultrapure air for ~ 10 minutes to ensure the zero point did not drift, according to the ultrapure air condition, the LOD of CAPS was 0.88 and 0.02 ppbv (3 $\sigma$) at an integration time of 35 and 100 s, respectively; b. injected a wide range of NO$_2$ concentration (from 0–160 ppbv) prepared from a NO$_2$ standard gas (with the original concentration of 2.08 ppmv) mixed with ultrapure air into the CAPS-NO$_2$ monitor, repeated the experiments for three times at each NO$_2$ concentration, the final results are shown in Fig. S16.

[Figure]

**Figure S16: Calibration results of the CAPS NO₂ monitor at different NO₂ mixing ratios. The y-axis represents the NO₂ mixing ratios measured by the CAPS NO₂ monitor, and the x-axis represents the prepared NO₂ mixing ratios prepared from the diluted NO₂ standard gas.**

We fitted the calibration results with a 68.3 % confidence level, and the blue line in Fig. S16 represents the maximum fluctuation range under this confidence level, $(O_{X_{CAPS}})_{error}$ was then calculated from the fluctuation range of the 68.3 % confidence interval of the calibration curve, the relationship between the $(O_{X_{CAPS}})_{error}$ and the measured Ox value ($[Ox]_{measured}$) can be expressed as a power function curve, as shown in Eq. (S2) :

$$(O_{X_{CAPS}})_{error} = 9.72 \times [Ox]_{measured}^{-1.0024} \tag{S2}$$

We acknowledge that this power function has been derived from calibration data of the $O_X$ concentrations ranged from 20 ppbv to 160 ppbv. Utilizing this function outside this calibrated range, especially at very low $O_X$ concentrations, may result in errors that are disproportionately large and may not accurately capture the true variability of the measurement errors. In this study, the $O_X$ concentrations ranged from 18 to 148 ppbv, which falls into the calibration range. Consequently, this power function is deemed appropriate for estimating the $(O_{X_{CAPS}})_{error}$ throughout the whole measurement period.

$(O_{X_\gamma})_{error}$ was derived from the light-enhanced loss of O₃ in the reaction and reference chambers at 2.1 L min⁻¹, the flow rate used during the observation campaign. To establish the calibration curve, we performed an outdoor experiment: O₃ (~ 130 ppbv), produced by an O₃ generator (P/N 97-0067-02, Analytic Jena US, USA), was induced into the two chambers. Zero air was co-injected with the O₃ to

suppress any photochemical $O_3$ production outdoors. This setup allowed us to monitor daytime changes in the photolysis frequencies of various species. We simultaneously recorded $J(O^1D)$, $T$, RH, $P$ and $O_3$ mixing ratios at the inlets and outlets of both chambers. $T$ and RH were measured with a thermometer (Vaisala, HMP110, USA). The light-enhanced $O_3$ loss coefficient ($\gamma$) was then calculated using Eq. (S3):

$$\gamma = \frac{d[O_3] \times D}{\omega \times [O_3] \times \tau}$$

(S3)

where $d[O_3]$ represents the difference between the $O_3$ mixing ratios at the inlets and outlets of both chambers (i.e., the light-enhanced $O_3$ loss); D is the diameter of the chambers; $\omega$ is the average velocity of $O_3$ molecules; $[O_3]$ is the injected $O_3$ mixing ratio at the inlet; $\tau$ is the average residence time of the air in the reaction and reference chambers. The relationship between $J(O^1D)$ and $\gamma$ is shown in Fig. S18, the obtained $\gamma$-$J(O^1D)$ equation was used to correct $d[O_3]$ in both chambers during the daytime, thereby eliminating the influence of light-enhanced loss. Our previous study has shown that after this correction, $d[O_3]$ showed no clear correlation with RH for either chamber (Hao et al., 2023), indicating that RH did not affect the $O_3$ mixing ratio during the observation period. When quantifying $d[O_3]$ from ambient air measurements, we first calculate $\gamma$ from the measured $J(O^1D)$ using the $\gamma$-$J(O^1D)$ equations listed in Fig. S17 for each chamber, then compute $d[O_3]$ from the measured $[O_3]$ and Eq. (S3).

[Figure]

**Figure S17: The relationship between $\gamma$ and $J(O^1D)$ in the reaction and reference chambers, the shaded areas represent the maximum range of fluctuation under this confidence level.**

When injecting ambient air into the NPOPR system, the error of $P(O_3)_{net}$ with a residence time of $\tau$ can be calculated using Eq. (S4):

$$P(O_3)_{net\_error}=\frac{\sqrt{(O_{X_\gamma})_{rea\_error}^2+((9.72\times[(O_X]_{rea\_measured}^{-1.0024}))_{rea\_std}^2+(O_{X_\gamma})_{ref\_error}^2+((9.72\times[(O_X]_{ref\_measured}^{-1.0024}))_{ref\_std}^2}}{\tau}$$

(S4)

where $(O_{X_\gamma})_{rea\_error}$ and $(O_{X_\gamma})_{ref\_error}$ represent the measurement error due to light-enhanced loss of $O_3$ in the reaction and reference chambers, respectively, and $(9.72\times[O_X]_{measured}^{-1.0024})_{rea\_std}$ and $(9.72\times[O_X]_{measured}^{-1.0024})_{ref\_std}$ represent the standard deviation of $O_X$ in the reaction and reference chambers, respectively, caused by the CAPS $NO_2$ monitor with an integration time period of 100 s. Combined with the associated residence time $\langle\tau\rangle$ under different flow rates, i.e., $\langle\tau\rangle$ was 0.16 h at a flow rate of 2.1 L min$^{-1}$. In our previous research (Hao et al., 2023), we evaluated the residence time error and determined it to be approximately 0.0015, when we considered this error in the calculation of '$P(O_3)_{net\_error}$', we observed a minimal reduction in the '$P(O_3)_{net\_error}$' values, ranging from 0 to 4% [0.25-0.75 percentile]. This impact is considered negligible in relation to the overall '$P(O_3)_{net\_error}$' as presented in Eq. S4. Consequently, we did not consider the uncertainty associated with the residence time in our calculations. We note that this collective measurement error of $P(O_3)_{net}$ is referred to as the measurement precision of the NPOPR detection system, which is different with the measurement accuracy of the NPOPR detection system described above."

I think this is certainly the case for the model, considering the VOC coverage (correctly) identified and uncertainties in e.g. knowledge of HONO (not measured directly), and I would ask the authors to consider carefully if the measured $P(O_3)$ is really good to 10% accuracy – and hence if quite such an extensive set of analysis is warranted. I'm conscious that lots of things will more-or-less co-vary diurnally within the measurement uncertainty (concentrations, $j$, T, $O_3$…) – is it really possible to extract missing reactants at a few % accuracy from within the combined measurement and model uncertainties?

Thank you for your insightful comment. We have thoroughly discussed the measurement error (precision) and accuracy of the custom-made NPOPR

detection system, as described above. The evaluated measurement uncertainty of the $P(O_3)_{net}$ during the observation period are shown in Fig. S13. We see that the measurement uncertainty decreased with increasing $P(O_3)_{net}$ values, which ranges from 0-23%.

"

[Figure]

**Figure S13: (a) Time series of measured $P(O_3)_{net}$_Mea, $P(O_3)_{net}^{+NO}$ and $P(O_3)_{net}^{+VOCs}$ based on sensitivity experiments using the NPOPR detection system, with an enlarged view for an $O_3$ pollution day (October 26, 2023) and a normal ($O_3$ non-pollution) day (October 13, 2023). The shaded areas represent the errors of each measured term, calculated from the instrument measurement uncertainties given in Hao et al. (2023). (b) Relative errors of measured $P(O_3)_{net}$_Mea, $P(O_3)_{net}^{+NO}$, and $P(O_3)_{net}^{+VOCs}$ as a function of their measured values, ..."**

More details concerning the measurement uncertainty are added in lines 503-506 in the modified manuscript: "The time series of measured $P(O_3)_{net}$_Mea, $P(O_3)_{net}^{+NO}$ and $P(O_3)_{net}^{+VOCs}$ based on sensitivity experiments using the NPOPR detection system are shown in Fig. S13. We see the measurement uncertainty decreased with increasing $P(O_3)_{net}$ values: it reaches approximately 23% when $P(O_3)_{net}$ is around 0 ppbv h$^{-1}$, but falls below 3% when $P(O_3)_{net}$ is around 50 ppbv h$^{-1}$."

Fig. S13b shows that the higher measurement uncertainty usually appears when the $P(O_3)_{net}$ is relatively low (e.g., in the early morning, evening, or late afternoon), whereas the relatively large $P(O_3)_{net}$ missing occur mainly around noon. The different diurnal patterns of measurement uncertainty (caused by concentrations, $j$, T, $O_3$…) and the modelling bias responsible for the $P(O_3)_{net}$ missing indicate that the two do not co-vary on a diurnal basis. The $P(O_3)_{net}$ missing may be small when averaged over daytime; however, it can reach ~ 33% around noon. This value is much higher than the measurement uncertainty at that time (< 3%), making it possible to quantify the $P(O_3)_{net}$ missing in the model within a few percent. We added this discussion in lines 406-408 of the modified manuscript:

"Accurate quantification of $P(O_3)_{net}$ missing is possible here because the diurnal patterns of measurement uncertainty and the modelling bias responsible for the $P(O_3)_{net}$ missing do not co-vary; consequently, measurement uncertainty is much smaller than modelling bias for most of the daytime, especially around noon."

Showing more raw data – maybe 2-3 individual days plotted in detail so the measured and modelled data can be made out – would help the reader understand the sensitivities of the various metrics, alongside the "integrated" plots of sensitivities. This might help focus the manuscript also, as the story would be clearer with fewer analyses (and fewer SI figures) which would then also give confidence that the degree of analysis is appropriate and the data not over-interpreted.

Thank you for your suggestion. We have enlarged view for an $O_3$ pollution day (October 26, 2023) and a normal ($O_3$ non-pollution) day (October 14, 2023) in Figs. 3, 5, and S13 (as shown below).

"

[Figure]

**Figure 3: The time series and diurnal variations of $P(O_3)_{net}\_Mea$ and $P(O_3)_{net}\_Mod$ (Case $D_1–D_4$) during the observation period, with an enlarged view for an $O_3$ pollution day (October 26, 2023) and a normal ($O_3$ non-pollution) day (October 14, 2023); The shaded areas in (a) represent rainy days.**

[Figure]

**Figure 5: (a) Time series and (b) diurnal variations of $P(O_3)_{net}\_Mea$ and $P(O_3)_{net}\_Mod$ (Case $D_1–E_3$) during the observation period, with an enlarged view for an $O_3$ pollution day**

**(October 26, 2023) and a normal (O₃ non-pollution) day (October 14, 2023);** (b) Diurnal variations excluding rainy days. The shaded areas in (a) represent rainy days.

From Fig. 3a, we see the modelled results on individual days varies, which make the $P(O_3)_{net}$ missing different with the overall diurnal variation (shown in Fig. 3b). For example, $P(O_3)_{net}$ _Mod of Case D2 increased more on 17 October 2023 than that on October 14 2023, while $P(O_3)_{net}$ _Mod of Cases D3-D4 increased more on October 17 2023 than that on October 14 2023. These results indicated that the overall increasing or decreasing trends can't represent all individual days during the observation period. On the other hand, the constraining of more OVOCs only corrects the outcome, not the related reaction processes. For example, the $RO_2$ radicals, which are the main intermediates that drive the oxidation chain during photochemical $O_3$ formation, may vary day-to-day. We added the related description for Fig. 3 in lines 339-341 of the modified manuscript:

"Furthermore, the enlarged days in Fig. 2 reveal day-to-day variations in $P(O_3)_{net}$_Mod across the different cases, underscoring that the overall diurnal pattern described above does not resolve this variability."

We added the related description for Fig. 5 in lines 492-494 of the modified manuscript:

"However, we observe a slight difference in the diurnal trends of $P(O_3)_{net}$ across different days (enlarged view in Fig. 5); this depicts the overall pattern for the observation period described above does not capture day-to-day variability."

We detected an error when check the $P(O_3)_{net}$_Mod data from Case D₂ and Case D1, the overall daytime $P(O_3)_{net}$_Mod from Case D2 showed a slightly decreasing trend (0.5%) compared to $P(O_3)_{net}$_Mod from Case D1, therefore,

we have changed the sentence "However, the daytime average value of $P(O_3)_{net\_}$Mod from Case $D_2$ increased by only 0.5 % compared to Case $D_1$, this indicates that the dominant OVOCs species that causes $P(O_3)_{net\_}$Missing may vary between Heshan and Dongguan." to "However, the daytime mean $P(O_3)_{net\_}$Mod in Case $D_2$ decreased by 0.5 % compared with Case $D_1$, indicating that the dominant OVOC species responsible for $P(O_3)_{net\_}$Missing may differ between Heshan and Dongguan." in lines 390-391 of the modified manuscript.

I recognise the method is published in the Hao et al (2023) paper, but would encourage the authors to include more discussion of measurement uncertainties here, and at the least a detailed justification of the statement "…measurement uncertainties around 10%" (L79). this should include, separately, accuracy, precision, and selectivity/bias – the impact of wall artefacts on the measured P(O3), which may vary with conditions (j, RH, VOC/NOx levels), and would best appear around L120.

Thank you for your suggestion. The relevant discussion and data analysis of measurement uncertainty of the custom-made NPOPR detection system are provided above. In summary, our measurement error is calculated in real time based on the light intensity and the Ox concentrations of the ambient air. As mentioned above, the related experiments and data analysis are shown in the Supplementary Materials S1. The related discussion has been added in lines 127-137 of the modified manuscript:

"The mean residence time in the reaction chamber is 0.15 h at the air flow rate of 2.1 L min⁻¹, and the limit of detection (LOD) of the NPOPR detection system is 0.86 ppbv h⁻¹ at the sampling air flow rate of 2.1 L min⁻¹, which is obtained as three times the measurement error of $P(O_3)_{net}$(Hao et al., 2023). The measurement error of $P(O_3)_{net}$ is determined by the uncertainty in the Ox mixing ratio estimated for both the reaction and reference chambers. This uncertainty combines (i) the measurement uncertainty of the CAPS-NO₂ monitor used to derive Ox and (ii) the error induced by light-enhanced O₃ loss inside the chambers. Taken together, these contributions define the measurement precision of the NPOPR detection system. In addition,

the measurement accuracy of the NPOPR detection system is 13.9 %, corresponding to the maximum systematic error arising from photochemical $O_3$ production in the reference chamber (Hao et al., 2023; Zhou et al., 2024b); details are given in Sect. S1 in the supplementary materials."

Furthermore, we checked the measurement uncertainties of different $O_3$ production sensors worldwide and confirmed that the uncertainties ranged from 10-30%. Therefore, we have changed the sentence to "Through practical applications in field observations, scholars generally agree that these detection systems offer rapid stability and high precision, with measurement uncertainties ranged from 10-30 %." in line 84 of the modified manuscript.

Corrections / Comments

L47 and following – it would be useful to distinguish between NO titration of O3 – ie NOx/O3 PSS shifts – and net production of Ox (which is what we really mean by ozone production). Several points in the text later (eg L134) there is reference to titration reducing ozone production – I'd argue that this is PSS shift, not a change in the ozone production chemistry, and a different terminology might help.

We apologize for the ambiguous description. We have changed the sentence to: "IR is defined as the change in $P(O_3)_{net}$ per unit change in precursor concentration ($\Delta S(X)$): a negative IR value indicates that reducing the precursor concentration increases $O_3$ production (e.g., decrease NOx would increase $O_3$ through OH mediate effect), …" in line 156 of the modified manuscript.

Furthermore, we changed "This midday transition to $NO_X$-limited conditions is chemically reasonable, where intensified $NO_2$ photolysis boosts $O_X$ production while concurrently diminished NO titration and declining VOCs emissions collectively favor $NO_X$-sensitive chemistry during peak sunlight hours (Wang et al., 2023)." to "This midday transition to $NO_X$-limited conditions is chemically reasonable, where intensified $NO_2$ photolysis boosts

$O_X$ production while persistent photochemistry consumption without replenishment (Wang et al., 2023)." In lines 512-513 of the modified manuscript to make the description clearer.

L77 please acknowledge / include the pioneering work of Brune and colleagues (Cazorla et al., 2012) as the first "modern" MOPS system developers (I realise this is referenced later).

Thank you for your reminder. We have added the pioneering work of Brune and Cazorla as references (Cazorla and Brune, 2009; Cazorla et al., 2012) in line 81 of the modified manuscript:

"To date, several $P(O_3)_{net}$ detection systems based on the dual-reaction chamber technique have been developed, referred to as measurement of $O_3$ production sensor (MOPS), $O_3$ production rate measurement system (O3PR), $O_3$ production rates instrument (OPRs), net photochemical $O_3$ production rate detection system (NPOPR), Mea-OPR, or $O_3$ production rate-cavity ring-down spectroscopy system (OPR-CRDS) (Baier et al., 2015; Cazorla and Brune, 2009; Cazorla et al., 2012; Sadanaga et al., 2017; Sklaveniti et al., 2018; Hao et al., 2023; Wang et al., 2024c; Tong et al., 2025)."

L79 Measurement uncertainties – I do not think 10% is realistic – see general comments above

We apologize that we made a mistake here. As mentioned above, we checked the measurement uncertainties of different $O_3$ production sensors worldwide and confirmed that the uncertainties ranged from 10-30%. Therefore, we have changed the sentence to "Through practical applications in field observations, scholars generally agree that these detection systems offer rapid stability and high precision, with measurement uncertainties ranged from 10-30 %." in the modified manuscript.

L100 is there much emission / chemical heterogeneity around the site? e.g. on the timescale of NOx PSS (1 min+) or HONO PSS (10-15min+)?

Thank you for your thoughtful consideration. The observation site, the Guangdong Atmospheric Supersite of China, is located in a farmland conservation zone and forested region at the suburban area of Heshan City. There are no major local industrial emission sources and the motorcycles dominate urban transport in Heshan City. However, the supersite experiences minimal spatial heterogeneity in either primary emissions or chemical composition as it is located on a small mountain approximately 3 km from the nearest area with heavy traffic emissions. With a mean wind speed is 2.8 m s$^{-1}$ during the observation period, the air mass originating from the traffic corridor requires ~ 17 min to reach the supersite; consequently, rapid dilution and initial photochemical processing of exhaust plumes occur before they reach the supersite.

We changed the sentence "The supersite is situated in the downwind area of Guangzhou and Foshan and is characterized by active secondary reactions. It lies at the intersection of forest-agricultural and urban systems, representing a typical rural station. The surrounding area primarily consists of farmland conservation zones and forested areas, with no significant industrial emissions. It is suitable for comprehensive monitoring and research on regional atmospheric complex pollution in the PRD (Mazaheri et al., 2019)." to "The supersite is situated in the downwind area of Guangzhou, Foshan, and Dongguan, a region characterized by active secondary reactions and serving as a receptor for pollution transported from the industrial and urban centers (Luo et al., 2025; Huang et al., 2020). The surrounding area is primarily composed of farmland conservation zones and forested regions, with no major industrial sources. The supersite sits on a small mountain ~ 3 km from the nearest area heavy traffic corridor; at the observed mean wind speed of 2.8 m s$^{-1}$, the air mass from the corridor takes ~ 17 min to arrive. This separation limits spatial heterogeneity in both emissions and chemical composition, making the site well-suited for comprehensive monitoring and research on complex regional air pollution in the PRD (Mazaheri et al., 2019)." in lines 104-111 of the modified manuscript.

L131 Explain how the VOC addition amounts were determined/apportioned between the two species

According to the previous study, the selection of the VOCs indicator for the $O_3$ formation sensitivity measurement can be determined using the VOCs measured previous to the $O_3$ formation sensitivity (OFS) measurements (Carter et al., 1995; Wu et al., 2022).  We first calculated the total VOCs reactivity using the daytime VOCs measured from 20 September to 3 October and from 4-11 October 2023 at the observation site. For the OFS measurement from 4-11 October, we used the total VOCs reactivity measured from 20 September to 3 October 2023, VOCs indicators included isopentane as the representative alkane, ethylene and isoprene as the representative alkenes, and toluene as the representative aromatic hydrocarbon. From 13-26 October, we used the averaged daytime total VOCs reactivity measured from 4 to 11 October 2023, VOCs indicators included Ethylene was used as the representative non-methane hydrocarbon (NMHC) indicator and formaldehyde as the representative oxygenated volatile organic compound (OVOC) indicator. The related description is added in lines 143-151 of the modified manuscript:

"Following Carter et al. (1995) and Wu et al. (2022), we select VOCs surrogates for the OFS measurement on the basis of ambient measurements previous to the measurements. From 4–11 October, the tracer mixture was formulated from the average daytime total VOC reactivity measured during 20 September–3 October 2023, and isopentane served as the alkane surrogate, ethylene and isoprene as the alkene surrogates, and toluene as the aromatic surrogate. For 13–26 October 2023, we used the average daytime total VOC reactivity obtained during 4–11 October 2023; ethylene represented non-methane hydrocarbons (NMHCs) and formaldehyde represented oxygenated VOCs (OVOCs). Each surrogate was mixed in proportion to its category's share of the ambient reactivity, and the effective precursor strength (NO or VOCs) should increase by 20 % relative to the original ambient level."

L202 E10 – I do not follow how the net P(O3) is equal to P(O3) multiplied by the change in P(O3) divided by the (natural log of) change in X.

We apologize for the error in the equation expression. The description has been revised in lines 223-227 of the modified manuscript:

"We calculated the modelled OFS using the absolute $P(O_3)_{net}$ sensitivity, adapted from the logarithmic derivative approach of Sakamoto et al. (2019). It is defined as the change in $P(O_3)_{net}$ with respect to the natural logarithm of $O_3$ precursor concentrations. This method facilitates the quantitative assessment of how reductions in $O_3$ precursors contribute to the overall reduction of $P(O_3)_{net}$ over a period or within a region. The formula is as follows:

$$\text{Absolute } P(O_3)_{net} = \frac{dP(O_3)_{net}}{d\ln[X]}$$

(10)"

L234 not sure "stronger" photochemical reactions is right word – do you mean higher photolysis rates? An alternative explanation for the (slightly) lower concs might be greater solar heating/higher BLH/more dilution on the hotter/sunnier/higher P(O3) days – evidence in the BLH data?

Thank you for your suggestion. Yes, we mean higher photolysis rates occur on $O_3$ pollution days (see Fig. S4a). We further checked the planetary boundary-layer height (PBLH) on both $O_3$ pollution days and normal days. We found that in the morning the PBLH was higher on $O_3$ pollution days, whereas in the afternoon it became lower than on normal days (see Fig. S4k). During the period of strongest sunlight (11:00-14:00), the PBLH on $O_3$ pollution days and normal days does not appear statistically different (*t-test*, *p*=0.45). Therefore, the lower concentrations of TVOC and $NO_X$ on $O_3$ pollution days compared with normal are not likely due to changes in PBLH change or increased dilution.

Accordingly, we have changed the description from "This suggests that stronger photochemical reactions occur on $O_3$ pollution days, leading to lower daytime concentrations of precursors compared to normal days." to "As the PBLH on $O_3$ pollution days and normal days does not differ statistically during the period of strongest solar radiation (11:00-14:00, *t-test*, *p*=0.45, see Fig. S4k), the lower daytime concentrations/mixing ratios of $O_3$ precursors on $O_3$ pollution days than on normal days may be due to higher photolysis rates on $O_3$ pollution days (see Fig. S4a)." in lines 257-259 in the modified manuscript.

L305 give the missing P(O3) as a % also – maybe 24 hour mean. Statistical test – I assume *P* level of 0.05 not 0.5?

Sorry for the confusion description. We have modified the sentence to make it clearer, as shown in lines 338-343 in the modified manuscript:

"On non-rainy days, the averaged daytime $P(O_3)_{net\_}$Missing reached $4.5\pm7.6$ ppbv h$^{-1}$, accounting for 31% of the total measured $P(O_3)_{net}$. The averaged daytime $P(O_3)_{net\_}$Missing values on $O_3$ pollution days were statistically higher than those on normal days (*t-test*, *p*<0.05), suggesting that while the supplementary mechanisms explored in the model may contribute to some extent, they are unlikely to be the dominant cause of the $P(O_3)_{net\_}$Missing."

L325+: Is there really sensitivity in the correlations to identify particular causes?

Thank you for your question. The correlation analysis is only a preliminary examination of the factors that may be related to the $P(O3)_{net\_}$Missing in the model, aimed at guiding further investigation; it therefore does not allow us to identify specific causes. We have softened the wording in the sentences in lines 367 to 374in the modified manuscript:

"To explore the possible drivers of $P(O_3)_{net\_}$Missing, we correlated it with TVOC, $NO_X$, $J_{O1D}$, $T$, and $O_X$ separately for $O_3$ pollution days and normal days (Fig. S11). On $O_3$ pollution days, $P(O_3)_{net\_}$Missing exhibited a moderate positive correlation with VOCs ($r^2$ =0.4, R=0.2, t=2.9) and $NO_X$ ($r^2$=0.5, R=0.2, t=3.8), confirming that the $P(O_3)_{net\_}$Missing is larger at higher

precursor concentrations/mixing ratios (both t > critical 2.0, $p < 0.05$), consistent with earlier box-model studies (Whalley et al., 2021; Ren et al., 2013; Zhou et al., 2024a). A moderate positive correlation is also found with $J_{O1D}$ on both $O_3$ pollution days and normal days, with $r^2$ values of 0.5 and 0.4, respectively.   On normal days all correlations collapse ($r^2 < 0.2$, $p > 0.1$), implying that the model deficit is not tied to the measured precursors under low-$NO_x$ conditions and may instead related to the missing mechanisms for unmeasured photolabile VOCs."

L345 is it valuable to include all of D1-D4 – cut straight to the final case, D4?

By setting different simulated scenarios from Cases D1–D4, we primarily wanted to check quantitively whether the additional mechanisms and the measured OVOCs have a significant influence on $P(O_3)_{net\_}$Missing. The configurations of each scenario are as follows: Case A considers only the simplified chemical reaction mechanism from MCM v3.3.1; Case B incorporates the $HO_2$ uptake by ambient aerosols mechanism based on Case A; Case C further includes the dry deposition processes of key species on top of Case B; Case D1 extends Case C by adding the $N_2O_5$ uptake mechanism and Cl · related heterogeneous reaction mechanisms. Case D2 includes the measured OVOCs based on Case D1— namely, acetaldehyde, acrolein, acetone, and butanone —which were considered qualitatively as potential contributors to $P(O_3)_{net\_}$Missing in Dongguan in our previous study (Zhou et al., 2024); Case D3 constrained all measured OVOC species in Heshan based on Case D2; Case D4 constrained chlorine-containing VOCs (i.e., all measured VOC species listed in Table S8 that could be input into the OBM model). Detailed simulation parameter settings are provided in the main text and the Supplementary Materials (Table S3). We believe that this step-by-step simulation process is necessary for a better understanding of the different mechanisms and the impact of OVOCs on $P(O_3)_{net\_}$Missing.

Fig S13 – It is very hard to see the change in $PO_3$ from the added NO/added VOCs – suggest show a zoom in on a polluted/non-polluted day in addition so the data can be seen. The uncertainty ranges look very small on this figure?

Thank you for your suggestion. We have added a zoom in on a $O_3$ pollution day (October 26, 2023) and a normal ($O_3$ non-pollution) day (October 13, 2023) in Fig. S13 to make it clearer.

"

[Figure]

Figure S13: (a) Time series of measured $P(O_3)_{net}$_Mea, $P(O_3)_{net}^{+NO}$ and $P(O_3)_{net}^{+VOCs}$ based on sensitivity experiments using the NPOPR detection system, with an enlarged view for an $O_3$ pollution day (October 26, 2023) and a normal ($O_3$ non-pollution) day (October 14, 2023). The shaded areas represent the errors of each measured term, calculated from the instrument measurement uncertainties given in Hao et al. (2023); **(b) Relative errors of measured $P(O_3)_{net}$_Mea, $P(O_3)_{net}^{+NO}$, and $P(O_3)_{net}^{+VOCs}$ as a function of their measured values; …**"

The measurement uncertainties, shown as shaded areas in Fig. S13, are calculated as the measurement error described in Hao et al. (2023) and Zhou et al. (2024): when injecting ambient air into the NPOPR system, the error of $P(O_3)_{net}$ with a residence time of $\tau$ can be calculated using this equation:

$$P(O_3)_{net\_error}=\frac{\sqrt{(O_{X_Y})_{rea\_error}^{2}+((9.72\times[(O_X]_{rea\_measured}^{-1.0024})_{rea\_std})^{2}+(O_{X_Y})_{ref\_error}^{2}+((9.72\times[(O_X]_{ref\_measured}^{-1.0024})_{ref\_std})^{2}}}{\tau}$$

$(O_{X_Y})_{rea\_error}$ and $(O_{X_Y})_{ref\_error}$ represent the measurement error due to light-enhanced loss of $O_3$ in the reaction and reference chambers, respectively, and $(9.72\times[O_X]_{measured}^{-1.0024})_{rea\_std}$ and $(9.72\times[O_X]_{measured}^{-1.0024})_{ref\_std}$ represent the standard deviation of $O_X$ in the reaction and reference chambers, respectively, caused by the CAPS $NO_2$ monitor with an integration time period of 100 s. Combined with the associated residence time $\langle\tau\rangle$ under different flow rates, i.e., $\langle\tau\rangle$ was 0.063 h at a flow rate of 5 L min$^{-1}$. Therefore, the instrument measurement error is determined by the measurement error of $O_X$ in the reaction and reference chambers, which may also be influence by the light-enhanced loss of $O_X$ in the reaction and reference chambers under ambient conditions when the light intensity (especially $J(O1D)$) and $O_3$ mixing ratios are high. The related description is added now in the Supplementary Materials S1.

To check the relative error of $P(O_3)_{net}\_Mea$, $P(O_3)_{net}^{+NO}$ and $P(O_3)_{net}^{+VOCs}$, we plotted the measurement error as a function of their measured values (see Fig. S13b). We find that the uncertainty decreased with increasing data values: it reaches approximately 23% when $P(O_3)_{net}$ is around 0 ppbv h$^{-1}$, but falls below 3% when $P(O_3)_{net}$ is around 50 ppbv h$^{-1}$.

We have now added such kind of description in lines 503-506 in the modified manuscript: "The time series of measured $P(O_3)_{net}\_Mea$, $P(O_3)_{net}^{+NO}$ and $P(O_3)_{net}^{+VOCs}$ based on sensitivity experiments using the NPOPR detection system are shown in Fig. S13. We see the measurement uncertainty decreased with increasing $P(O_3)_{net}$ values: it reaches approximately 23% when $P(O_3)_{net}$ is around 0 ppbv h$^{-1}$, but falls below 3% when $P(O_3)_{net}$ is around 50 ppbv h$^{-1}$."

L440ish: Ozone regime – there is only one data point in the afternoon showing a VOC limited regime (14:00). Is there really a shift from VOC to NOx to VOC

to NOx limited through the day – can the data really show this? I am conscious that there are not many days going into these averages. How is "transition regime" defined for the measurements? The explanation (L442): P(O3) measurement is not affected by NOx/O$_3$ titration (or PSS) – rather it measures change in the net Ox production.

Yes, we obtained only one data point showing a VOC-limited regime in the afternoon from the direct measurement. However, this diurnal profile shown in Fig. S14 is compiled from all days on which O$_3$ formation sensitivity was directly measured; it represents the overall trend during the observation period and does not reflect the day-to-day variation. As described in the main text, in total 11 days were incorporated into this calculation during the observation period, which includes 4–5, 11, 13–17, and 24–26 October 2023. We added the related discussion in lines 507-509 of the modified manuscript:

"Fig. S14 shows the diurnal variation of the directly measured IR index compiled from all 11 days of OFS experiments, together with the absolute $P(O_3)_{net}$ sensitivity to $NO_X$ and VOCs calculated with the box model (Case $D_1$, Eq. (10)). It therefore depicts the overall trend across the observation period and does not reflect the day-to-day variability."

Here, we define the transition regime as the region over which the IR shows a simultaneous increase or decrease upon addition of both VOCs and NO. We added it in lines 161-162 in Sect. 2.1. "We define the transition regime as the region over which the IR shows a simultaneous increase or decrease upon addition of both VOCs and NO."

Since we are measuring $P(O_3)_{net}$ in our NPOPR detection system, the measurement result is not affected by NOx/O$_3$ titration (or PSS), we have modified the sentence "This midday transition to $NO_X$-limited conditions is chemically reasonable, where intensified $NO_2$ photolysis boosts $O_X$ production while concurrently diminished NO titration and declining VOCs emissions collectively favor $NO_X$-sensitive

chemistry during peak sunlight hours (Wang et al., 2023)." to "This midday transition to $NO_X$-limited conditions is chemically reasonable, where intensified $NO_2$ photolysis boosts $O_X$ production while persistent photochemistry consumption without replenishment (Wang et al., 2023)." in lines 511-513 of the modified manuscript.

Fig 6 – please show the mean diurnals for PO3 (from the measurements) for the three regimes identified. Not sure that the rapid changes in emissions can be the explanation – the model is constrained to the observed concentrations, so it has this "built in".

Thank you for your suggestion. We have now conducted the mean diurnal cycles of $P(O_3)_{net}$ for the three regimes identified from the direct measurements. The $O_3$ formation sensitivity (OFS) of each day was diagnosed from its daily-integrated measurements; the mean diurnal variation for all days within the same OFS category was calculated. In total, eight days were classified as transition regime (4-5, 11, 14-15, 24-26 October 2023), two as VOC-limited regime (13 and 16 October 2023), and one as NOx-limited regime (17 October 2023). The resulting diurnal profiles are shown in Fig. S13c-d. Between 08:00 and 12:00 the mean diurnal profiles reveal a gradual shift from VOC-limited toward $NO_x$-limited conditions within the VOC-limited category, and a similar progression from transition to $NO_x$-limited within the transition category. We added the related description in lines 536-539 of the modified manuscript:

"To illustrate that the diurnal shift in OFS depicted in Fig. 6 is not random noise but reflects the general rule, we grouped the 11 days of direct measurements by their initial $O_3$-formation regime, calculated their average diurnal variations, and thus reproduced the "morning-transition" phenomenon in Fig. S13c–d."

And added the mean diurnal variation of $P(O_3)_{net}$ for the three regimes in Fig. S13c-d:

"

**Figure S13: … (c-e) Mean diurnal profiles of the three O₃ formation regimes identified: eight days classified as transition regime (4-5, 11, 14-15, 24-26 October 2023, two as VOC-limited regime (13 and 16 October 2023), and one as NOx-limited regime (17 October 2023).**"

We agree with the reviewer that the mean diurnal variations in O₃ formation sensitivity (OFS) are not influenced by rapid emission changes. As mentioned above, the site is situated on a small mountain ~3 km from the nearest heavy-traffic corridor; at the observed mean wind speed of 2.8 m s⁻¹, an air mass from the corridor takes ~17 min to arrive. This separation reduces spatial heterogeneity in both emissions and chemical composition at the observation site (see Sect. 2.1 of the revised manuscript).

A greater challenge in this regard may be the P(O3) measurements which average over an hour effectively?

We designed experiments to determine the OFS from direct measurements conducted daily from 8:00-18:00. Each measurement cycle lasted 1 hour: the first 20 min consisted of NO addition (denoted $P(O_3)_{net\_Mea}^{+NO}$), the next 20 min of ambient baseline measurement ($P(O_3)_{net\_Mea}$), and the final 20 min of VOCs addition ($P(O_3)_{net\_Mea}^{+VOCs}$). Therefore, we first interpolated $P(O_3)_{net\_Mea}^{+NO}$, $P(O_3)_{net\_Mea}$, and $P(O_3)_{net\_Mea}^{+VOCs}$ to 4-min resolution and then averaged these values over 1 h to eliminate the influence of the data fluctuation. However, the 1-hour averaging may smooth out transient

responses of the measured $P(O_3)_{net}$. The related description is now added in lines 151-153 in the modified manuscript:

"For data treatment, we first interpolated $P(O_3)_{net}$_Mea$^{+NO}$, $P(O_3)_{net}$_Mea, and $P(O_3)_{net}$_Mea$^{+VOCs}$ to 4-min resolution and then averaged them over 1 h to suppress data fluctuations. We caution that this 1-hour averaging may smooth out transient responses in the measured $P(O_3)_{net}$."

Isnt it more that you have already shown that the model (not unexpectedly) has bias from missing VOCs and this is reflected in these analyses also?

Yes, we have shown that the bias in the model may be attributed to missing VOCs, as reflected by the comparison of OFS results obtained from direct measurements and from modeling cases D1 and E1–E3 during the rising, stable, and declining phases of $P(O_3)_{net}$ (as described in Sect. 4). Briefly, in modeling cases where $P(O_3)_{net}$_Missing was reduced (Cases E1–E3), the simulated OFS occasionally shifted toward NOx-limited conditions during certain periods. This contradictory phenomenon may be related to the model's incomplete representation of the chemical mechanisms of unknown highly reactive VOCs (e.g., aldehydes and ketones), which is consistent with previous studies suggesting that diagnostic methods based on box models tend to overestimate VOC sensitivity due to the neglect of unidentified VOCs in anthropogenic emissions or their secondary products (Xu et al., 2022; Lu et al., 2010).

Therefore, we deleted the sentence "This low consistency may be related to rapid changes in precursor concentrations in the morning: the concentrations of VOCs and NO$_X$ concentrations change quickly during this period, particularly due to traffic emissions and industrial activities. These rapid variations make it challenging for the model to accurately capture the instantaneous reaction dynamics (Cao et al., 2021).", and added "These results demonstrate that the bias between measured and modeled OFS arises chiefly from missing

VOCs or shortcomings in the model's chemical mechanism." in lines 551-552 to explain the reason for the inconsistency between OFS derived from the direct measurement and the model simulation methods.

L489/Fig S15 – what are cases E1-E3? Not mentioned previously and I cannot find a definition / description of these.  I cannot follow L490-L505 as these cases/ scenarios are not defined

Sorry for the unclear description. In Case E1, the overall TVOC concentration was increased to compensate for $k_{OH}$_Missing without distinguishing VOCs categories. In Case E2, ethylene and formaldehyde were increased to compensate for $k_{OH}$_Missing. In Case E3, only formaldehyde concentration was expanded to compensate for $k_{OH}$_Missing. The detailed settings of each simulation case are listed in Table S3, and the explanations concerning Cases E1-E3 are listed in lines 476-484 in the modified manuscript:

"In Case $E_1$, where the overall TVOC concentration was increased to compensate for $k_{OH}$_Missing without distinguishing VOCs categories, the compensation effect was limited due to the dilution effect of low-reactivity VOCs, resulting in a reduction of the daytime average $P(O_3)_{net}$_Missing proportion from 26.3 % (calculated as $P(O_3)_{net}$_Missing/$P(O_3)_{net}$_Mea) to 10.3 %. In Case $E_2$, where the concentrations of ethylene and formaldehyde were expanded to compensate for $k_{OH}$_Missing, the daytime average $P(O_3)_{net}$_Missing proportion reduced from 26.3 % to 17.2 %. This proportion is higher than that obtained from Case E1, which may be due to the relatively low reactivity of ethylene limited the overall compensation effect. In contrast, Case $E_3$ compensated for $k_{OH}$_ Missing solely by expanding the formaldehyde concentration. More details concerning the cases settings are shown in Table S3."

Conclusions – is there a comment to make on the impact of different chemical mechanisms (eg L65+) on the model/measurement agreement?

We have investigated the influence of some missing mechanisms in MCM v3.3.1, such as HO2 uptake by ambient aerosols, dry deposition, N2O5 uptake,

and ClNO₂ photolysis (Case $D_1$), to the modelling results. However, some other reaction mechanisms, such as the $RO_2$ isomerization (Crounse et al., 2012), autoxidation (Wang et al., 2017), and the accretion reactions (Berndt et al., 2018) can also effect modelled $P(O_3)_{net}$, but these processes have not been investigated in this study. Therefore, we have added a comment on the impact of different chemical mechanism as follows in lines 418-421 of the modified manuscript:

"Previous studies have shown that the $RO_2$ isomerization (Crounse et al., 2012), autoxidation (Wang et al., 2017), and the accretion reactions (Berndt et al., 2018) can also effect modelled $P(O_3)_{net}$, but these processes have not been investigated here."

And lines 584-588 of the modified manuscript:

"These results also demonstrate that incorporating the aforementioned missing mechanisms and measured VOC species cannot fully eliminate simulation bias. Other processes, i.e., the $RO_2$, autoxidation, and the accretion reactions can also affect modelled $P(O_3)_{net}$, but they have not been examined here. The negative correlation of $P(O_3)_{net\_Missing}$ with the air mass aging indicates that the $P(O_3)_{net}$ missing is not likely caused by unaccounted secondary production."

Conclusions – do you wish to add an overall comment vs the NOx- vs VOC-control on O₃ observed at the site, and implications for policy vs reducing ozone formation rates? this might usefully also remind the reader of the distinction between the in-situ formation rate (focus here) and the total level experienced (integration of local formation and upwind chemistry / advection).

Yes, we have now added an overarching statement in lines 609-616 of the modified manuscript:

"In conclusion, we quantitatively assessed the $P(O_3)_{net}$ simulation deficits and their impact on OFS diagnosis by comparing the measured and modelled $P(O_3)_{net}$, and found that the unmeasured VOCs —rather than the secondary atmospheric formation —are the primary causative factor of $P(O_3)_{net\_Missing}$. Furthermore, both direct measurements and model results

reveal a diurnal OFS shift dominated by the morning regime; transition and VOC-limited conditions prevailed, so prioritizing VOCs while co-controlling $NO_X$ is the most effective approach to $O_3$ pollution control in PRD region. Our results also demonstrate that the persistent model biases risk under-estimating the local photochemical formation contribution to $O_3$ pollution, thereby has weakening its perceived impact relative to physical transportation. Future studies should expanded VOCs measurements and combine direct $P(O_3)_{net}$ observations with regional transport model to separate local production from up-wind advection."

Model Approach Clarifications

-A summary table of the different scenarios, A-E, would be very helpful – I think this is referred to but I cannot find in the SI?

There is a detailed description of different modelling scenarios from A-E, as shown in Table S3 in the SI:

**Table S3.** Description of different modelling scenarios and the parameter settings

| Case | Description | Parameter settings | references |
|---|---|---|---|
| A | Ambient gases (NO, $NO_2$, $SO_2$, CO, $O_3$), HONO, 44 VOCs, meteorological parameters (T, RH, P, BLH), photolysis rates, and $O_3$ dry deposition | $O_3$ (0.27 cm s$^{-1}$) | (Xue et al., 2014) |
| B | Case A with the addition of $HO_2$ uptake | $r_{H2O}$ =0.19 | (Zhu et al., 2020; Zhou et al., 2021) |
| C | Case B with the addition of trace gases ($NO_2$, $SO_2$, $H_2O_2$, $HNO_3$, PAN, HCHO) dry deposition | $NO_2$ (0.6 cm s$^{-1}$)

$SO_2$ (0.8 cm s$^{-1}$)

$H_2O_2$ (1.2 cm s$^{-1}$)

$HNO_3$ (4.7 cm s$^{-1}$)

PAN (0.4 cm s$^{-1}$)

HCHO (0.9 cm s$^{-1}$) | (Zhang et al., 2003; Xue et al., 2014) |
| $D_1$ | Case C with the addition of $N_2O_5$ non-homogeneous absorption reactions and $ClNO_2$ photolysis | $r_{N2O5}$ =0.02

$\varphi_{ClNO2}$ =0.6 | (Xue et al., 2014; Badger et al., 2006; Xia et al., 2019; Xia et al., 2020) |

[Figure]

| | | | |
|---|---|---|---|
| D₂ | Case D₁ with increased constraints for acetaldehyde, acrolein, acetone, and butanone | | —— |
| D₃ | Case D₁ with increased constraints for all measurable OVOCs | Constraints based on measurement data | —— |
| D₄ | Case D₃ with increased constraints for all measurable chlorinated VOCs | | —— |
| E₁ | Case D₁ with overall VOCs concentration in constraints increased | | —— |
| E₂ | Case D₁ with increased concentrations of ethylene and formaldehyde in constraints | Increase based on the correlation between $P(O_3)_{net\_Missing}$ and $k_{OH\_Missing}$ | —— |
| E₃ | Case D₁ with increased formaldehyde concentration in constraints | | —— |

Notes: Parameter values for modelling scenarios from Case A to Case D₁ are set the same as those in Zhou et al. (2024a).

-Model observation constraint: It would be helpful to explain how the constraint to observations was implemented – we have model outputs on an hourly basis but how frequently where the constraints applied to the model and did concentrations evolve freely in between (shorter model integration timestep)? If the model species are not in balance a "saw tooth" effect can result in the simulated concs at the higher model time resolution between observation constraint points – was this the case and if so impacts on the P(O3) which is in effect averaged over this period?

Thank you for your insightful suggestion. In this study, the box model was constrained by on-site observations of VOCs, NO, CO, HONO, and meteorological parameters (i.e., photolysis rates, RH, $T$, and P), as described in Sect. 2.2. Constraints were applied every hour, with no free concentration evolution in between. We have added the relevant description in lines 181–182 of the revised manuscript.

"The constraints are applied to the model every 1 h, with no free concentration evolution in between."

From Figs.3, 5, and S13, we do not see the "saw tooth" effect. Unfortunately, the data obtained in this campaign were mostly in 1 hour time resolution, thus we cannot evaluate how the model behaves at sub-hourly resolution between constraint points. We regard this as a critical issue that could materially affect modelled $P(O_3)_{net}$ and will therefore be addressed explicitly in our next study.

-Approach to HONO – tucked away in the SI, the use of MARGA measurements of soluble nitrite for gas-phase HONO is mentioned. I appreciate that some approach was needed, but the sensitivity of the model results to this assumption is needed – eg what shift in P(O3) does a 20% change in HONO concs (or whatever is reasonable) result in. If this is a large shift – are the subsequent analysis all valid, i.e. can you be confident in pushing the model explorations so far? What is the time resolution of the MARGA data vs the observed temporal variation of e.g. NOx?

Thank you for the suggestion. According to Xu et al. (2019), a large number of two-channel WD/IC instruments represented by the Monitor for AeRosols and Gases in ambient Air (MARGA) instruments was widely used to obtain aerosol composition information, as well as acid trace gas levels, including HONO (Stieger et al., 2018). However, the application of HONO data was limited because of the measurement uncertainty. The measurement uncertainty of HONO database obtained by MARGA was evaluated by Xu et al. (2019) and Spindler et al. (2003). For this purpose, Xu et al. (2019) used a MARGA and more accurate equipment (LOPAP) to simultaneously measure the HONO concentration at the Station for Observing Regional Processes of the Earth System (SORPES) in the YRD of east China; Spindler et al. (2003) performed the laboratory and field experiments as well as direct kinetic laboratory studies to quantify an artefact by the aqueous phase formation of HONO from dissolved

NO$_2$ and SO$_2$ at wetted denuder walls. In this study, we used the method proposed by Spindler et al. (2003) to check the measurement error of HONO by MARGA, and then checked its influence to the modelled $P$(O$_3$)$_{net}$. More details are added in the Supplementary Materials S2.

"However, previous studies have shown the HONO may be overestimated by MARGA due to aqueous phase formation of HONO from dissolved NO$_2$ and SO$_2$ at wetted denuder walls (Stieger et al., 2018; Spindler et al. 2003). The measurement error of HONO by MARGA was evaluated by Xu et al. (2019) and Spindler et al. (2003). In this study, we used the method proposed by Spindler et al. (2003) to evaluate measurement uncertainty of HONO database obtained by MARGA, and then checked its influence to the modelled $P$(O$_3$)$_{net}$. The overall artefact formation measurement error of HONO by MARGA is expressed as a sum in Eq. (S5):

$$[HNO_2]_{art} = 0.056[NO_2] + (0.0032/ppb)\ [NO_2][SO_2] \tag{S5}$$

where 0.0032 is the reciprocal value of the slope of the straight line between the HNO$_2$ concentration corrected for the HNO$_2$ content in purified air, the mean NO$_2$ artefact and the concentration product of NO$_2$ and SO$_2$. We further modelled $P$(Ox)$_{net}$_Case D$_4$ with the corrected HONO, and found that the corrected HONO could decrease the modelled $P$(Ox)$_{net}$_Case D$_4$ by 0-8%, as shown in Fig. S18. Therefore, we note that with the measurement error of HONO by MARGA, the modelling method may consistently underestimate the modelled $P$(Ox)$_{net}$ in all cases, and the $P$(Ox)$_{net}$_missing in our study should be regarded as the lower limit values.

[Figure]

**Figure S18: The modelled $P(Ox)_{net}$_Case D₄ and the with and without the HONO correction.**"

Accordingly, we added the related description in lines 334-336 of the modified manuscript:

"Due to the measurement error of HONO by MARGA in this study, the modelled $P(O_3)_{net}$ tends to be underestimated (as shown in SM: S2); thus, we define the $P(O_3)_{net}$_Missing obtained from all simulation cases as the upper-limit values."

And lines 574-576 of the modified manuscript:

"Systematic underestimation of modelled $P(O_3)_{net}$ ($P(O_3)_{net}$_Mod) was found when compared to the measured $P(O_3)_{net}$ ($P(O_3)_{net}$_Mea); this difference is defined as upper-limit $P(O_3)_{net}$_Missing due to the overestimation of HONO by MARGA in this study."

The nitrogen oxides analyzer (Fengyue Aorui-1014, China) provided $NO_X$ data at a temporal resolution of 1 min, while the $NO_2^-$ data from the MARGA were recorded at 1 h resolution. Therefore, all data were averaged to 1 h for model input, as stated in lines 163–165 of the main text.

"In addition to $P(O_3)_{net}$ and OFS, hourly data such as $PM_{2.5}$, $O_3$, NO, $NO_2$, $SO_2$, carbon monoxide (CO), photolysis rates ($j_{O^1D}, j_{NO_2}, j_{H_2O_2}, j_{NO_3\_M}, j_{NO_3\_R}, j_{HONO}, j_{HCHO\_M}, j_{HCHO\_R}$), HONO,

and VOCs concentrations were monitored (more details about the measurements are shown in Table S1)."

-Lots on the deposition velocities but how well do we know the boundary layer height BLH?

The planetary boundary layer height (PBLH) is the reanalysis data obtained from the website of the NOAA Air Resources Laboratory (https://ready.arl.noaa.gov/READYamet.php). The diurnal changes of PBLH on $O_3$ pollution days and normal days are added in Fig. S4 in the Supplementary Materials. We have provided an additional description on the source of the PBLH data in lines 167-169 in the revised manuscript:

"The PBLH data used in the model here was obtained from the web portal of the Real-time Environmental Applications and Display sYstem (READY) of the National Oceanic and Atmospheric Administration (NOAA) Air Resource Laboratory (https://ready.arl.noaa.gov/READYamet.php)."

L172 – is the $HO_2$ uptake process an irreversible loss (in the model)?

Yes, we used the data obtained from Zhou et al. (2020, 2023) measured from the ambient air and assumed it is an irreversible loss in the model.

L174 – is the $N_2O_5$ uptake an irreversible loss – if I follow some scenarios included recycling via $ClNO_2$ – not quite clear how the condensed phase processed were simulated?

Previous study has found that the absorption of $N_2O_5$ on aerosols surfaces containing chloride ions leads to the formation of $ClNO_2$ (Finlayson-Pitts et al., 1989). This $N_2O_5$ uptake process is an irreversible loss: it converts $N_2O_5$ into stable soluble nitrate and potentially volatile $ClNO_2$, thereby permanently removing gaseous $N_2O_5$. $ClNO_2$ will be photolyzed into $Cl·$ and $NO_2$ under

sunlight (Chen et al., 2023; Ma et al., 2023; McNamara et al., 2020; Peng et al., 2021). The detailed reactions are as follows:

$$N_2O_5 + Cl^-(aq) \rightarrow \varphi ClNO_2 + (2 - \varphi)NO_3^-(aq)$$

$$ClNO_2 \xrightarrow{k_2} Cl \cdot + NO_2$$

$$k_2 = j_{ClNO_2}$$

Where $\varphi$ is the production yield of $ClNO_2$, $k_2$ is the photolysis rate of ClNO$_2$ ($j_{ClNO_2}$). These details are shown in Supplementary Materials S3.

However, the current chemical mechanism, MCM v3.3.1, lacks these reaction processes, so we added the mechanism of N$_2$O$_5$ uptake by aerosols and photolysis of ClNO$_2$ to MCM v3.3.1 to simulate ClNO$_2$ and explored its impact on $P$(O$_3$)$_{net}$. We set the heterogeneous uptake coefficient of N$_2$O$_5$ as 0.02 ($\gamma_{N_2O_5} = 0.02$), and the production yield of $ClNO_2$ as 0.6 ($\varphi = 0.6$). Here the ClNO$_2$ was derived from model simulations and its related reactions are added into MCM v3.3.1 as:

"% ASA*24175.8*0.02/4:N2O5 = 0.6 ClNO2+1.4 PNO3;"

"%J<45>: ClNO2 = Cl +NO2;"

Where ASA represent the surface area of the ambient aerosols, 24175.8 represent the mean molecular speed of N$_2$O$_5$ (cm s$^{-1}$), 0.6 represent the production yield of ClNO$_2$, J<45> represent the photolysis rate of ClNO$_2$. As our previous study has thoroughly described these mechanisms (Zhou et al., 2024a), we have modified the description in lines 199-200 of the modified manuscript:

"…, and Case D$_1$ extends Case C by adding the N$_2$O$_5$ uptake mechanism and Cl· related photochemical reactions. Detailed simulation parameter settings can be found in our previous study (Zhou et al., 2024a) and the supplementary information (Table S3)."

"**Table S3.** Description of different modelling scenarios and the parameter settings

| Case | Description | Parameter settings | references |
|---|---|---|---|
| A | Ambient gases (NO, NO$_2$, SO$_2$, CO, O$_3$), HONO, 44 VOCs, meteorological parameters ($T$, RH, $P$, BLH), photolysis rates, and O$_3$ dry deposition | O$_3$(0.27 cm s$^{-1}$) | (Xue et al., 2014) |
| B | Case A with the addition of HO$_2$ uptake | $\gamma_{HO2}$=0.19 | (Zhu et al., 2020; Zhou et al., 2021) |
| C | Case B with the addition of trace gases (NO$_2$, SO$_2$, H$_2$O$_2$, HNO$_3$, PAN, HCHO) dry deposition | NO$_2$(0.6 cm s$^{-1}$)

SO$_2$(0.8 cm s$^{-1}$)

H$_2$O$_2$(1.2 cm s$^{-1}$)

HNO$_3$(4.7 cm s$^{-1}$)

PAN(0.4 cm s$^{-1}$)

HCHO(0.9 cm s$^{-1}$) | (Zhang et al., 2003; Xue et al., 2014) |
| D$_1$ | Case C with the addition of N$_2$O$_5$ non-homogeneous absorption reactions and ClNO$_2$ photolysis | $\gamma_{N2O5}$=0.02

$\varphi_{ClNO_2}$ =0.6 | (Xue et al., 2014; Badger et al., 2006; Xia et al., 2019; Xia et al., 2020) |
| D$_2$ | Case D$_1$ with increased constraints for acetaldehyde, acrolein, acetone, and butanone | | —— |
| D$_3$ | Case D$_1$ with increased constraints for all measurable OVOCs | Constraints based on measurement data | —— |
| D$_4$ | Case D$_3$ with increased constraints for all measurable chlorinated VOCs | | —— |
| E$_1$ | Case D$_1$ with overall VOCs concentration in constraints increased | | —— |
| E$_2$ | Case D$_1$ with increased concentrations of ethylene and formaldehyde in constraints | Increase based on the correlation between $P$(O$_3$)$_{net}$_Missing and $k_{OH}$_Missing | —— |
| E$_3$ | Case D$_1$ with increased formaldehyde concentration in constraints | | —— |

Notes: Parameter values for modelling scenarios from Case A to Case D$_1$ are set the same as those in

Zhou et al. (2024a)."

Minor Points

Please define OFS where first used (abstract, L61)

Thank you for your suggestion. We have added the definition of $O_3$ formation sensitivity (OFS) in lines 44-49 in the modified manuscript:

"The sensitivity of $O_3$ formation to its precursors is defined as the $O_3$ formation sensitivity (OFS), which can be classified into three regimes: NOx-limited, VOC-limited, or mixed sensitivity (Seinfeld & Pandis, 2016; Sillman, 1999). In an NOx-limited regime, the VOC/NOx ratio is high and $O_3$ production is controlled primarily by changes in NOx. In a VOC-limited regime, the VOC/NOx ratio is low, so $O_3$ decreases with additional NOx and increases with higher VOCs. In the mixed-sensitivity regime, $O_3$ rises when either NOx or VOC emissions increase (Wang et al., 2019)."

Accordingly, we used OFS directly in lines 65 in the modified manuscript:

"These gaps lead to systematic biases in the simulated $P(O_3)_{net}$ (Woodward-Massey et al., 2023; Tan et al., 2017; Tan et al., 2019), thereby affecting the accurate determination of OFS."

The related references are added in the references list:

"Seinfeld, J. H. and Pandis, S. N. (Eds. 3): Atmospheric Chemistry and Physics: From Air Pollution to Climate Change, John Wiley & Sons, Hoboken, ISBN 978-1-118-94740-1, 2016.

Sillman, S.: The relation between ozone, NOx and hydrocarbons in urban and polluted rural environments, Atmospheric Environment, 33, 1821–1845, https://doi.org/10.1016/S1352-2310(98)00345-8, 1999.

Wang, P., Chen, Y., Hu, J., Zhang, H., and Ying, Q.: Attribution of tropospheric ozone to NOx and VOC emissions: considering ozone formation in the transition regime, Environmental Science & Technology, 53, 1404–1412, https://doi.org/10.1021/acs.est.8b05981, 2019."

Various places – ppb etc are mixing ratios not concentrations

We changed "concentrations" to "mixing ratios" when describing "ppb" throughout the manuscript.

L165 het loss can be important for $HO_2$ removal globally – but wont HO2 + NO dominate under the conditions of these BL measurements?

Yes, $HO_2$+NO dominates under our field measurement conditions, as demonstrated in Fig. S8. However, this dominance refers only to the $HO_2$ removal rate. For $O_3$ production, $HO_2$+NO is a positive source as it simultaneously produces $NO_2$, while $HO_2$ uptake act as a negative source, similar to other termination reactions (e.g., $HO_2$+$HO_2$, $HO_2$+$RO_2$). Therefore, in terms of the negative impact on $P(O_3)_{het}$, the heterogeneous reaction of $HO_2$ with ambient aerosols is more important than $HO_2$+NO.

Therefore, we modified the sentence "Although the heterogeneous uptake of $HO_2$ is not the dominant loss pathway of $HO_2$, it accounts for approximately 10–40 % of global $HO_2$ loss (Li et al., 2019); as a termination reaction, its direct negative impact on photochemical $O_3$ production is non-negligible." in lines 189-191 in the modified manuscript.

References:

Berndt, T., Mentler, B., Scholz, W., Fischer, L., Herrmann, H., Kulmala, M., Hansel A.: 2018. Accretion product formation from Ozonolysis and OH radical reaction of α-Pinene: mechanistic insight and the influence of isoprene and ethylene, Environmental Science & Technology, 52, 11069–11077, doi: 10.1021/acs.est.8b02210, 2018.

Carter, W. P. L., A. Pierce J. A., Luo, D., Malkina, I. L.: Environmental chamber study of maximum incremental reactivities of volatile organic-compounds, Atmospheric Environment, 29, 2499, doi.org/10.1016/1352-2310(95)00149-S, 1995.

Chen, X., Xia, M., Wang, W., Yun, H., Yue, D., and Wang, T.: Fast near-surface $ClNO_2$ production and its impact on $O_3$ formation during a heavy pollution event in South China, Science of The Total Environment, 858, 159998, doi: 10.1016/j.scitotenv.2022.159998, 2023.

Crounse, J. D., Knap, H. C., Ørnsø, K. B., Jørgensen, S., Paulot, F, Kjaergaard, H. G., and Wennberg, P. O.: Atmospheric Fate of Methacrolein. 1. Peroxy Radical Isomerization Following Addition of OH and $O_2$. The Journal of Physical Chemistry A, 116, 5756-5762, doi.org/10.1021/jp211560u, 2012.

Finlayson-Pitts, BJ., Ezell, MJ., and Pitts, JN.: Formation of chemically active chlorine compounds by reactions of atmospheric NaCl particles with gaseous N2O5 and $ClONO_2$, Nature, 337, 241-244, doi:10.1038/337241a0, 1989.

Hao, Y., Zhou, J., Zhou, J. P., Wang, Y., Yang, S., Huangfu, Y., Li, X. B., Zhang, C., Liu, A., Wu, Y., Zhou, Y., Yang, S., Peng, Y., Qi, J., He, X., Song, X., Chen, Y., Yuan, B., and Shao, M.: Measuring and modeling investigation of the net photochemical ozone production rate via an improved dual-channel reaction chamber technique, Atmospheric Chemistry and Physics, 23, 9891-9910, 10.5194/acp-23-9891-2023, 2023.

Ma, W., Feng, Z., Zhan, J., Liu, Y., Liu, P., Liu, C., Ma, Q., Yang, K., Wang, Y., He, H., Kulmala, M., Mu, Y., and Liu, J.: Influence of photochemical loss of volatile organic compounds on understanding ozone formation mechanism, Atmospheric Chemistry and Physics, 22, 4841–4851, doi:10.5194/acp.22.4841, 2022.

McNamara, SM., Kolesar, KR., Wang, S., Kirpes, RM., May, NW., Gunsch, MJ., Cook, RD., Fuentes, JD., Hornbrook, RS., Apel, EC., China, S., Laskin, A., Pratt, and Pratt KA.: Observation of road salt aerosol driving inland wintertime atmospheric chlorine chemistry, ACS Central Science, 6, 684-694, doi:10.1021/acscentsci.9b00994, 2020.

Peng, X., Wang, W., Xia, M., Chen, H., Ravishankara, AR., Li, Q., Saiz-Lopez, A., Liu, P., Zhang, F., Zhang, C., Xue, L., Wang, X., George, C., Wang, J., Mu, Y., Chen, J., and Wang, T.: An unexpected large continental source of reactive bromine and chlorine with significant impact on wintertime air quality, National Science Review, 8, 99-109, doi:10.1093/nsr/nwaa304, 2021.

Spindler, G., Hesper, J., Brüggemann, E., Dubois, R., Müller, T., and Herrmann, H.: Wet annular denuder measurements of nitrous acid: laboratory study of the artefact reaction of $NO_2$ with S(IV) in aqueous solution and comparison with field measurements: Atmospheric Environment, 37, 2643, doi:10.1016/S1352.2310(03)00209.7, 2003.

Stieger, B., Spindler, G., Fahlbusch, B., Müller, K., Grüner, A., Poulain, L., Thöni, L., Seitler, E., Wallasch, M., and Herrmann, H.: Measurements of $PM_{10}$ ions and trace gases with the online system MARGA at the research station Melpitz in Germany – A five-year study, Journal of Atmospheric Chemistry, 75, 33–70, doi.org/10.1007/s10874-017-9361-0, 2018.

Wang, S., Wu, R., Berndt, T., Ehn, M., and Wang, L.: Formation of highly oxidized radicals and multifunctional products from the atmospheric oxidation of Alkylbenzenes, Environmental Science & Technology, 51, 8442-8449, doi:10.1021/acs.est.7b02374, 2017.

Wu, S., Lee, H. J., Anderson, A., Liu, S., Kuwayama, T., Seinfeld, J. H., and Kleeman, M. J.: Direct measurements of ozone response to emissions perturbations in California, Atmospheric Chemistry and Physics, 22, 4929-4949, oi.org/10.5194/acp-22-4929-2022, 2022.

Xu, Z., Liu, Y., Nie, W., Sun, P., Chi, X., and Ding, A.: Evaluating the measurement interference of wet rotating-denuder‒ion chromatography in measuring atmospheric HONO in a highly polluted area: Atmospheric Measurement Techniques, 12, 6737‒6748, doi:/10.5194/amt.12.6737, 2019.

Zhou, J., Wang, W., Wang, Y., Zhou, Z., Lv, X., Zhong, M., Zhong, B., Deng, M., Jiang, B., and Luo, J.: Intercomparison of measured and modelled photochemical ozone production rates: Suggestion of chemistry hypothesis regarding unmeasured VOCs, Science of The Total Environment, 951, 175290, doi:10.1016/j.scitotenv.2024.175290, 2024a.

Zhou, J., Wang, W., Wang, Y., Zhou, Z., Lv, X., Zhong, M., Zhong, B., Deng, M., Jiang, B., Luo, J., Cai, J., Li, X.-B., Yuan, B., and Shao, M.: Intercomparison of measured and modelled photochemical ozone production rates: Suggestion of chemistry hypothesis regarding unmeasured VOCs, Science of The Total Environment, 951, 175290, doi:/10.1016/j.scitotenv.2024.175290, 2024b.